# Nitrifying niche in estuaries is expanded by the plastisphere

Xiaoxuan Su[1,2,3,14], Xinrong Huang[1,2,4,14], Yiyue Zhang [1,2,14], Leyang Yang[1,2,4], Teng Wen[5], Xiaoru Yang[1,2,4], Guibing Zhu [4,6], Jinbo Zhang[5,7], Yijia Tang [8], Zhaolei Li[3], Jing Ding[9], Ruilong Li[10], Junliang Pan [11], Xinping Chen [3], Fuyi Huang [1,2], Matthias C. Rillig [12,13] & Yong-guan Zhu [1,2,4,6] ✉

The estuarine plastisphere, a novel ecological habitat in the Anthropocene, has garnered global concerns. Recent geochemical evidence has pointed out its potential role in influencing nitrogen biogeochemistry. However, the biogeochemical significance of the plastisphere and its mechanisms regulating nitrogen cycling remain elusive. Using [15]N- and [13]C-labelling coupled with metagenomics and metatranscriptomics, here we unveil that the plastisphere likely acts as an underappreciated nitrifying niche in estuarine ecosystems, exhibiting a 0.9 ~ 12-fold higher activity of bacteria-mediated nitrification compared to surrounding seawater and other biofilms (stone, wood and glass biofilms). The shift of active nitrifiers from $O_2$-sensitive nitrifiers in the seawater to nitrifiers with versatile metabolisms in the plastisphere, combined with the potential interspecific cooperation of nitrifying substrate exchange observed among the plastisphere nitrifiers, collectively results in the unique nitrifying niche. Our findings highlight the plastisphere as an emerging nitrifying niche in estuarine environment, and deepen the mechanistic understanding of its contribution to marine biogeochemistry.

Biofilms represent a crucial microbial life mode in oceans[1]. The microorganisms found in biofilms encompass diverse microbial kingdoms (i.e., bacteria, fungi, archaea, protists, viruses), contributing markedly to global marine biogeochemical fluxes[1,2]. Beyond natural biofilms developed on stone debris and floating wood pieces, an emerging "artificial" biofilm (microbial colonization on plastic surfaces) termed as the plastisphere[3–5] has elicited widespread interest amidst escalating plastic pollution on the planet[6–8]. With massive amounts of plastic debris entering oceans through estuaries, there is a potentially important threat to estuarine organisms and ecosystem stability[9–11]. Hence, the biological consequences of the estuarine plastisphere and ecosystem-level impacts of this floating plastic debris warrant increased attention.

Microbial biomass of the "artificial" plastisphere biofilm is substantial within the global marine environment[3], rivaling that of natural biofilms on stone debris and floating wood pieces ($10^8 \sim 10^{11}$ cells g$^{-1}$

[1]Key Laboratory of Urban Environment and Health, Ningbo Observation and Research Station, Institute of Urban Environment, Chinese Academy of Sciences, Xiamen, China. [2]Zhejiang Key Laboratory of Urban Environmental Processes and Pollution Control, CAS Haixi Industrial Technology Innovation Center in Beilun, Ningbo, China. [3]Interdisciplinary Research Center for Agriculture Green Development in Yangtze River Basin, Southwest University, Chongqing 400715, China. [4]University of Chinese Academy of Sciences, 19A Yuquan Road, 100049 Beijing, China. [5]School of Geography, Nanjing Normal University, Nanjing 210023, China. [6]Research Center for Eco-Environmental Sciences, Chinese Academy of Sciences, 100085 Beijing, China. [7]Liebig Centre for Agroecology and Climate Impact Research, Justus Liebig University, Gießen, Germany. [8]School of Life and Environmental Sciences, The University of Sydney, Sydney, NSW 2015, Australia. [9]School of Environmental and Material Engineering, Yantai University, Yantai 264005, China. [10]School of Marine Science, Guangxi University, Nanning 530004, China. [11]School of Electrical Engineering, Chongqing University, Chongqing 400044, China. [12]Freie Universität Berlin, Institute of Biology, Berlin, Germany. [13]Berlin-Brandenburg Institute of Advanced Biodiversity Research, Berlin, Germany. [14]These authors contributed equally: Xiaoxuan Su, Xinrong Huang, Yiyue Zhang. ✉ e-mail: ygzhu@rcees.ac.cn

wet weight)[2]. We therefore need to explore if the plastisphere mirrors the biogeochemical potential of natural biofilms in estuarine ecosystems or exhibits unique characteristics. In particular, extracellular polymeric substances (EPS) secreted by sessile microorganisms on the plastic surfaces could generate $O_2$ and nutrient gradients in the plastisphere biofilms[12,13]. As a consequence, microbes residing within the self-produced matrix of the plastisphere likely exhibit biogeochemical features different from their planktonic counterparts[3,12]. Thus, comparison of the characteristics between plastisphere biofilms (sessile mode) and surrounding seawater (planktonic mode) is vital for the evaluation of biogeochemical fluxes and ecological effects in estuarine ecosystems.

Nitrification is an important part of the estuarine nitrogen cycle, which impacts primary productivity and maintains global nitrogen balance[14]. The process is conventionally carried out by two separate microbial guilds, consisting of ammonia oxidizers and nitrite oxidizers. Most studies have focused on ammonia-oxidizing archaea (AOA) and bacteria (AOB) because ammonia oxidation is generally regarded as a rate-limiting nitrification step. By contrast, nitrite oxidation receives less attention due to the difficulty of obtaining pure cultures of the organisms, and because of its sole function of transforming nitrite to nitrate[15]. The finding of complete ammonia oxidizers (Comammox, COM) within the *Nitrospira* genus[16] has stimulated global interest in exploring nitrite-oxidizing bacteria (NOB) and *Nitrospira* COM nitrifiers. In estuarine environment, the plastisphere and the surrounding water column differ in physical and chemical properties. It is still unknown whether the four types of nitrifiers (AOA, AOB, NOB and COM) possess niche preference for plastic surfaces or the water column, and if their communities make different contributions to marine nitrification.

Nitrification can cause the emission of the greenhouse gas $N_2O$ as a byproduct, but exists different mechanisms in AOB and AOA. $N_2O$ emission from AOB strains occurs via hydroxylamine ($NH_2OH$) oxidation and nitrifier denitrification pathways[17-19], whereas AOA-mediated emission primarily arises from abiotic (hybrid) formation, involving one N atom from $NH_2OH$ and another from $NO_2^{-}$[20,21]. Notably, these pathways are subject to different redox conditions. The redox potential controlling $NH_2OH$ oxidation and hybrid formation is around 20% $O_2$, while nitrifier denitrification-derived $N_2O$ emission occurs just with over 0.5% $O_2$[17,22]. Given the microsite gradients of $O_2$ concentrations within the plastisphere biofilms[3,12,13], $N_2O$ dynamics in these biofilms may diverge from those in surrounding water column. The contribution of COM nitrifiers to $N_2O$ emission remains contentious[23], with some suggesting a non-negligible contribution despite lower levels of $N_2O$[24], while others arguing that COM nitrifiers do not produce $N_2O$[25]. This discrepancy largely hinges on the biomass of COM nitrifiers and environmental factors[23]. With diverse microenvironments present within the plastisphere biofilms, nitrification-derived $N_2O$ emissions may vary between the plastisphere and surface seawater; however, this has scarcely been characterized.

Here we select three estuarine regions in China (Fig. 1a) and conduct a series of in-situ incubations and lab-scale experiments based on biofilm type (plastic, glass, stone and wood) and plastic type (polyethylene, polystyrene and polyvinylchloride) to investigate the nitrification potential, and then to compare the core nitrifiers between the plastisphere biofilms (sessile mode) and surrounding seawater (planktonic mode). The experimental workflow of this study is outlined in Supplementary Fig. 1. The specialized features of the plastisphere lead us to hypothesize that (1) the estuarine plastisphere represents an overlooked and even unique niche of nitrification with higher nitrifying activity than the surrounding seawater and other biofilms, and (2) it harbors distinctive active nitrifiers and metabolic behaviors from the seawater. To test the hypotheses, we measure nitrification rates ($NH_3$ oxidation and $NO_2^{-}$ oxidation), and $N_2O$ emission and related pathways ($NH_2OH$ oxidation and nitrifier

denitrification) using $^{15}N$ isotope tracing and $N_2O$ isotopocules methods. Next, employing $^{13}C$-DNA stable isotope probing (DNA-SIP) and sequencing of amplicons and metagenomes, we identify active nitrifier communities. Finally, we reveal the potential metabolic differences of these active nitrifiers between the plastisphere (sessile mode) and seawater (planktonic mode) using metagenome-assembled genomes (MAGs)-centric metatranscriptomic analyses. Our study offers insight into biogeochemical significance of the plastisphere in the Earth system, and reveals the distinctive metabolic mechanisms of sessile nitrifiers in this new plastic niche.

## Results
### Nitrifying activity in different biofilms
Four types of plastic and other non-plastic materials, including plastic bags, glass balls, stone and wood debris, were placed in the three estuaries (Xiamen XM, Yan YT and Nanning NN sites) spanning a distance of 1870 km for 28 days (Fig. 1a, b). Microbial aggregates densely adhered to stone surfaces (Supplementary Fig. 2). The biofilms were less abundant on the glass surfaces, and were loose and easily dispersed on the wood surfaces. The stone and wood biofilms harbored more microbial biomass and nitrifiers (AOA and AOB) than the plastisphere and seawater ($P < 0.001$, Supplementary Figs. 3 and 4). Nitrifier biomass in the glass biofilms was the lowest. Except at NN site, AOB abundances ($0.16 \times 10^4$ - $0.72 \times 10^6$ copies $L^{-1}$) consistently surpassed those of AOA ($0.54 \times 10^2$ - $0.21 \times 10^4$ copies $L^{-1}$) at XM and YT sites, regardless of biofilms and seawater (Supplementary Fig. 4), indicating the predominance of AOB.

We subsequently conducted a 36-h incubation to explore nitrification potential of different biofilms and estuarine seawater. During the incubation, the headspace $O_2$ concentrations decreased rapidly from 26% to 3% (Supplementary Fig. 5a), but oxic conditions were still maintained throughout the experiment. Among the three sampling sites, variations of $NH_4^{+}$ and $NO_3^{-}$ concentrations in the plastisphere and stone biofilms were more pronounced than in the glass and wood biofilms as well as in the seawater (Supplementary Fig. 6). Notable $NO_2^{-}$ accumulation was observed in the plastisphere and stone biofilms, except at the YT site. The average rates of ammonia and nitrite oxidation in the plastisphere were 1.79 - 3.59 fmol cell $h^{-1}$ and 1.39 - 3.71 fmol cell $h^{-1}$, respectively, significantly higher than in the glass, wood biofilms and the surrounding seawater ($P < 0.001$ - 0.028, Fig. 1c-e). The rates of glass and wood biofilms were lower than those of the surroundings at all sites ($P = 0.002$ - 0.072). More importantly, we found that despite stone biofilms harboring more nitrifying biomass (Supplementary Figs. 3 and 4), their nitrification rates were comparable ($P > 0.05$, Fig. 1d, e) or even lower ($P = 0.004$, Fig. 1c) than those in the plastisphere.

At the end of the incubation, nitrification-derived $N_2O$ emission from the plastisphere was significantly higher than that from stone biofilm at the XM site ($P < 0.001$, Fig. 2a), and also higher than emissions from other biofilms and the surrounding seawater at all sites ($P < 0.001$, Fig. 2a-c). We further used $N_2O$ isotopocules coupled with the $N_2O$-SP value to discern the relative contributions of each $N_2O$ emission pathway during nitrification, including $NH_2OH$ oxidation and nitrifier denitrification[14]. The values of SP, $^{15}N^{\alpha}$ and $^{15}N^{\beta}$ of $N_2O$ are shown in Supplementary Fig. 7. By calculating the $N_2O$-SP ($\delta^{15}N^{\alpha}$-$\delta^{15}N^{\beta}$), we found that the average values in the biofilms (4.58-9.46‰) were remarkably lower than those in the seawater (19.81-26.68‰, $P < 0.001$, Supplementary Fig. 7), implying distinct $N_2O$ emission dynamics in the biofilms. The primary $N_2O$ emission in the biofilms was from nitrifier denitrification, contributing 53-70% of the total emissions (Fig. 2e); however, $NH_2OH$ oxidation dominated in the seawater (62-79%). To elucidate contributions of AOB and AOA to the ammonia oxidation process, we conducted an additional 36-h incubation with and without adding 100 μM of penicillin which inhibits bacteria but not archaea[26]. With the addition of penicillin

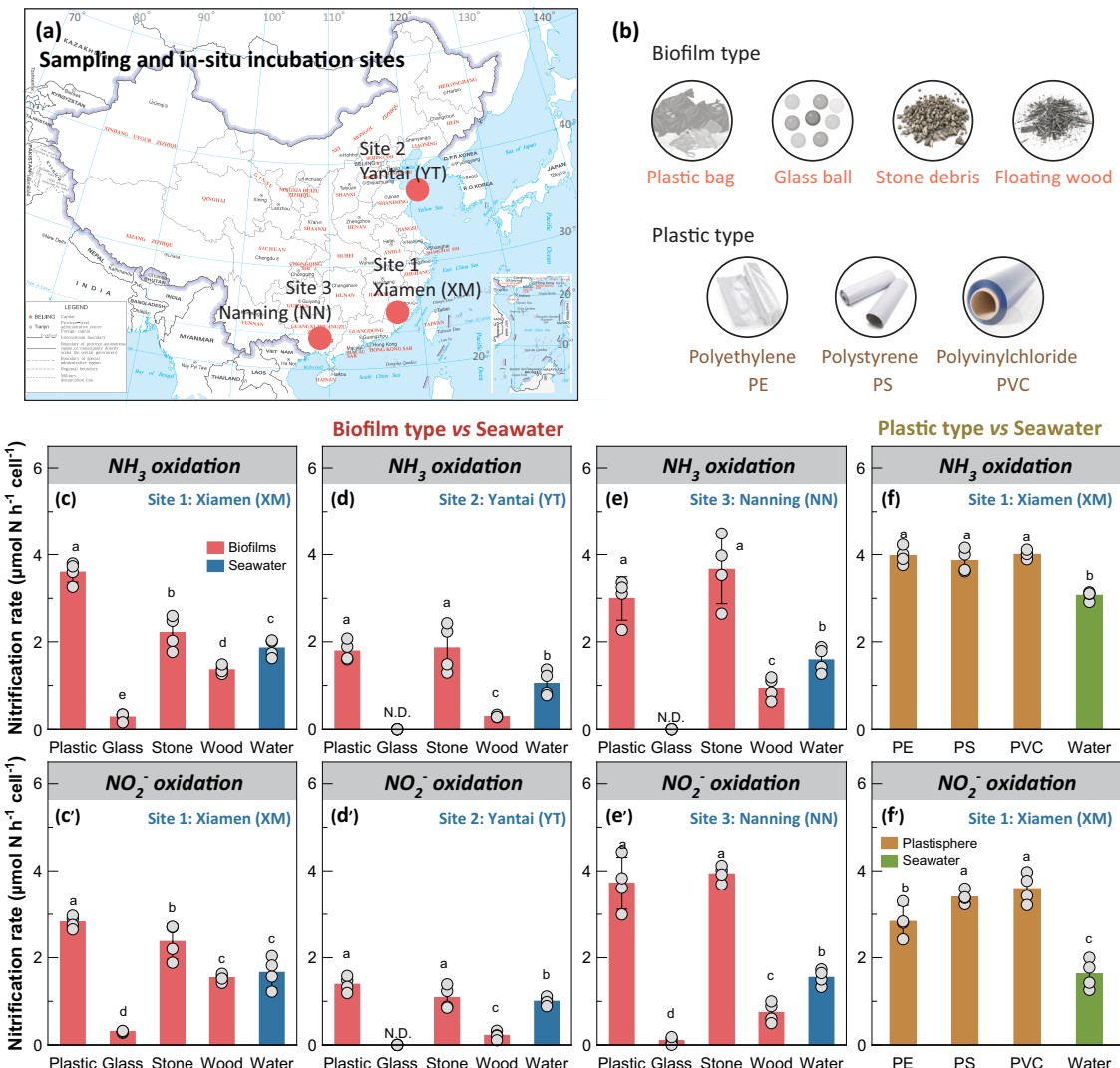

**Fig. 1 | Sampling and incubation sites, biofilm type- and plastic type-based materials, and nitrification rates. a** Sampling and in-situ incubation sites, including site 1: Xiamen (XM) in Fujian Province (118°11′E, 24°57′N), site 2: Yantai (YT) in Shandong province (121°47′E, 37°46′N) and site 3: Nanning (NN) in Guangxi Province (108°27′E, 22°84′N). **b** Biofilm type materials include plastic bags, glass balls, stone debris and wood debris; plastic type materials include PE, PS, and PVC.

**c–f** Ammonia oxidation rates ($n = 4$, biological replicates) and nitrite oxidation rates ($n = 4$, biological replicates) in different biofilms, plastisphere types and surrounding seawater across the three sampling sites. Data are presented as mean value ± standard deviation. Different letters indicate the significant differences (one-way ANOVA, $P < 0.001 - 0.045$).

(Supplementary Fig. 8), 67-85% of bacteria-mediated $NH_4^+$ transformation was significantly inhibited, further indicating a dominant contribution from AOB. In combination, these findings indicate that plastisphere exhibited a greater bacterial nitrifying potential compared to wood and glass biofilms as well as the surrounding seawater, acting as an overlooked nitrifying niche in estuarine ecosystems. Stone biofilms harbored more nitrifier biomass yet comparable activity to the plastisphere at most sites, further highlighting the distinct niche of this artificial interface.

## Nitrifying activity in different plastisphere

We further explored the effects of plastisphere types on nitrification potential (Supplementary Fig. 9). Three types of plastic debris (PE, PS and PVC) were incubated in XM estuarine seawater for 28 days, followed by a 36-h incubation. Oxic conditions were still maintained throughout the experiment (Supplementary Fig. 5b). Consistent with the results from Experiment 2, the average rates of ammonia and nitrite oxidation in the plastisphere were higher than in the surroundings ($P < 0.01$, Fig. 1f). No significant difference in nitrification

rates was observed between the plastisphere types ($P > 0.05$), except the lower nitrite oxidation rate in the PE plastisphere ($P = 0.041$, Fig. 1f).

Nitrification-derived $N_2O$ emission from the plastisphere (3.5–4.7 fmol $N_2O$ cell$^{-1}$) after 36 h was 1.6–2.2-fold higher than that from the surrounding seawater (2.2 fmol $N_2O$ cell$^{-1}$, $P < 0.001$, Fig. 2d). Furthermore, the emission via PVC plastisphere was slightly higher than from PE and PS plastisphere ($P = 0.014$ and 0.042, Fig. 2d). Similar with previous results (Fig. 2e), nitrifier denitrification dominated the $N_2O$ emission in the plastisphere (67–77%) and $NH_2OH$ oxidation was the main source in the seawater (58–65%, Fig. 2f), with AOB-derived $NH_2OH$ oxidation (77–85%) contributing more than the AOA-derived path (15–23%). No significant differences in pathway contribution of $N_2O$ emission were found between the three types of plastisphere ($P = 0.23$-0.86). Overall, our results demonstrate that while the type of plastisphere exerted a minimal influence on the nitrification process, all the plastisphere exhibited bacterial nitrifying activities surpassing those of the surrounding seawater. This implies that sessile-mode bacterial nitrifiers likely display an enhanced activity compared to their planktonic counterparts. To confirm this, below we will explore

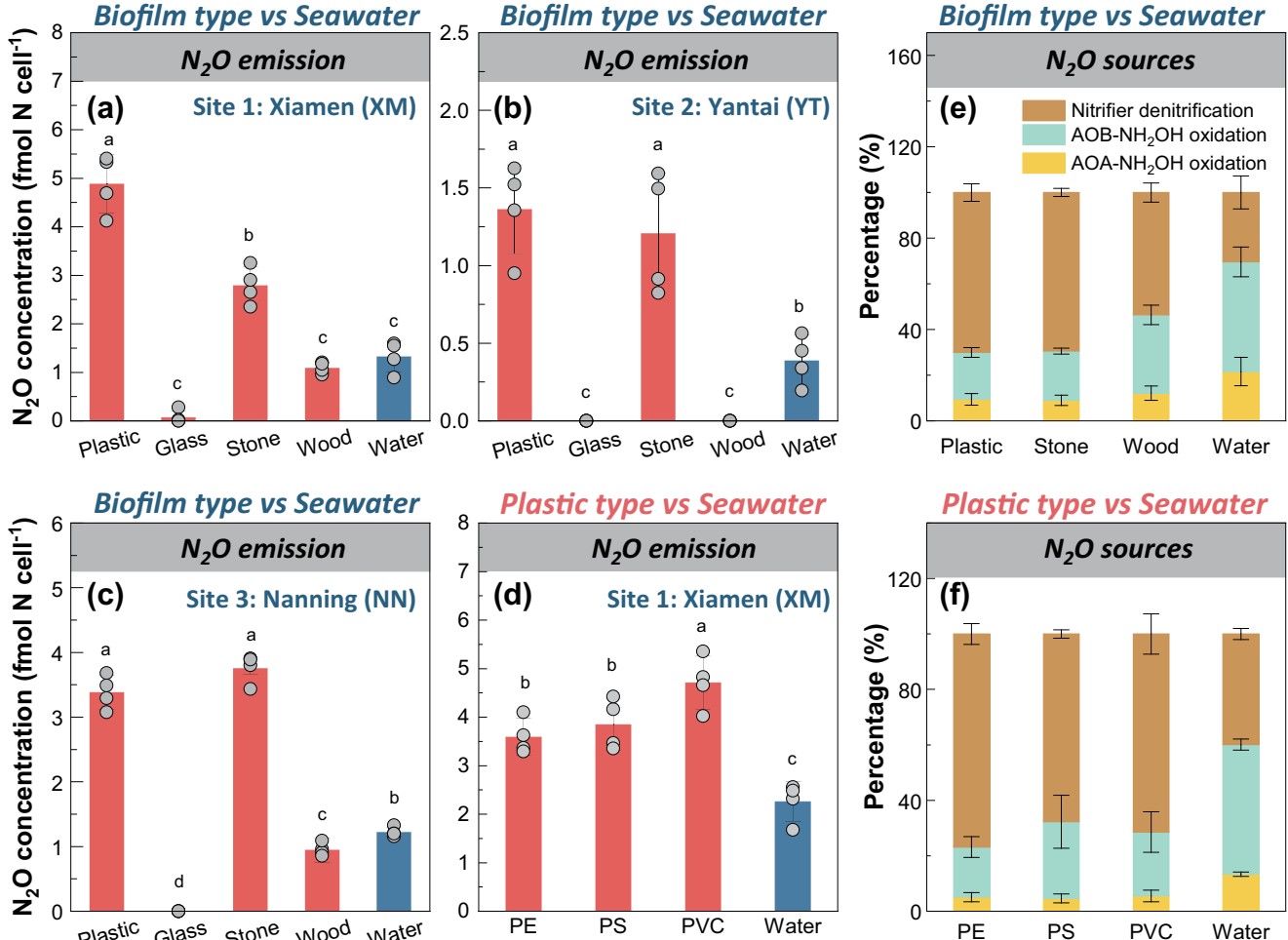

**Fig. 2 | Nitrification-derived N2O emission and its source tracing. a–c** N2O emission of different biofilms after the 36-h incubation across the three sampling sites (Experiment 2, $n = 4$, biological replicates). **d** N2O emission of different plastisphere after the 36-h incubation (Experiment 3, $n = 4$, biological replicates). **e, f** The relative contributions of AOA and AOB-derived $NH_2OH$ oxidation and nitrifier denitrification to N2O emission during nitrification in different biofilms and plastisphere (Methods 2.3, $n = 4$, biological replicates). Calculation errors were estimated using Monte Carlo simulation, which is shown in Supplementary Fig. 19. Nitrifier denitrification dominates N2O emission in the biofilms, while $NH_2OH$ oxidation (mainly as the AOB-derived) is the major N2O source in the seawater. Data are presented as mean value ± standard deviation. Different letters indicate the significant differences (one-way ANOVA, $P < 0.001 - 0.042$).

biogeochemical mechanisms inherent to the plastisphere and compare them with those in seawater.

### Niche differentiation of active nitrifiers

To identify the active nitrifiers, mainly nitrifying bacteria, in the plastisphere and surrounding seawater, a 30-d flush-feeding incubation was conducted by adding 5% $^{12}CO_2/^{13}CO_2$ and 2 mM $NaH^{12}CO_3/NaH^{13}CO_3$ (Supplementary Fig. 10). Following ultracentrifugation, the labeled $^{13}C$-DNA and the control $^{12}C$-DNA were collected. AOB abundances were the highest, followed by AOA (Fig. 3a). COM nitrifiers presented markedly lower abundances than both AOA and AOB. Thus, quantification of AOA- and AOB-*amoA* gene abundances as a function of CsCl-DNA buoyant density was performed to illustrate the labeling of active nitrifiers (Fig. 3a and Supplementary Fig. 11). High peaks of active AOB in the $^{13}C$-DNA were observed in buoyant density 1.699–1.702 g mL$^{-1}$, while they remained 1.680–1.686 g mL$^{-1}$ in the $^{12}C$-microcosms in the plastisphere or seawater samples. The shift in CsCl buoyant density from the $^{12}C$ to the $^{13}C$ genomes for AOB is 0.016-0.022 g mL$^{-1}$. For active AOA, the peak buoyant densities of $^{13}C$-DNA and $^{12}C$-DNA were in 1.695–1.699 and 1.687–1.695 g mL$^{-1}$, respectively (Supplementary Fig. 11). This minimal change (0.008–0.012 g mL$^{-1}$) of buoyant density suggested a poor

fractionation of AOA, in comparison to AOB, across both plastisphere and seawater. Sequencing of metagenomes, 16S rRNA, AOA-*amoA*, AOB-*amoA*, and COM-*amoA* was subsequently performed on fractions 9-10 of the $^{13}C$-DNA samples to explore the composition of the active nitrifiers. No labeling of NOB-*nxrA/B* was detected though we used different primers.

Compositions and abundances of active nitrifiers revealed by $^{13}C$-DNA-based metagenomics and metatranscriptomics showed that most sequences affiliated to nitrifiers were nitrifying bacteria including AOB, NOB and Comammox (COM) in both the plastisphere and seawater, whereas nitrifying archaea were observed less (Fig. 3b, c). This further underscores the dominant role of nitrifying bacteria in the nitrification process. Microbial community structure showed that the plastisphere hosted a distinct nitrifier community compared to seawater (Fig. 3c and Supplementary Fig. 12). At the genus level, the major AOB and NOB in the plastisphere were *Nitrosospira*, *Nitrobacter* and *Nitrospira*, with abundances of 8.7%, 2.5% and 8.8% at the gene level and 4.4%, 3.6% and 5.8% at the transcript level, respectively (Fig. 3b). These *Nitrosospira*-like and *Nitrobacter*-like nitrifiers included *Nitrosospira_sp.*, unclassified_*Nitrosospira* and *Nitrobacter_sp.* (Supplementary Fig. 13). For seawater, the most abundant nitrifiers were *Nitrosomonas* (13.2% and 5.7%) and *Nitrotoga* (4.7% and 4.0%) at the gene and transcript levels

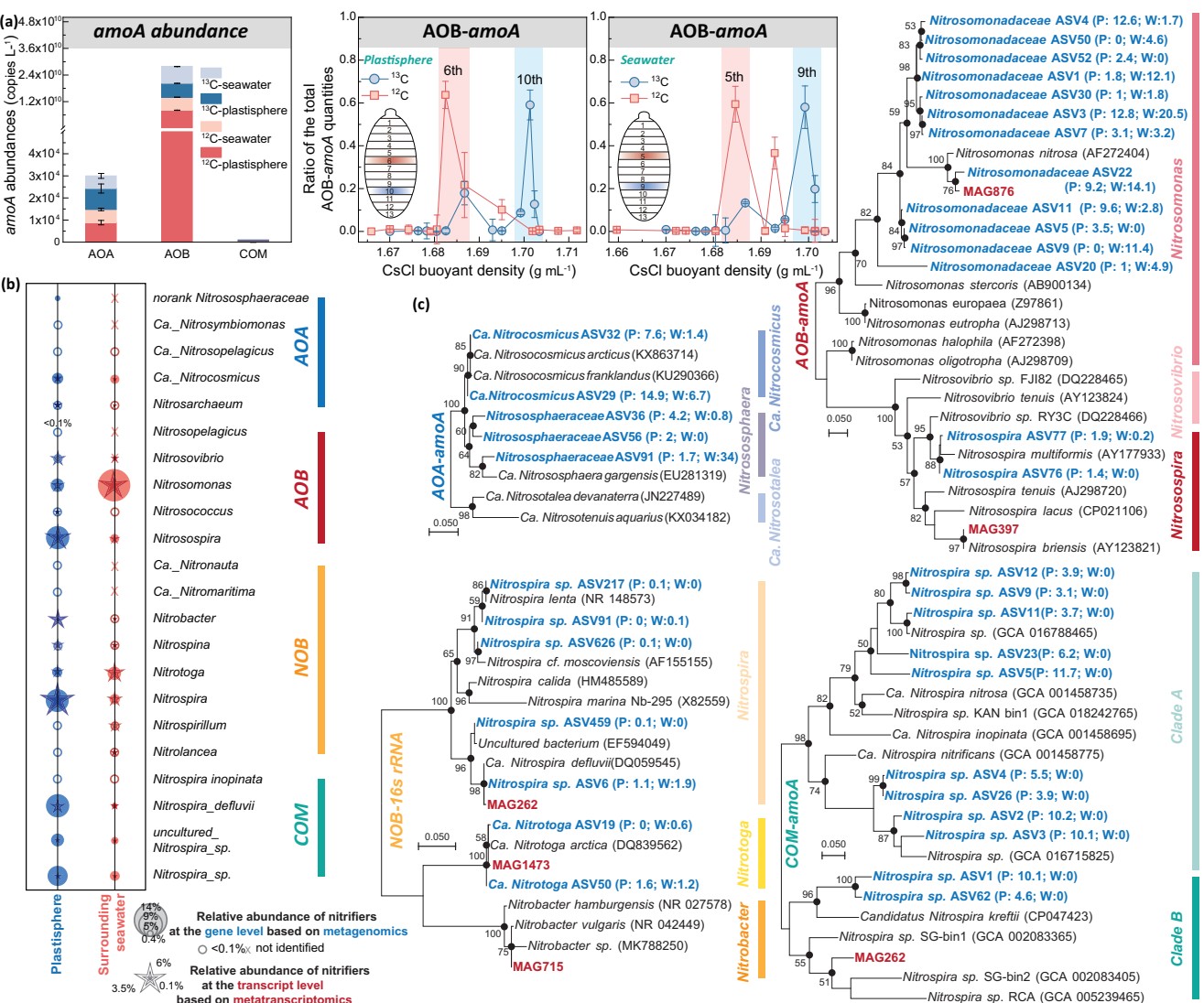

**Fig. 3 | Active nitrifiers in the plastisphere and the surrounding seawater.**
**a** DNA-SIP ($n = 3$, biological replicates). The $^{13}$C-DNA (heavy) and the $^{12}$C-DNA (light) of AOB are shown by qPCR of *amoA* across the CsCl buoyant density gradient after the 30-d flush-feeding incubation (Experiment 4). The results are normalized using the ratio of AOB-*amoA* copy number in each DNA fraction to the total AOB-*amoA* copy numbers of all fractions in each sample (AOA see Supplementary Fig. 11). The $^{13}$C-DNA and the $^{12}$C-DNA are accumulated in buoyant density 1.699–1.702 and 1.680–1.686 g mL$^{-1}$, respectively, in both the plastisphere and seawater. Data are presented as mean value ± standard deviation. **b** Compositions (%) of the active nitrifiers including AOA, AOB, NOB and COM bacteria, based on the sequencing of metagenomics with $^{13}$C-DNA (circle) and the metatranscriptomics (star). The species of the major genera, such as *Nitrosomonas, Nitrosospira, Nitrobacter,* and *Nitrotoga* are shown in Supplementary Fig. 13. **c** Phylogenetic analysis of active nitrifiers based on the sequencing of AOA-*amoA*, AOB-*amoA*, COM-amoA and NOB-16S rRNA (maximum-likelihood method). For MAGs, AOB-*amoA*, COM-*amoA* or 16S rRNA sequences were extracted and then integrated into the trees. Species in black, blue and red are the sequences from NCBI database, amplicon sequence data, and MAGs associated with nitrifiers, respectively. Labels "P" and "W" represent the plastisphere and seawater origins, respectively. For example, the designation of .... ASV32 (P: 7.6; W:1.4) indicates its proportion of the total sequences·7.6% in the plastisphere and 1.4% in seawater. Bootstrap support values exceeding 50% are noted at branching nodes, based on 1000 replicates. Information of all MAGs is detailed in Fig. 4.

(Fig. 3b). *Nitrosomonas* sp., *Nitrosomonas_nitrosa, Nitrotoga_MKT* and *Nitrotoga fabula* were the main species in the seawater. Notably, the plastisphere surprisingly harbored more abundant COM nitrifiers than the seawater (Fig. 3b). Based on the metagenomics and metatranscriptomics, the most abundant COM nitrifiers containing *amoABC* and *nxrAB* genes were *Nitrospira_defluvii* (8.6% and 2.0%), followed by some uncharacterized *Nitrospira* members (3.9–6.5% and 0.08–0.12%) (Fig. 3b). The compositions of these active nitrifiers were further supported by the amplicon sequencing analyses of 16S rRNA and COM-*amoA* genes (Supplementary Fig. 14).

## MAG-centric transcriptomic analysis of active nitrifiers

Metagenomic data were binned into 67 metagenome-assembled genomes (MAGs), of which 46 were recovered as medium-quality MAGs (completeness:>75%, contamination:<10%) and high-quality MAGs (completeness:>90%, contamination:<5%, Supplementary Data). The abundances and expressions of each MAG are depicted in Fig. 4, with most MAGs affiliated to Proteobacteria and Bacteriodota. Five medium- and high-quality MAGs related to nitrification were obtained, including MAG262 (*Nitrospira*), MAG397 (*Nitrosospira*) and MAG715 (*Nitrobacter*) in the plastisphere, and MAG876 (*Nitrosomonas*) and MAG1473 (*Nitrotoga*) in the seawater (Fig. 4). The average nucleotide identity (ANI) of all MAGs exceeded 96%, suggesting that the five MAGs could closely represent the associated nitrifying strains within the same species. Only MAG262 *Nitrospira* was assigned to a specific species *Nitrospira_defluvii* (Fig. 4). This was also corroborated by the phylogenetic analysis and the amplicon sequencing results (Fig. 3c).

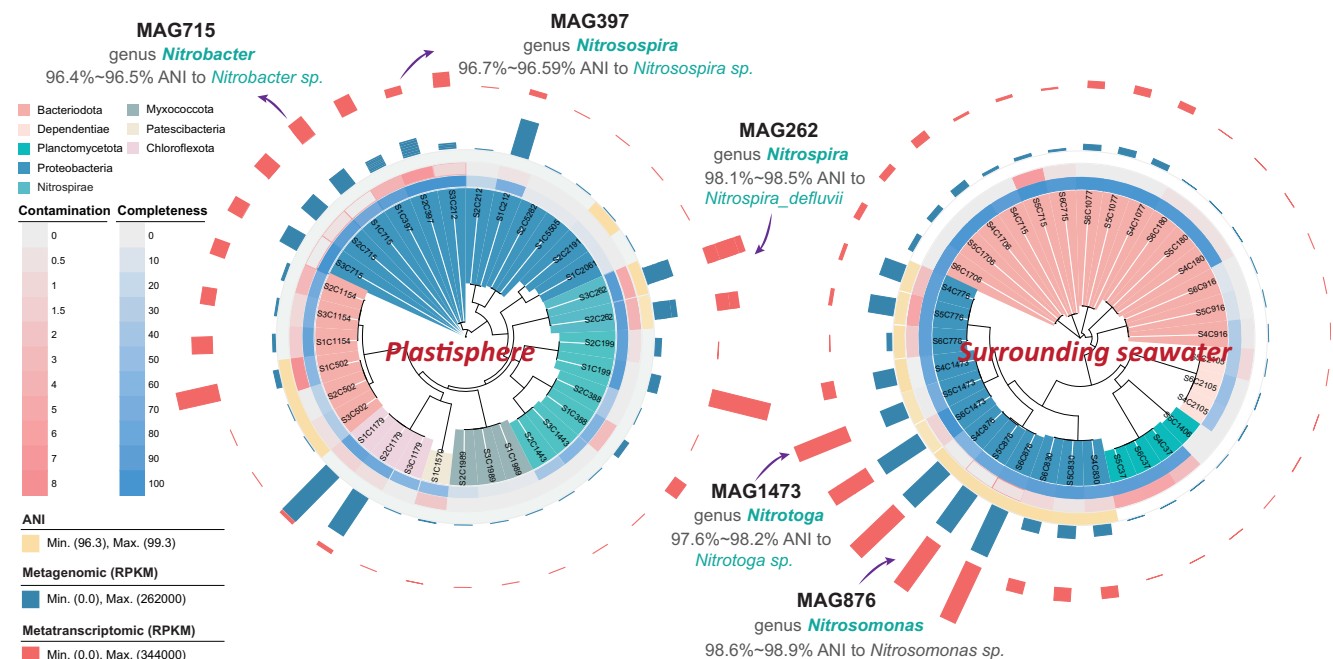

**Fig. 4 | Phylogenetic tree of metagenome assembled genomes (MAGs) in the plastisphere and the surrounding seawater.** Taxonomic classification was conducted using the Genome Taxonomy Database, and the detailed information of each MAG is provided in the Supplementary data file. Each branch represents a MAG and is annotated to the genus level. The background color of each MAG label indicates phylum information. The completeness, contamination and genome-wide average nucleotide identities (ANI) of MAGs are depicted using heatmap. The relative abundances (blue bar, ¹³C-DNA metagenomics) and expressions (red bar,

metatranscriptomics) of each MAG were calculated based on Reads Per Kilobase of exon model per Million mapped reads (RPKM). A total of five medium- and high-quality MAGs (completeness > 75% and contamination < 10%) associated with nitrification process are identified in the plastisphere (three) and seawater (two). MAG715 (*Nitrobacter*), MAG397 (*Nitrosospira*) and MAG262 (*Nitrospira*) are in the plastisphere; MAG1473 (*Nitrotoga*) and MAG876 (*Nitrosomonas*) are in the surrounding seawater. The five nitrifying MAGs are further analyzed in Fig. 5.

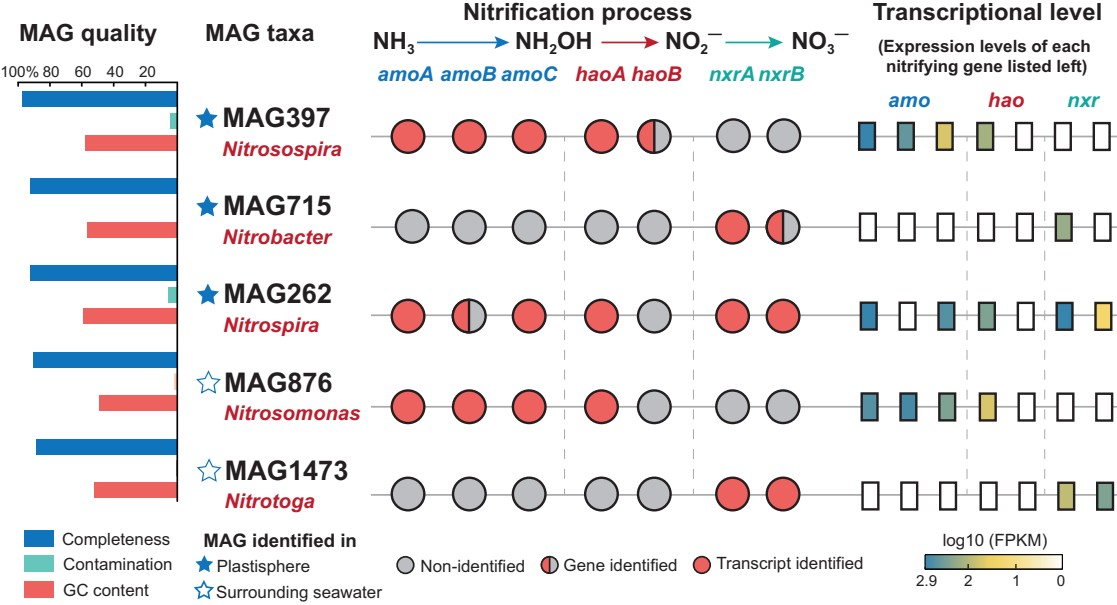

**Fig. 5 | Genomic and transcriptional information of the five MAGs associated with nitrification process in the plastisphere and the surrounding seawater.** The MAG quality includes completeness, contamination, and GC content. Taxonomic classification is based on the Genome Taxonomy Database. The presence or

absence of the seven nitrifying genes is shown at both the gene and transcript levels. The transcriptional levels of each nitrifying gene were calculated on the basis of Fragments Per Kilobase of exon model per Million mapped reads (FPKM).

Genome sketches of the five MAGs are established in Supplementary Fig. 15, with size ranging from 2.0 to 3.5 Mb. A closer inspection of the nitrifying MAGs is shown in Fig. 5. All the completeness and contamination of these five MAGs were over 90% and less

than 9%, respectively. MAG397 (*Nitrosospira*) and MAG876 (*Nitrosomonas*) contained genes related to ammonia oxidation (*amoABC*) and hydroxylamine oxidation (*haoA*). MAG397 also possessed *haoB* gene, but it was not expressed at the transcriptional level (Fig. 5), suggesting

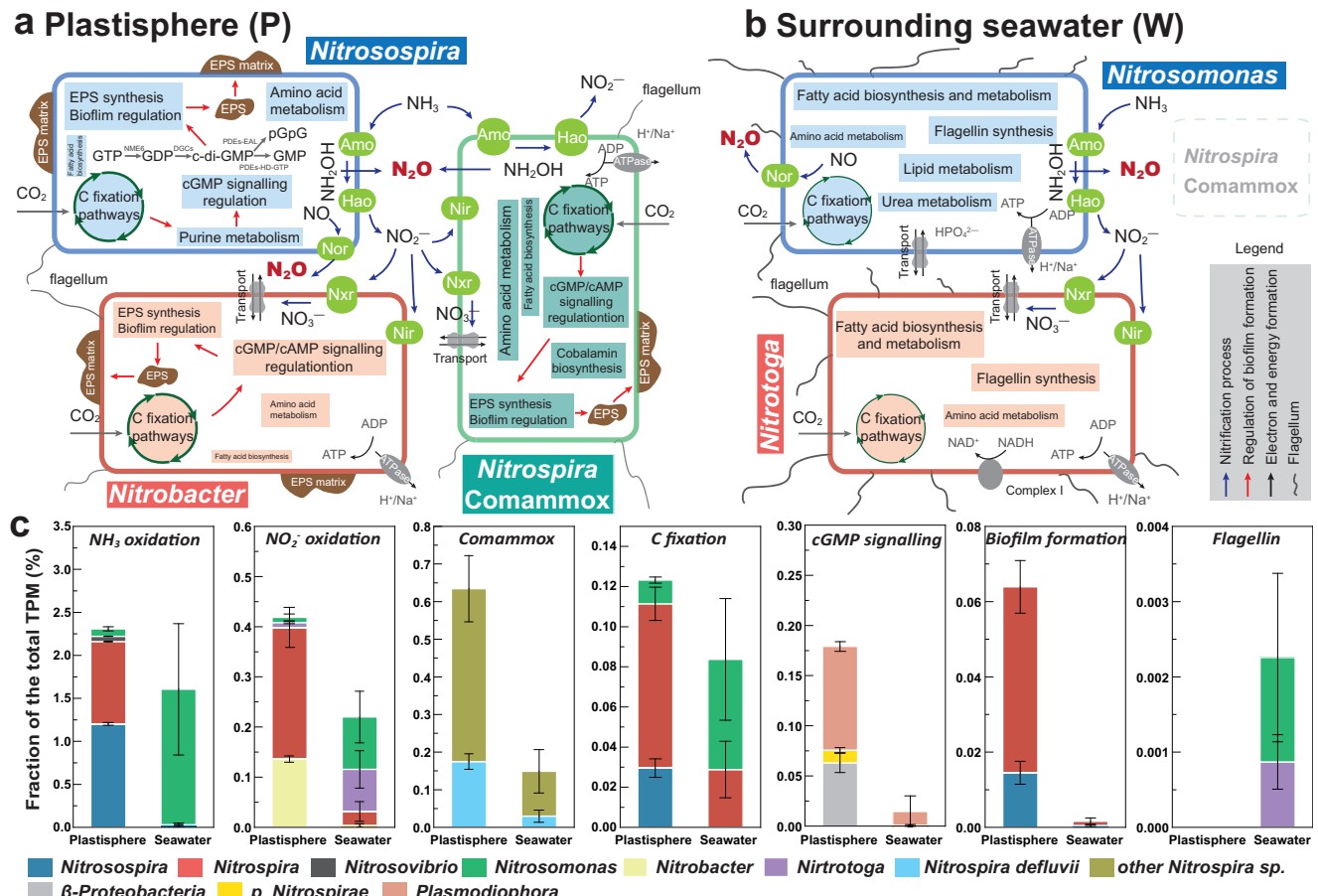

**Fig. 6 | Metatranscriptomics showing the differences in metabolisms and cooperations of active nitrifiers in the plastisphere and the surrounding seawater.** Schematic representations, at the transcriptional level, of microbial metabolisms of the five nitrifying MAGs and their cooperations in the plastisphere (**a**, sessile mode) and surrounding seawater (**b**, planktonic mode). Only the pathways related to plastisphere formation and nitrification process are selected. The size of each metabolism indicates the expression level. EPS extracellular polymeric substances, GTP guanosine triphosphate, GDP guanosine diphosphate, GMP guanosine monophosphate, pGpG 5′-Phosphoguanylyl-(3′,5′)-guanosine. **c** Relative expressions of main active nitrifiers to microbial metabolisms of interest. The total Transcripts Per Million (TPM) in this study was assessed for each target gene ($n = 3$, biological replicates). We used KEGG annotation to identify Open Reading Frames (ORFs) coding for ammonia oxidation, nitrite oxidation, Comammox, carbon fixation pathways, cGMP signaling, biofilm formation, and flagellin. Data are presented as mean value ± standard deviation.

that the nitrifier may regulate the expression of its nitrification machinery to suit the changing microenvironments in the plastisphere. NOB-related MAG715 (*Nitrobacter*) and MAG1473 (*Nitrotoga*) contained the gene of nitrite oxidation (*nxrA* or *nxrB*). We found a full suite of nitrifying genes, including *amoABC*, *haoAB* and *nxrAB*, in the MAG262 (*Nitrospira_defluvii*) recovered from the plastisphere (Fig. 5), with all but *amoB* being expressed at varying levels. No MAGs associated with COM process were retrieved from the surrounding seawater.

**Cooperation and metabolic differences**

Using metatranscriptomics, the nitrifying system and metabolic differences of active nitrifiers between the plastisphere (sessile mode) and seawater (planktonic mode) were further investigated. Compared to the seawater, the plastisphere had a greater nitrification potential at the gene expression level, encompassing $NH_3$ oxidation (KEGG Module: M00528) and COM (M00804, Supplementary Fig. 16). This aligns with the observed nitrification rate and $N_2O$ emission (Figs. 1 and 2). Notably, nitric oxide (NO) reductase transcripts (*norBD*) were detected in *Nitrosospira* (MAG397) and *Nitrosomonas* (MAG876), but not in the COM nitrifier *Nitrospira_defluvii* (MAG262) (Supplementary Fig. 15). This absence suggests an inability of the COM nitrifier to generate $N_2O$ via nitrifier denitrification, likely necessitating cooperation with AOB for $N_2O$ production within the plastisphere.

The nitrifying MAGs-centric metatranscriptomic analysis revealed the metabolic disparities between the plastisphere and seawater nitrifiers (Fig. 6). Carbon fixation pathways, i.e., Calvin cycle and reductive TCA cycle, were highly expressed in the plastisphere nitrifiers (MAG262, MAG397, MAG715, Fig. 6). The genes *sdhB*, *acnA*, *pcrA*, *pycB* and *rbcL* involved in carbon fixation showed higher transcriptional levels (Supplementary Fig. 17). The elevated transcriptional activity of amino acid metabolisms containing *mmsB* and *aroE* genes was also observed in *Nitrospira_defluvii* (MAG262) and *Nitrosospira* (MAG397) within the plastisphere. Conversely, the transcriptional activity of lipid metabolism and fatty acid metabolism (*fabD*, *fabY*, *ACACA* and *ATS1* genes) in all plastisphere MAGs was typically lower than in the seawater MAGs (Supplementary Fig. 17). A highly-expressed pathway related to cobalamin biosynthesis was detected in the COM nitrifier *Nitrospira_defluvii* (Fig. 6a). Cobalamin (vitamin B12) is an essential cofactor in intracellular primary and secondary metabolisms[27], and the elevated expression in the COM nitrifier can relieve the toxicity of accumulated $NO_2^-$ in the plastisphere[28]. Importantly, we found that EPS synthesis, purine metabolism and quorum sensing (i.e., cGMP/cAMP signaling) pathways were robustly expressed in the three plastisphere nitrifiers (Fig. 6a, b), whereas *FlrC* and *filC* genes regulating flagellin synthesis had substantially lower expression

levels in the plastisphere. These pathways are vital for the formation of plastisphere biofilms[29–31].

Relative expression of each gene involved in nitrification process and metabolisms of interest was calculated (Fig. 6c). Members of *Nitrosospira*, *Nitrobacter* and *Nitrospira* expressed genes for ammonia and nitrite oxidation at a high level in the plastisphere (0.13–1.2% of the total transcripts per million, TPM), while *Nitrosomonas* and *Nitrotoga* predominantly expressed these genes in the seawater (0.1–1.6% of TPM). Other active nitrifiers also expressed these genes but at lower levels (Fig. 6c). The expression of COM-*amoA* gene in *Nitrospira_defluvii* and other *Nitrospira*_sp. was more in the plastisphere than in the surroundings, supporting the notion that the plastisphere is a hotspot of the COM process. In addition, relative expressions of genes associated with biofilm formation in the plastisphere nitrifiers (*Nitrosospira*, *Nitrospira*, *β-Proteobacteria*) were substantially higher than those in the seawater (Fig. 6c). *Nitrosomonas* and *Nitrotoga* in the seawater were the main nitrifiers in regulating the pathway of flagellin synthesis, with no equivalent activity detected in the plastisphere, suggesting stable adherence of active nitrifiers to plastic surfaces.

## Discussion

Estuarine ecosystems offer vast surfaces for microbial colonization and growth, fostering dense microbial networks that drive key biochemical processes[1,2]. The plastisphere as a footprint of anthropogenic activities represents a new artificial interface[3], and its role and significance in biogeochemical cycling are igniting global interest[32–34]. Our results show that the estuarine plastisphere exhibited heightened bacterial nitrifying activity relative to the glass and wood biofilms (Figs. 1 and 2). Although stone biofilms hosted abundant nitrifying biomass, their nitrifying activities were either comparable or inferior to those in the plastisphere, further underscoring the high nitrification potential of the plastisphere. Notably, we find that the sessile nitrifiers within the plastisphere presented an elevated nitrifying activity compared to the planktonic counterparts in surrounding seawater. The striking niche partitioning among the active core nitrifiers, especially COM nitrifiers, was further observed between the plastisphere and seawater (Fig. 3). The plastisphere harbored more biomass of COM nitrifiers, which also had greater expression activities. Through the analysis of medium- and high-quality MAGs, we identified a COM nitrifier *Nitrospira_defluvii* (MAG262) in the plastisphere but failed to assemble such nitrifiers in the seawater (Figs. 4 and 5). These insights corroborate our hypotheses, highlighting the plastisphere as an underestimated and unique nitrifying niche in estuarine environments.

Dense biofilm formation enriching planktonic nitrifiers, microsite $O_2$ environments yielding diverse redox conditions, and spatial structure of biofilms improving cooperations and interactions among sessile nitrifiers in the plastisphere collectively facilitate the formation of the unique nitrifying niche. In this study, we found that carbon fixation pathways (i.e., Calvin cycle, reductive TCA cycle etc.) were highly expressed in the plastisphere nitrifiers (Fig. 6a). More importantly, these sessile nitrifiers also exhibited significant transcriptional activity in quorum sensing pathways such as purine metabolism, cGMP/cAMP signaling and EPS synthesis pathways, which are crucial for biofilm formation (Fig. 6). It has been reported that microbial c-di-GMP level regulated by the cGMP/cAMP signaling pathway is positively correlated with EPS contents[32,35]. The elevated expressions of these pathways thus enhance EPS production around these nitrifiers, forming the self-produced extracellular matrix on the plastic surfaces[29]. Additionally, we also found that the expressions of genes regulating flagellin synthesis were missing in these sessile nitrifiers (Fig. 6c) and flagellar motor function was thus inhibited. The lack of flagellin allows the pioneer nitrifiers to adhere to plastic surfaces with extracellular matrix and gradually form stable biofilms[36], thereby enriching more nitrifiers from the surroundings. The higher abundances of nitrifiers in the plastisphere than those in the seawater (Supplementary Fig. 4)

reinforce the reasoning. Collectively, our results indicate that the increased expression of carbon fixation pathways and the activation of purine metabolism and cGMP/cAMP signaling in the sessile nitrifiers inhibited flagellin synthesis, promoted EPS synthesis, and ultimately facilitated dense biofilm formation. More planktonic nitrifiers from the surrounding seawater thus can be enriched in the plastisphere biofilms, fostering the nitrifying niche.

Microenvironments in the plastisphere can create transitional "oxic-microoxic-hypoxic" conditions, generating a steep $O_2$ gradient within biofilms[12,13], and thus strengthening the bacterial nitrification process. This is supported by the higher nitrification rates in the plastisphere, especially the nitrite oxidation rate (Fig. 1). The favorable redox conditions coupled with the high concentration of substrate $NO_2^-$ in the plastisphere selected the unique core nitrifiers residing in the biofilms compared to the seawater (Fig. 3). These active sessile nitrifiers in the plastisphere have been reported to possess versatile metabolisms and flexible adaptability to environments[37–41]. For instance, members of *Nitrobacter*, *Nitrosospira* and the COM nitrifier *Nitrospira_defluvii* observed in the plastisphere are capable of functioning under low oxygen levels[42] and even utilizing simple organic compounds (i.e., acetate and hexose sugars) as alternative energy sources as well[43], which enables them to increase the nitrification potential of the estuarine plastisphere. By contrast, the nitrifiers such as *Nitrotoga* members observed in the seawater commonly perform a more energy-demanding pathway to utilize $CO_2$, i.e., the Calvin–Benson–Bassham (CBB) cycle[37,39], thus burdening their nitrification system. In addition, we found an obvious accumulation of $NO_2^-$ in the plastisphere (Supplementary Figs. 6 and 9), which is generally toxic to microorganisms due to the formation of free nitrous acid[44]. Compared to the periplasmic $NO_2^-$ oxidoreductase (NXR) in *Nitrotoga* members, the plastisphere nitrifiers such as *Nitrobacter* contain a cytoplasmic NXR that can maintain activity at higher $NO_2^-$ levels[45]. This improves their survival and offers functional advantages in the plastisphere. Overall, the pronounced change of active core nitrifiers from $O_2$-sensitive and high nutrient-dependent nitrifiers in the seawater (*Nitrosomonas*-like AOB and *Nitrotoga*-like NOB) to nitrifiers with versatile metabolisms and high affinity for substrates in the plastisphere (*Nitrosospira*-like AOB, *Nitrobacter*-like NOB and COM nitrifier *Nitrospira_defluvii*) further explains the heightened nitrifying activity of the plastisphere.

Spatial structure of biofilms and arrangement of microbial cells in the plastisphere promoting the cooperations and interactions[46] among sessile nitrifiers also likely drive the formation of the distinctive nitrifying niche. Utilizing MAGs-centric metatranscriptomics, we observed a possible cooperation of substrate ($NO_2^-$) exchange among the COM nitrifier *Nitrospira_defluvii* (MAG262), AOB *Nitrosospira* (MAG397) and NOB *Nitrobacter* (MAG715) in the plastisphere (Fig. 6a). Elevated expressions of *nirK* ($NO_2^-$ reductase) and *norBD* (NO reductase) genes in *Nitrobacter* and *Nitrosospira*, respectively, indicate their capabilities of $NO_2^-$ and NO reduction. Although *Nitrospira_defluvii* possessed high expression levels of *amoAC* and *nirK* genes, the absence of *norBD* expressions (Supplementary Fig. 17) suggests that this COM nitrifier cannot produce $N_2O$ via nitrifier denitrification but could provide NO and $NO_2^-$. To date, biogeochemical evidence of NO exchange among nitrifiers remains scant; however, a form of "reciprocal feeding" has been observed, whereby certain AOB, NOB, and COM nitrifiers trade $NO_2^-$ to counteract substrate deficiency[15,43]. Thus, we hypothesized that these *Nitrospira*-like COM nitrifiers may provide $NO_2^-$ for neighboring AOB (*Nitrosospira*) to produce $N_2O$, and in return, can receive $NO_2^-$ from neighboring NOB (*Nitrobacter*) for NO production. The released NO may reshape the cell membrane of nitrifiers to generate symbiont-like aggregates[15,47], which could further enhance collaborative $N_2O$ production in the plastisphere. Such interspecific cooperation among the plastisphere nitrifiers, inferred metatranscriptomically, occurs outside cells to ensure substrate

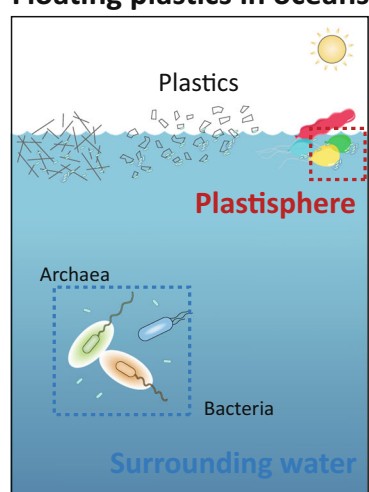

**Fig. 7 | Schematic diagram showing microbial nitrification process in the plastisphere (sessile mode) and surrounding seawater (planktonic mode).** The estuarine plastisphere has distinctive active nitrifying residents from the surrounding seawater, and could be an expanded "hotspot" of bacterial nitrification process and associated N$_2$O emission.

sharing[25], likely thereby increasing N$_2$O emission in the plastisphere. Confirmation through NanoSIMS and isotopic tracing of substrate transfer (NO$_2^-$, NO etc.) among nitrifiers is encouraged for future research.

In combination, here we propose a conceptual model of plastisphere potential in modulating global nitrogen biogeochemical cycling: the estuarine plastisphere serves as an expanded "hotspot" for bacterial nitrification and N$_2$O emission due to the unique nitrifier it hosts (Fig. 7). The plastisphere has an underappreciated role in biochemical processes by acting as a bridge between plastic debris and the surrounding seawater, facilitating their interactions and linking plastisphere dynamics to the microbial loop[48]. Its distinctive physical and chemical properties lead to niche differentiation among functional microorganisms with unique metabolic capabilities, setting them apart from those in the surrounding environment. Nevertheless, this study also acknowledges two limitations. Firstly, our SIP assays utilized supplementary high levels of NH$_4^+$ to enrich $^{13}$C-labeled DNA in nitrifiers, potentially biasing towards AOB and overlooking AOA. While our mechanistic study reveals AOB metabolic differences between sessile (plastisphere) and free-living (seawater) environments, future research should explore AOA dynamics and their metabolic adaptations for a comprehensive nitrifying blueprint in the plastisphere. Secondly, our exploration of nitrifying potential and metabolic mechanisms within the plastisphere predominantly conducted under laboratory conditions. While insightful, expanding these findings to estuarine ecosystems requires large-scale, in-situ monitoring. Such endeavors are essential for comprehensively grasping nitrogen cycling nuances, offering a holistic understanding of nitrification's ecological significance in estuarine plastisphere.

## Methods
### Sampling sites and experimental materials
The study areas are located in the estuaries and coasts, including (1) Xiamen (XM), Fujian province (118°11′E, 24°57′N); (2) Yantai (YT), Shandong province (121°47′E, 37°46′N); (3) Nanning (NN), Guangxi province (108°27′E, 22°84′N). XM and NN have a subtropical climate with 21 °C and 22 °C mean air temperature and 1000 mm and 1300 mm annual rainfall, respectively. YT has a temperate continental climate with 13 °C mean air temperature and 524 mm annual rainfall. These estuaries are influenced by anthropogenic activities (such as input of wastewater, nutrients, heavy metals or other pollutants).

We sampled the surface seawater of the three sampling sites from June to August of 2022 and 2023. After collection, the samples were kept in a 4 °C ice box and transported back to laboratory. Their water chemical characteristics were measured within 24 h. XM site (average values ($n = 3$)): pH 6.8, 28 °C, 6.98 mg L$^{-1}$ of dissolved oxygen, 72.9 mg L$^{-1}$ of total organic carbon, 1.06 mg L$^{-1}$ of NO$_3^-$, 0.26 mg L$^{-1}$ NH$_4^+$, 0.06 mg L$^{-1}$ of NO$_2^-$. YT site: pH 7.3, 24 °C, 7.23 mg L$^{-1}$ of dissolved oxygen, 129.7 mg L$^{-1}$ of total organic carbon, 1.37 mg L$^{-1}$ of NO$_3^-$, 0.39 mg L$^{-1}$ NH$_4^+$, 0.32 mg L$^{-1}$ of NO$_2^-$. NN site: pH 7.1, 27 °C, 7.06 mg L$^{-1}$ of dissolved oxygen, 78.4 mg L$^{-1}$ of total organic carbon, 0.79 mg L$^{-1}$ of NO$_3^-$, 0.17 mg L$^{-1}$ NH$_4^+$, 0.09 mg L$^{-1}$ of NO$_2^-$.

For biofilm type-based materials including plastic and other non-plastic ones (Fig. 1b), we used plastic bags (~20 cm × ~20 cm polyethylene, purchased from Cleanwrap Co., China), glass balls (3 mm, purchased from Jinggong Co., China), stone debris and wood debris (~1–5 cm collected from environment). These materials were cleaned with 70% ethanol and sterile water prior to experiments. For plastic type-based materials (Fig. 1b), we used three types of commercial plastics (polyethylene-PE, polystyrene-PS and polyvinylchloride-PVC) with densities ranging from 0.88 to 0.97 g cm$^{-3}$. These plastics are used for food bags (PE purchased from Cleanwrap Co., China) or cling films (PS from Chuanguan Co., China; PVC from Jusu Co., China). Prior to in-situ incubation, the plastics were immersed in 70% ethanol solution for 4 h to mitigate the impacts of surface microorganisms and additives in the plastics. The plastic samples (~20 cm × ~20 cm) were chosen to be representative of the common types that are present in surface water[11], rather than attempting to cover all types of commercial plastics.

### Incubation experiments
Four principal experiments were conducted in this study to investigate the nitrifying capability of the plastisphere and distinguish keystone nitrifiers, as illustrated in Supplementary Fig. 1.

Experiment 1: in-situ incubations for 28 days. The prepared biofilm type-based materials (plastic, glass, stone and wood) were each placed into 1 mm net bags, which were then connected by cotton cords[32] and placed in the three estuarine seawaters (XM, YT and NN) for 28 days. The plastic type-based materials (PE, PS and PVC) were connected using cotton cords as well, and then submerged at the XM site for 28 days. All the materials floated at 0.1–0.3 m under the water surface. We also placed 15-50 pieces of materials as back-up samples in nearby regions at 10-m intervals. After 28 days, the samples and the

surrounding surface seawater (50 L, 0.3 m below) were harvested. It should be noted that given the negligible changes in both nitrification activity and nitrifier community in seawater between the initial and end of the in-situ incubation (Supplementary Fig. 18), surface seawater was only sampled at the day 28. After taken to laboratory, the samples were split up into subsamples: a portion was used for water chemical analysis; another portion was used for the nitrification assays (Experiment 2 and 3) and rate measurements; the third subsample of the plastic type-based materials was used in the $^{13}$C-labeled incubation (DNA-SIP assays, Experiment 4).

Experiment 2: biofilm type-based lab-scale nitrification assays for 36 h. We established five groups in this experiment, including (i) plastic biofilm, (ii) glass biofilm, (iii) stone biofilm, (iv) wood biofilm and (v) surrounding seawater groups. Each group was in quadruplicate ($n = 4$). A total of 3 (three sampling sites) × 5 (four types of biofilms and seawater) × 4 (quadruplicate samples) = 60 experimental units were obtained in Experiment 2. The harvested materials from Experiment 1 were divided into 10 pieces and transferred into a 120-mL serum bottle. Meanwhile, the sterile seawater from the three sites was achieved by filtration with a 0.22-μm polycarbonate membrane and this water was then used for all biofilm groups to avoid the effects of seawater microorganisms. The bottles were tightly capped after adding 50 mL of sterile seawater and in-situ seawater. Selection of the 50 mL seawater here was due to the minimum water volume for sample submersion. 1 mL of $(NH_4)_2SO_4$ and 5 mL of $O_2$ were injected, respectively, reaching final concentrations of 50 μM $NH_4^+$ and 26% $O_2$ in the bottles. The higher $NH_4^+$ level was selected as the initial concentration mainly aiming at comparatively assessing the potential nitrifying capacity of the biofilms and surrounding seawater. These bottles were then incubated in the dark at 25 °C and 120 rpm for 36 h. During the incubation, $NH_4^+$, $NO_2^-$, $NO_3^-$ concentrations were measured with an Ion Chromatograph (Dionex, IC-3000, USA, detection limits: < 100 ng/L)[49]. $O_2$ and $N_2O$ were determined with a gas chromatography (Agilent 7890 A, USA) equipped with TCD and ECD detectors[50]. Detection limits of $O_2$ and $N_2O$ are ~3000 ppm and ~320 ppb, respectively. After the incubation, $N_2O$ emission and isotopocules, microbial biomass (16S rRNA gene abundance and cell number) and nitrifier abundances (bacterial and archaeal *amoA*) were measured. Other details are provided in Supplementary Fig. 1. To differentiate the contributions of bacteria and archaea to the nitrification process, we repeated the nitrification assays with adding 100 μM of penicillin. Penicillin can inhibit bacterial growth but not archaea[26,51]. Other procedures are as described above.

Experiment 3: plastic type-based lab-scale nitrification assays for 36 h. Following Experiment 2, we further explored the effects of different plastics on the nitrification process. This experiment included four groups, i.e., the surrounding surface seawater group and the three plastisphere groups (PE, PS and PVC). Each group was in quadruplicate ($n = 4$). A total of 1 (one sampling site, XM) × 4 (three types of plastisphere and seawater) × 4 (quadruplicate samples) = 16 experimental units were obtained in Experiment 3. In the plastisphere groups, each type of harvested plastics from Experiment 1 was cut into 10 pieces and transferred into a 120-mL serum bottle. The following procedures were consistent with those in Experiment 2.

Experiment 4: lab-scale flush-feeding incubations with $^{13}CO_2$ and $NaH^{13}CO_3$ for 30 days. As the plastisphere biofilm typically presented the higher nitrifying activity than other biofilms (Experiment 2) and little differences in nitrogen transformation and $N_2O$ emission existed among each plastisphere (Experiment 3), we established two groups in this experiment: (i) the plastisphere group mixing PE, PS and PVC, and (ii) the surrounding surface seawater group to explore active sessile and planktonic nitrifiers and their metabolic differences. $^{13}$C-labeled and $^{12}$C-labeled microcosms were established in each group and each microcosm was set up in triplicate ($n = 3$). Thus, a total of 2 (plastisphere or seawater group) × 3 (triplicate samples) × 2 ($^{13}$C-labeled or $^{12}$C-labeled samples) = 12 experimental units were obtained in

Experiment 4. Similar to Experiment 3, ten pieces of mixed plastic debris and 50 mL of sterile seawater were the plastisphere group; the bottle only with 50 mL of in-situ seawater was the surrounding seawater group. We added 1 mL of 1 mM $(NH_4)_2SO_4$ and 2 mM $NaH^{13}CO_3$ or $NaH^{12}CO_3$ (Sigma-Aldrich, USA), finally reaching concentrations of 50 μM $NH_4^+$ and 100 μM $HCO_3^{2-}$. These bottles were then sealed and 5 mL $O_2$ and $^{13}CO_2$ or $^{12}CO_2$ (Sigma-Aldrich, USA) were injected[52]. All the microcosms were incubated under the same conditions as Experiment 2/3 and were resupplied with $(NH_4)_2SO_4$, $NaHCO_3$, $O_2$ and $CO_2$ every 2 days. This is because we aim to culture and enrich $^{13}$C-labeled DNA in nitrifiers, and thus to distinguish metabolic differences of active nitrifiers between the plastisphere (sessile) and seawater (free-living). $NH_4^+$, $NO_2^-$ and $NO_3^-$ concentrations were measured before each resupplementation. To prevent the accumulation of $NO_3^-$, we replaced the incubation media with fresh sterile or in-situ seawater every 10 days. The incubation of Experiment 4 is detailed in Supplementary Fig. 10.

## $N_2O$ isotopocules and emission pathways

To distinguish the contribution of $N_2O$ emission via $NH_2OH$ oxidation and nitrifier denitrification, we measured the site preference (SP) of $N_2O$. $N_2O$ produced from the two pathways has a unique preferential cleavage of the $^{14}N$-$^{16}O$ and $^{15}N$-$^{16}O$ in the intermediates, generating different enrichments of $^{15}N^\alpha$ ($^{14}N$-$^{15}N$-$^{16}O$) and $^{15}N^\beta$ ($^{15}N$-$^{14}N$-$^{16}O$) and thus leading to the unique SP-$N_2O$ value[53,54]. After Experiment 2 and 3, 1 mL of headspace gas was taken and transferred to a 12-mL pre-vacuumed vial (Labco Exetainer, UK) to measure $N_2O$ isotopocules. The vial was then filled with high purity helium (He) gas. The detailed procedures are given in our previous study[49]. Briefly, a Precon+Gasbench coupled with an isotope ratio mass spectrometer (Delta V plus, USA) was used to detect $\delta^{15}N^{bulk}$-$N_2O$, $\delta^{15}N^\alpha$-$N_2O$ ($^{14}N$-$^{15}N$-$^{16}O$) and $\delta^{15}N^\beta$-$N_2O$ ($^{15}N$-$^{14}N$-$^{16}O$) abundances. The gas sample was enriched in a liquid $N_2$ trapper and then separated by a 30-m gas chromatography column. High-purity He gas was used to transport the samples to a mass spectrometer at 2 mL min$^{-1}$ speed. $N_2O$ isotopocules were identified by capturing ions $N_2O^+$ (m/z: 44, 45 and 46) and $NO^+$ (m/z: 30 and 31). The scrambling factor was 0.085. High-purity $N_2O$ (>99.99%) was used as the reference gas, and the $N_2O$ isotopocules values of the reference gas were analyzed at the Thünen Institute of Climate-Smart Agriculture (ICSA), Germany. Two $N_2O$ standard gases used in this study are kindly provided by Dr. Anette Goeske and Dr. Reinhard Well, and were applied to perform two-point calibrations for values of SP-$N_2O$. The $\delta^{15}N^{bulk}$-$N_2O$, $\delta^{15}N^\alpha$-$N_2O$, $\delta^{15}N^\beta$-$N_2O$ and SP-$N_2O$ values are estimated as follow[53]:

$$\delta^{15}N^i - N_2O = (^{15}N^i_{sample} - {}^{15}N_{standard})/{}^{15}N_{standard} \qquad (1)$$

$$\delta^{15}N^{bulk} - N_2O = (\delta^{15}N^\alpha + \delta^{15}N^\beta)/2 \qquad (2)$$

$$SP - N_2O = \delta^{15}N^\alpha - \delta^{15}N^\beta \qquad (3)$$

where $\delta^{15}N^i$ represents $\delta^{15}N^{bulk}$, $\delta^{15}N^\alpha$ or $\delta^{15}N^\beta$. $N_2O$ isotopic values are expressed as ‰ relative to atmospheric $^{15}N$-$N_2$. The typical detection precisions for $N_2O$-$\delta^{15}N^{bulk}$, $N_2O$-$\delta^{15}N^\alpha$ and $N_2O$-$\delta^{15}N^\beta$ are 0.9‰, 0.9‰ and 0.3‰, respectively.

As all samples were incubated under oxic conditions during Experiment 2 and 3, heterotrophic denitrification could not occur in this study. Thus, the fractions of $N_2O$ from $NH_2OH$ oxidation ($F_A$) and nitrifier denitrification ($F_N$) are possible to be distinguished as[14]:

$$F_N = \frac{SP - SP_A}{SP_N - SP_A} \qquad (4)$$

$$F_A = 1 - F_N \qquad (5)$$

where SP is the measured SP-N$_2$O value, SP$_A$ is the SP-N$_2$O value of NH$_2$OH oxidation (32.0 ~ 38.7‰, average value: 35‰)[54,55], and SP$_N$ is the SP-N$_2$O value of nitrifier denitrification (−13.6 ~ 1.9‰, average value: −5.9‰)[54] (Supplementary Table 1). Here we omitted N$_2$O contributions from AOA, AOB and COM due to overlapping SP-N$_2$O values between AOA and AOB, and scant isotopic data on COM-derived N$_2$O. Nevertheless, we applied a respiration inhibitor to discern the AOA and AOB contributions (See Method 2.2 Experiment 2). To assess errors of the above calculation, we further applied the Monte Carlo sampling method using MATLAB software[32,56] (Codes are provided in SI. The Monte Carlo simulation results with 10000 samplings are shown in Supplementary Fig. 19. Combining Eqs. (4) and (5), the relative contributions of NH$_2$OH oxidation (F$_A$) and nitrifier denitrification (F$_N$) to N$_2$O emission in the biofilms and surrounding seawater were obtained.

## Nitrification rate measurement

After Experiment 2 and 3, we further measured the nitrification rates of the biofilms and the surrounding seawater. The $^{15}$N-amended substrates (($^{15}$NH$_4$)$_2$SO$_4$ (99% $^{15}$N atom, Aladdin, China) or Na$^{15}$NO$_2$ (98% $^{15}$N atom, Aladdin, China)) and 1 mM NaHCO$_3$ were added into each 12-mL vial, including 5 pieces of materials and 50 mL sterile seawater (biofilm group) and 50 mL in-situ seawater only (seawater group), to determine the ammonia oxidation and nitrite oxidation rates[57,58]. These bottles were then aerobically incubated at 25°C for 8 h. Ammonia oxidation rate was quantified as the $^{15}$NO$_2^-$ production from incubations with ($^{15}$NH$_4$)$_2$SO$_4$ amendment. Briefly, 200 μL of 0.5 mM $^{15}$NH$_4^+$ solution was added to each bottle, which was then terminated at 4 h and 8 h, respectively. To measure the concentration of $^{15}$NO$_2^-$, 100 μL of 16.5 mM sulfamic acid (H$_3$NO$_3$S) was added aiming to reduce $^{15}$NO$_2^-$ to $^{29}$N$_2$. The reaction time lasted over 12 h to ensure conversion completely. Nitrite oxidation rate was quantified as the $^{15}$NO$_3^-$ production from incubations with Na$^{15}$NO$_2$ amendment. Briefly, 100 μL of 0.2 mM Na$^{15}$NO$_2$ solution was added to each bottle, and 100 μL of 1 mM ZnCl$_2$ was injected to terminate the reaction at 4 h and 8 h, respectively. To measure $^{15}$NO$_3^-$, sulfamic acid was added to remove initial NO$_2^-$ content prior to detection. Then, 1 g of sponge cadmium was added (adjusting pH: 7–8) aiming to reduce the $^{15}$NO$_3^-$ produced to $^{15}$NO$_2^-$. The following steps were consistent with those above in the measurement of ammonia oxidation rate. The concentration of N$_2$ was quantified by an isotope ratio mass spectrometry (IRMS, Delta V Advantage, Germany) with a detection limit of 0.1 μM. 16 S rRNA gene abundance of each bottle was quantified by qPCR, and microbial cell numbers were calculated by 16S rRNA abundance/4.1 on the basis of Ribosomal RNA Operon Copy Number Database[59]. Nitrification rate is expressed as fmol N h$^{-1}$ cell$^{-1}$ to assess the nitrifying activity.

## DNA fractionation

After Experiment 4, ten pieces of plastic debris were collected for DNA extraction (FastDNA Kit for Soil, MP, USA). The surrounding seawater group was filtered through 0.22-μm filters and then was collected for DNA extraction with the same Kit. The DNA quantity was assessed with a Nanodrop Spectrophotometer (Thermo, USA). To fractionate the DNA, 3.0 μg of DNA was mixed with 1.6 mL GB buffer and 6.4 mL CsCl stock solution (Sigma-Aldrich, USA) to obtain an initial buoyant density of 1.710 g mL$^{-1}$. GB buffer (pH 8.0) contains 0.1 M of Tris-HCl, 0.1 M of KCl and 1.0 mM of EDTA. Next, the DNA mixture was transferred to a 7.5-mL Beckman ultracentrifuge, and ultracentrifugation was then performed with a VTi-65.2 vertical rotor (Beckman, UAS) at 36,000 rpm for 48 h at 20 °C. The gradient mixture was fractionated using an automatic-sampler (BSZ-100, China), and a total of thirteen DNA fractions (~438 μL each) were harvested for each sample. The buoyant density was measured using an AR200 digital refractometer (Reichert, USA). The fractionated DNA was precipitated using PEG 6000 for 2.5 h and then centrifuged for 30 min at 13,000 × *g*. The

pelleted DNA was washed with ethanol (70%) and then stored in 50 μL TE buffer. The fractionated DNA was subsequently used to quantify *amoA* gene abundances to locate the $^{13}$C-DNA fraction, which was then used for amplicon and metagenome sequencing.

## Amplicon sequencing and gene abundance

In this study, the procedures for DNA extraction from biofilms are detailed in Supplementary Information. The 16S rRNA (bacterial and archaeal communities), bacterial-*amoA* (AOB), archaeal-*amoA* (AOA) and Comammox-*amoA* genes (COM) were selected as marker genes for amplicon sequencing[26,60,61]. The primer sets and amplification conditions are listed in Supplementary Table 2. The PCR system of 16S rRNA included 21 μL of sterile water, 25 μL of SYBR Premix (TaKaRa, Japan), 1 μL of the forward/ reverse primers and 2 μL of DNA. The PCR system of bacterial-*amoA* (B-*amoA*) and archaeal-*amoA* (A-*amoA*) included 6.4 μL of sterile water, 10 μL of SYBR Premix (TaKaRa, Japan), 1.6 μL of the forward/reverse primers and 2 μL of DNA. The PCR system of COM-*amoA* included 6.4 μL of sterile water, 10 μL of SYBR Premix (TaKaRa, Japan), 1.6 μL of the forward/reverse primers and 2 μL of DNA. All PCR products were then purified and recovered before library construction and sequencing. Purified libraries containing 16S rRNA, AOB-*amoA*, and COM-*amoA* genes were sequenced on the Illumina MiSeq PE300 platform (Illumina, San Diego, CA), while the AOA-*amoA* gene was sequenced on the PacBio Sequel IIe System (Pacific Biosciences, CA, USA). Data from the MiSeq PE300 system were processed by demultiplexing and quality filtering the obtained sequences using Fastp (version 0.20.0)[62], followed by merging with FLASH (version 1.2.11)[63]. The high-quality sequences were then denoised using the DADA2 pipeline[64] in QIIME 2 (version 2020.2)[65] with default parameters. Data from the PacBio system were processed by first obtaining high-fidelity reads from raw sub-reads generated via circular consensus sequencing (CCS) by Single Molecule Real-Time (SMRT, version 11.0)[66]. These high-fidelity reads were then length-filtered and denoised as described above. Taxonomy was compared with GenBank and UNITE databases. Phylogenetic analysis of nitrifiers was conducted using AOA-*amoA*, AOB-*amoA*, COM-*amoA*, and NOB-16S rRNA sequencing data. The 5, 14, 11 and 7 typical ASVs contributing to 30.4-42.9%, 60.3-77.3%, 73% and 3.0-3.8% of the total AOA, AOB, COM and 16S rRNA sequences, respectively, were selected to construct the trees. Specially for NOB, we designated ASVs from 16S rRNA sequences affiliated with *Nitrospira*, *Nitrobacter* and *Nitrotoga* as candidate NOB ASVs for phylogenetic reconstruction. Homologous sequences from NCBI were used for constructing maximum likelihood phylogenetic trees with the Kimura 2-parameter model and 1000 bootstraps in MEGA7.

The absolute abundances of 16S rRNA, bacterial-*amoA* and archaeal-*amoA* genes were quantified with qPCR technique. The primer sets are 515F/907R, amoAF/amoAR and bamoA1F/bamoA2R, which are the same primer sets used for the sequencing. The reaction system was conducted in a 20-μL mixture: 6.9 μL of sterile water, 10 μL of SYBR Premix (TaKaRa, Japan), 1.6 μL of the forward/reverse primers and 1.5 μL of DNA. The qPCR conditions of 16 S rRNA were 95 °C for 3 min, 39 cycles of 95 °C for 30 s, 55 °C for 30 s and 72 °C for 30 s. The qPCR conditions of bacterial-and archaeal-*amoA* were 95 °C for 3 min, 39 cycles of 95 °C for 30 s, 55 °C for 30 s and 72 °C for 45 s. All the amplification efficiencies were 95.6–100%, with R$^2$ ranging from 0.990 to 1.000.

## Metagenomics and metatranscriptomics

For the three replicates, the $^{13}$C-DNA (fractions 9 and 10) in both the plastisphere and surrounding seawater groups form Experiment 4 were used for metagenomic sequencing with a VAHTS Universal Plus DNA Library Prep Kit for Illumina (Vazyme Biotech, China). The pooled $^{13}$C-DNA (consisting of fractions 9 and 10) was then concentrated (α-1-2 LDplus, Germany), and quantified with a Qubit dsDNA Assay Kit (Life

Technologies, USA)[52,67]. The concentrated DNA (15 ng for each sample) was used for library preparation and sheared into 350 bp fragments which were subsequently subjected to PCR assays to verify the fragment length. The products were purified, amplified, and sequenced with the NextSeq550 platform (Illumina, USA), finally generating 2×150 bp paired-end reads, which were then processed using Fastp (version 0.20.0) to eliminate low-quality sequences and reads containing ambiguous N bases. Total RNA from the plastisphere and surrounding seawater groups was extracted with an RNA-prep Pure Kit (Tiangen, China). After removing gDNA with a TURBO DNA-free Kit (Ambion, USA), we obtained ~40 and ~31 ng μL$^{-1}$ RNA in each plastisphere and seawater group, respectively. The RNA integrity number ranges from 8.0 to 9.1. We used 16S rRNA-based PCR assays to confirm DNA was removed in RNA samples. Prior to metatranscriptomic library construction using a TruSeq RNA-Prep Kit (Illumina, USA), the extracted RNA was first pooled and fragmented into 250–300 bp (Covaris, USA). The raw reads were processed with Fastp to trim bases with a quality score (<30) and to remove sequences containing adapters and contaminants. The quality-controlled reads were then co-assembled using Megahit (version 1.2.9) with iterative k-mer sizes of 31, 41, 51, 61, 71, 81, and 91. Each metagenome size of plastisphere and seawater samples was 22.17 ± 3.36 Gb.

To identify the active nitrifiers and obtain the complete genomes of these nitrifiers, genome assembly and binning were conducted with metaWRAP (version 1.2.1) pipeline[68]. The clipped reads were assembled using Megahit (version 1.2.9) to obtain clean contigs, with k-mer sizes ranging from 47 to 97 in steps of 10. The binning with contigs over 1000 bp was then carried out by the CONCOCT (version 0.4.0)[69], MaxBin2 (version 2.2.2)[70] and MetaBAT (version 2.12.1)[71]. The generated bins were transferred into a complete bin set within the module of Bin_refinement, and were amended with the module of Reassemble_bins to obtain the Metagenome-Assembled Genomes (MAGs). The quality of the obtained MAGs was examined with CheckM (version 1.0.5)[72]; both MAGs with completeness ≤ 75% and contamination ≥ 10% were discarded, and the remaining was used for pairwise dereplication comparison with dRep (version 1.4.3). A threshold of 98% of average nucleotide identity (ANI) was selected as a cutoff for dereplication[73]. Taxonomy affiliation of MAGs was determined by GTDB-Tk (version 0.3.2)[74]. In this study, we mainly focused on the MAGs (nitrifiers) associated with the nitrification process. The nitrifiers containing *amoABC* and *nxrAB* were identified as COM nitrifiers. The mRNA reads were linked to the nitrifying bins and counted in KALLISTO (version 0.46)[75]. To correct the relative expressions in all MAGs, the counts of transcript were normalized to 1 million per each MAG (transcripts per million TPM)[76]. Open reading frames (ORFs) were predicted using Prodigal (version 2.6.3)[77]. KEGG pathways of each nitrifying MAG were predicted with BlastKOALA[67]. Relative expression of each gene involved in nitrification process and metabolisms of interest was calculated on the basis of the total TPM[78]. Phylogenetic analysis of all medium- and high-quality MAGs was conducted with FastTree (version 2.1.10)[79] based on 120 bacterial and 122 archaeal marker genes to evaluate the phylogenetic placement and relative evolutionary divergence (RED) of genomes within the GTDB reference tree. Phylogenomic trees were inferred with WAG and GAMMA models and 1000 bootstraps, based on alignments of these marker genes, and visualized using iTOL(v4). The abundances of each nitrifying MAG in both metagenomic and metatranscriptomic datasets were quantified using the module of "Quant bins" in metaWRAP (version 1.2.1). The module applied Salmon to align reads in each plastisphere and seawater samples to the assembled contigs and also to produce the corresponding coverage values, which were then standardized by library size and by contig length, similar to transcripts per million (TPM) in RNAseq analysis. The library size was for every 1,000,000 metagenomic reads. The quality checked reads from metatranscriptomic were mapped against the bowtie2 index of contig that is constructed from a chained contigs file aiming to quantify the expression levels of the loci's contigs. Specifically, we primarily focused on the expression levels of the nitrifying MAGs. The obtained files were first converted to BAM files and then the CoverM software (version 0.3.1) was applied to remove low alignments (<75% identity, < 75% alignment coverage)[67].

## Statistical analysis

Prior to analysis, we tested for the homogeneity of variances (Levene's test) and the normality of residuals. One-way analysis of variance (One-way ANOVA) combined with the Tukey post hoc test was then performed for the significance test (such as for nitrification rate, N$_2$O emission and isotopes) between each type of biofilms and the surrounding seawater (SPSS version 22.0). $P$ values < 0.05 indicate a significant difference.

## Reporting summary

Further information on research design is available in the Nature Portfolio Reporting Summary linked to this article.

## Data availability

Sequencing data generated in this study have been deposited in the NCBI database under accession number SUB12931929 for amplicon sequencing data, SUB12931935 for metagenome data, and SUB12931940 for metatranscriptome data. All other data of this study are available in Supplementary information, supplementary data, GitHub (https://github.com/xuangood/estuarine-plastisphere) or figshare: MAGs (https://doi.org/10.6084/m9.figshare.26085544), representative sequences (https://doi.org/10.6084/m9.figshare.26087422), figures with raw data (https://doi.org/10.6084/m9.figshare.26087419).

## Code availability

Custom scripts and codes in this study can be searched on the GitHub (https://github.com/xuangood/estuarine-plastisphere) and figshare (https://doi.org/10.6084/m9.figshare.26087410). Figures are created by Origin 9.0 and Adobe Illustrator CS6.

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

## Acknowledgements

This project was supported by the Science Fund for Creative Research Groups of the National Natural Science Foundation of China (42021005, Y.-G.Z.), National Key Research and Development Program of China (2022YFD1901402, X.S.), and Ningbo S&T project (2021-DST-004). We also thank the Climate Change and its Environmental Implications (CCEI) Thrust from Southwest University (SWU-XDJH202320).

## Author contributions

S.X.X., H.X.R., Y.L.Y, Z.Y.Y. Z.G.B., and Z.Y-G. conceived the study and conducted the incubations, analyzed the data and wrote the manuscript. S.X.X. Z.Y.Y. H.X.R. and T.Y.J. analyzed sequencing data. S.X.X., H.X.R. and Y.L.Y. measured N speciation. Y.X.R., W.T., P.J.L., and Z.J.B. detected isotopes and analyzed the data. D.J. and L.R.L. helped to conduct in-situ incubations in YT and NN sampling sites. S.X.X., L.Z.L., and H.F.Y. conducted DNA-SIP analysis. Y.X.R., T.Y.J, Z.Y-G., C.X.P. and M.R. edited the manuscript.

## Competing interests

The authors declare no competing interests.
