## [Peer Review File · Nature Communications]

Nitrifying niche in estuaries is expanded by the plastisphereReviewers' comments:

Reviewer #1 (Remarks to the Author):

Key results

These authors found that nitrification is enhanced by plastisphere species, compared to surrounding seawater. The nitrification approach also varies between these compartments, with commamox being more prevalent in the active nitrifying plastisphere community. Using metagenomics, the authors detail key nitrifying genera within each compartment; while overall similar in gene presence absence, this highlights the different pathways for nitrification between microbes in seawater and plastisphere, as illustrated schematically. This has implications for not only nitrification, but also N₂O emission, which was found to be higher in the plastisphere.

Validity

The authors' approach is very thorough, from a molecular science standpoint. Yet, a large portion of their results hinge around the active nitrifier community. While this is important and appreciated, I'd like some clarification on the separation between active/inactive (i.e., heavy and light DNA), as described below. There seems to be a difference in this separation between plastisphere and seawater samples, which may bias results. Moreover, as mentioned below, seawater was only collected at the end of the study; could these differences simply be an observed effect of surrounding seawater differences from time of early plastic colonization (i.e., starting seawater community) and final seawater community? It's unlikely, but an important consideration that the authors should address.

Significance

The general question – do AOA, AOB, NOB and COM populations differ between plastic biofilms and surrounding seawater, and what are their contributions to marine nitrification? – is interesting and a meaningful contribution to the field. Their results are a bit limited in scope, however. Plastic is just one material that can create surface habitat for microbes in seawater. Do we see the same nitrification increase from floating wood or glass debris, for example? This is a necessary comparison to make in order to make the conclusion that plastics, and not just surface substrate in general, is important for nitrification in seawater. This concern is underscored by the authors' microbial abundance data, illustrating the general importance of substrate in concentrating certain bacteria and archaea (likely, generally those with surface attachment propensity). This is also limited in location (sampling in one estuary) and plastics (using three different polymer films). I would favor more sample types over the number of analyses conducted here, but their approach is robust from a nitrification/genetic standpoint and very impressive!

Data and methodology

The comparison of plastic biofilm is important, but I am disappointed to see that the authors only collected seawater at one time point. For example, it would have been enlightening to collect seawater at the start and end of the experiment. They discuss the importance of microbial succession in their introduction, which can be based on changes in the surrounding seawater microbial community. If they authors have good reason to believe the seawater community did not

change significantly over the 28 day experiment, please include that rationale and any citation(s). Are there published papers that include some of the methods described here (e.g., for experiment 2, 13C work, etc.)? If so, please add these. Some details of the method – e.g., gas chromatograph settings, limits of detection, etc. – are not included here.

For statistical analysis, did the author's also confirm homogeneity of variance (e.g., Levene's test)? This is a requirement for ANOVA, and should be confirmed. I suspect it will pass, but if it doesn't pass this test, a non-parametric version of ANOVA should be used.

In the separation of active nitrifiers with 13C, there seems to be greater overlap in active vs inactive community in the plastisphere (layers 9, 10; Figure 3a) than in the surrounding seawater. This could account for differences in the abundances observed. Please see point about clarifying this figure to determine if this is a valid concern.

Analytical approach

Generally, very thorough. See previous comments about concerns. Otherwise, the authors perform a variety of analyses to bolster their conclusions.

Suggested improvements

Title: I think the title is missing some words for clarity. I suggest "Nitrifying niche in the ocean is expanded by the plastisphere". Also, 'nitrifying niche' is a bit confusing. Could 'nitrification' be a more direct wording?

Abstract:

Line 27: Please rephrase to "The oceanic plastisphere...". Can delete 'here' in line 32.

Methods:

Please provide further details on the plastics. Where were they sourced from? Did you perform any analysis on additives in these plastics? If not, do you suspect any based on their applications (e.g., where you purchased from or the product type). Additive content may be a bigger driver of biofilm formation than polymer chemistry itself.

Line 154 – don't spell out milliliter

Are the experiment numbers in figure S2 consistent with the text? I think experiment 2 and 3 are flipped.

Can the reader better define site preference in the text? Functionally, what does this mean? This is difficult to interpret, despite the figure (2b) and method being clear.

Figures and captions:

I suggest not using W and P for seawater and plastic. This is confusing in the caption, and not necessary for reading the figures. These words are fairly short, so please spell them out.

In figure 3a, can you add vertical lines to show collection of layers 9 & 10 (or, transparent shaded boxes over the density of regions 9 and 10)? This would help address potential bias of this approach mentioned previously. Figure 3c is also very blurry; please use high resolution copy for figure upload, if you have not already.

Figure 5: in the third part (FPKM) I assume each box correlates to the gene to the left, but this isn't explicitly detailed in the caption. Please clarify. Also, is there a test to validate statistical differences

between plastisphere and seawater here? Although in different genus, the genes look very similar overall (including all MAGs) in gene presence/absence and abundance between seawater and plastisphere.

Figure 6: it is difficult to distinguish between the different green colors, as well as the reds/oranges. Please consider different color choices.

References:

2 – first author last name is misspelled

Reviewer #2 (Remarks to the Author):

This manuscript describes research into the colonization, growth and activity of nitrifiers in the plastisphere – i.e. determining whether the colonization of plastic which is now abundant in the ocean represents a new nitrifying niche. The study involves a impressive amount of work using stable isotopes, metagenomics/transcriptomics and biogeochemical data to demonstrate (successfully) which populations are likely active in these samples. However, I have reservations about the choice of environment and interpretation of some of the experimental data, and whether the study demonstrates that the plastisphere is really a unique habitat for nitrification, or whether it simply shows that plastic provides a surface for microbial colonization like any other material in the ocean (e.g. glass, metal, rope, etc.).

A central assumption of the manuscript is that the plastisphere is a unique and previously unrecognized habitat for nitrifiers and nitrification in the ocean. For this to be something novel compared to communities and nitrifiers growing on surfaces and biofilms, I think there would have to be a demonstration that there is something unique to plastics which support their growth e.g. comparing growth/activity of nitrifiers on other control surfaces (e.g. glass). There is a lot of research showing that nitrifiers like to grow on surfaces (e.g. wastewater, laboratory reactors of both AOA AOB, etc.) and are hotspots for activity compared to planktonic cells, and I am uncertain that this really demonstrates something surprising.

The title of the paper and discussion would indicate that this study aims to look at the interaction of nitrifiers and plastics in the (open?) ocean. This study appears to have used an estuarine environment with a significant terrestrial influence. For example, the pH of the water is 6.8 (the open ocean is ~8.1 I think) and Nitrosocosmicus, Nitrososphaera and Nitrotoga are not typical marine nitrifiers found or active in the open ocean. This is fine, but I think interpretation to the ocean should be reconsidered.

DNA-SIP. The data presented and interpretation is puzzling. For both control water and plastic samples, a small increase in the buoyant of the most abundant fraction/the peak in the ^{13}C samples is interpreted as indicating growth of nitrifiers on ^{13}C . However, as this is profiling total 16S rRNA genes, and not nitrifier specific amplicons (e.g. amoA), would you expect to see shifts in the total (not partial) community from autotrophic ^{13}C incorporation alone, particularly when these populations represent less than 0.01% of the total community (based on numbers in Figure S7)?

This would require a lot of cross-feeding. Secondly, the shift in buoyant density from a fully 12C to a fully 13C genome is ~0.04 g ml⁻¹ (e.g. see profiles of Jia and Conrad, 2009; Zhang et al., 2012 ISME), and in this data, the shifts are in the order of 0.005 g ml⁻¹, and where the whole community is shifting slightly, and not just a small proportion of the (autotrophic) community growing. I am not sure what could explain these very slight shifts, but I assume it is unlikely that in a total community, all populations have received a minor and equal 13C enrichment. But in summary, interpreting these as indications of activity should be treated with caution. As you are preparing metagenomes and metatranscriptomes from recently colonized material, these will likely represent active communities at some level, but the use of DNA SIP for targeting these is not convincing.

Are the (high?) ammonium concentrations used in incubation experiments realistic for looking at realistic in situ activities of nitrifiers in the ocean? For the SIP incubations, the samples receive a lot of ammonium every two days, perhaps making it more like a laboratory enrichment than a realistic analogue of the open ocean. I think you could maybe need explain the choice of these concentrations.

Specific comments

71 – ‘Nitrospira-like’. They are not ‘like’, but unambiguously Nitrospira.

72 – A few issues with this statement. It reads as if NOB are fixing N₂. Also, this is a 2008 reference which predates, for example, discovery of comammox and likely good predictions of AOA activity. Do you mean that 90% of fixed nitrogen is transformed from nitrite to nitrate by NOB?

78 – I believe this is a purely AOB-centric view of N₂O production. N₂O production in AOA comes from the abiotic reactions of intermediates and generally are not considered to perform nitrifier denitrification. Maybe expand this sentence a little to differentiate the pathways of AOA and AOB (and comammox?)

132 – ‘To prevent any loss’? I am uncertain what this means. To you mean back-up samples?

151 – Add ‘filtration’ into this sentence.

273 – Do you need to repeat the cycling conditions here? They are also in the cited table.

279 – Conditions

380 – ‘The plastisphere harbored more microbial biomass, especially the nitrifiers, than the seawater (P<0.001, Figure S7), which explains the higher nitrification rates of the plastisphere (Figure 1).’ I am uncertain why this is ‘especially the nitrifiers’ as they seem to constitute a very low proportion of the community i.e. for the four plastics, total 16S rRNA gene abundance and cell numbers were greater than 10^{e8}, but AOA and AOB amoA gene abundances were around 10^{e2} and 10^{e4} respectively. While I appreciate that there are differences between cell vs gene abundances, or copy numbers per genome, nevertheless, this would indicate that they ~0.01 and 0.0001% of the total communities?

Figure 6 – This figure indicates NOB as a source of N₂O production. Can you confirm there is experimental evidence from cultures that NOB produce N₂O?

Table S2. Add references for primers. Archaea is spelt incorrectly. The Svedburg unit of 16S is with an uppercase S.

Figure S1 – spaces missing between numbers and units

Replies and Explanations to Comments

We thank the reviewers for their constructive comments. We have conducted a series of additional experiments and expanded our data analyses as suggested by the reviewers. We have now revised the manuscript thoroughly and highlighted the changes in track-change mode. All questions raised by the reviewers have been answered point-by-point below.

Reviewer #1 Comments:

These authors found that nitrification is enhanced by plastisphere species, compared to surrounding seawater. The nitrification approach also varies between these compartments, with Comammox being more prevalent in the active nitrifying plastisphere community. Using metagenomics, the authors detail key nitrifying genera within each compartment; while overall similar in gene presence absence, this highlights the different pathways for nitrification between microbes in seawater and plastisphere, as illustrated schematically. This has implications for not only nitrification, but also N₂O emission, which was found to be higher in the plastisphere.

1. Validity: The authors' approach is very thorough, from a molecular science standpoint. Yet, a large portion of their results hinge around the active nitrifier community. While this is important and appreciated, I'd like some clarification on the separation between active/inactive (i.e., heavy and light DNA), as described below. There seems to be a difference in this separation between plastisphere and seawater samples, which may bias results. Moreover, as mentioned below, seawater was only collected at the end of the study; could these differences simply be an observed effect of surrounding seawater differences from time of early plastic

colonization (i.e., starting seawater community) and final seawater community? It's unlikely, but an important consideration that the authors should address.

Answer:

Thanks for the comments.

As a part of our revision, we have supplemented experiments to re-analyze the DNA-SIP results and have clearly distinguished between the heavy DNA (^{13}C -DNA) and light DNA (^{12}C -DNA) in the plastisphere and surrounding seawater samples. Please see the Response to Comments 5 and 21 for details.

The reasons why we collected the seawater samples only at the end of the *in-situ* incubation are detailed in the Response to Comments 3, and we have also added new experiments to support this.

2. Significance: The general question – do AOA, AOB, NOB and COM populations differ between plastic biofilms and surrounding seawater, and what are their contributions to marine nitrification? – is interesting and a meaningful contribution to the field. Their results are a bit limited in scope, however. Plastic is just one material that can create surface habitat for microbes in seawater. Do we see the same nitrification increase from floating wood or glass debris, for example? This is a necessary comparison to the conclusion that plastics, and not just surface substrate in general, is important for nitrification in seawater. This concern is underscored by the authors' microbial abundance data, illustrating the general importance of substrate in concentrating certain bacteria and archaea (likely, generally those with surface attachment propensity). This is also limited in location (sampling in one estuary) and plastics (using three different polymer films). I would favor more sample types over the number of analyses conducted here, but their approach is robust from a nitrification/genetic standpoint and very impressive!

Answer:

Thanks for this constructive comment, which we address in detail below.

As a part of our revision, we have added another two sampling sites spanning 1870 km in China, and we selected four types of materials, including plastic surface (plastic bags (polyethylene)) and other, non-plastic surfaces (glass balls, stone debris, and wood debris) as suggested to conduct a series of additional experiments. Please see the figure below.

Response to comments Fig 1 (now added in the main text, **Fig. 1**). Sampling and incubation sites, biofilm type- and plastic type-based materials. (a) Sampling and *in-situ* incubation sites, including site 1: Xiamen (XM) in Fujian Province (118°11'E, 24°57'N), site 2: Yantai (YT) in Shandong province (121°47'E, 37°46'N) and site 3: Nanning (NN) in Guangxi Province (108°27'E, 22°84'N). (b) Biofilm type materials include plastic bag, glass ball, stone debris, and wood debris; plastic type materials include PE, PS and PVC.

Consistent with our previous experiments, these materials were placed in the three estuaries for 28 days, after which they and the surrounding seawater were collected for lab-scale incubations. During the incubations, we measured nitrogen speciation dynamics, nitrification rate, N₂O emission as well as microbial abundances, cell numbers, and bacterial and archaeal *amoA* gene abundances. Our results indicate that the plastisphere exhibited elevated nitrifying activity compared to the glass and wood biofilms and the surrounding seawater. Although stone

biofilms harbored more nitrifying biomass, their nitrifying activities were comparable or even lower than those in the plastisphere, which further highlights the higher nitrification potential of the plastisphere in estuarine ecosystems. Meanwhile, we also found that plastic types exerted a minimal impact on microbial nitrifying activity. The detail results are shown below.

(1) Biofilm formation

After 28 days, microbial aggregates densely adhered to stone surfaces, more so than to plastics. The biofilms were less abundant on the glass surfaces, and were loose and easily dispersed on the wood surfaces.

Response to comments Fig 2 (now added in the Supporting Information, **Fig. S5**). Photo images of biofilms on the surfaces of plastic bags, glass balls, stone debris, and wood before and after the *in-situ* incubations.

(2) Microbial abundances and *amoA* abundances.

The stone and wood biofilms harbored more microbial biomass and nitrifiers (AOA and AOB) than the plastisphere and seawater. Nitrifier biomass in the glass biofilms was the lowest. Except at NN site, AOB abundances consistently surpassed

those of AOA at XM and YT sites, regardless of biofilms and seawater.

Response to comments Fig 3 (now added in the Supporting Information, **Fig. S6**). Microbial abundances and cell numbers after the 36-h incubation in Experiment 2 (biofilm type-based) and Experiment 3 (plastic type-based). **(a)-(d)** 16S rRNA-based microbial abundances (n=4) at XM, YT and NN sites; **(e)-(h)** cell numbers (n=4) at XM, YT and NN sites. Microbial cell numbers were calculated by 16S rRNA abundance / 4.1 on the basis of Ribosomal RNA Operon Copy Number Database. Different letters indicate the significant differences (ANOVA, $P < 0.05$).

Response to comments Fig 4 (now added in the Supporting Information, **Fig. S7**). Abundances

of nitrifiers including AOA and AOB after the *in-situ* incubation. **(a)-(d)** AOA abundances ($n=4$) at XM, YT and NN sites; **(e)-(h)** AOB abundances ($n=4$) at XM, YT and NN sites. Different letters indicate the significant differences (ANOVA, $P < 0.05$).

(3) Nitrogen transformations during lab-scale incubations

We supplemented a 36-h incubation to explore nitrification potential of different biofilms and estuarine seawater at the three sites. During the incubation, the headspace O_2 concentrations decreased rapidly from 26% to 3%, but oxic conditions were still maintained throughout the experiment.

Response to comments Fig 5 (now updated in the Supporting Information, **Fig. S8**). Changes of O_2 concentration during the 36-h incubation in Experiment 2 (biofilm type-based) and Experiment 3 (plastic type-based) ($n=4$).

Among the three sampling sites, variations of NH_4^+ and NO_3^- concentrations in the plastisphere and stone biofilms were more pronounced than in the glass and wood biofilms as well as in the seawater. Notable NO_2^- accumulation was observed in the plastisphere and stone biofilms, except at the YT site.

Response to comments Fig 6 (now added in the Supporting Information, **Fig. S9**). Changes of NH_4^+ , NO_2^- and NO_3^- concentrations during the 36-h incubation (Experiment 2) in the four types of biofilms and surrounding seawater. (a), (c) and (e) are the variations of NH_4^+ and NO_2^- concentrations at XM, YT and NN sites, respectively. (b), (d) and (f) are the variations of NO_3^- at XM, YT and NN sites, respectively.

(4) Nitrification rate

We found that the average rates of ammonia and nitrite oxidation in the plastisphere were significantly higher than in the glass, wood biofilms and the

surrounding seawater ($P < 0.01 \sim 0.28$, Figure 1c-e). The rates of glass and wood biofilms were lower than those of the surroundings at all sites ($P = 0.002 \sim 0.072$). More importantly, we found that despite stone biofilms harboring more nitrifying biomass, their nitrification rates were comparable ($P > 0.05$, Figure 1d-e) or even lower ($P = 0.004 \sim 0.039$, Figure 1c) than those in the plastisphere.

Response to comments Fig 7 (now added in the main text, **Fig. 1**). Nitrification rates. **(c)-(f)** Ammonia oxidation rates ($n=4$) and nitrite oxidation rates ($n=4$) in different biofilms, plastisphere types and surrounding seawater across the three sampling sites. Different letters indicate the significant differences (ANOVA, $P < 0.001 \sim 0.045$).

(5) N_2O emission

We compared nitrification-derived N_2O emission from each type of biofilm, plastisphere and the surrounding seawater. At the end of incubation, nitrification-derived N_2O emission from the plastisphere was significantly higher than that from stone biofilm at the XM site ($P < 0.001$), and also higher than the emissions from other biofilms and the surrounding seawater at all sites ($P < 0.001$). We further used N_2O isotopocules coupled with N_2O -SP value to discern the relative contributions of each N_2O emission pathway during nitrification, including NH_2OH oxidation and nitrifier denitrification. By calculating the N_2O -SP ($\delta^{15}N^\alpha - \delta^{15}N^\beta$), we found that the average values in the biofilms (4.58~9.46‰) were remarkably lower than those in the seawater (19.81~26.68‰, $P < 0.001$), implying distinct N_2O emission dynamics

in the biofilms. The primary N_2O emission in the biofilms was from nitrifier denitrification, contributing 53~70% of the total emissions; however, NH_2OH oxidation dominated in the seawater (62~79%).

Response to comments Fig 8 (now added in the main text, **Fig. 2**). Nitrification-derived N_2O emission and its source tracing. **(a)-(c)** N_2O emission of different biofilms after the 36-h incubation ($n=4$). **(d)** N_2O emission of different plastisphere after the 36-h incubation ($n=4$). **(e)-(f)** Relative contributions of AOA and AOB-derived NH_2OH oxidation and nitrifier denitrification to N_2O emission during nitrification in different biofilms and plastisphere.

Response to comments Fig 9 (now added in the Supporting Information, **Fig. S10**). Isotopic abundances of SP- N_2O , $\delta^{15}N^{\alpha}$ - N_2O , $\delta^{15}N^{\beta}$ - N_2O . **(a)** and **(e)** are based the biofilm-type incubations in XM site. **(b)** and **(f)** are based the biofilm-type incubations at YT site. **(c)** and **(g)** are based the biofilm-type incubations at NN site. **(d)** and **(h)** are based the plastic-type incubations at XM site. Different letters indicate the significant differences ($n=4$, ANOVA, $P < 0.001-0.041$).

These findings indicate that plastisphere biofilms exhibited a greater nitrifying potential compared to other biofilms as well as the surrounding seawater, acting as an overlooked nitrifying niche in estuarine ecosystems. We have revised and rewritten the related part in the manuscript.

Introduction: " *Biofilms represent a crucial microbial life mode in oceans¹. The microorganisms found in biofilms typically span across the entire species, and contribute markedly to global marine biogeochemical fluxes^{1, 2}. Beyond natural biofilms developed on stone debris and floating wood pieces, an emerging "artificial" biofilm (microbial colonization on plastic surfaces) termed as the plastisphere^{3, 4} has elicited widespread interest amidst escalating plastic pollution on the planet^{5, 6}. With massive amounts of plastic debris entering oceans through estuaries, there is a potentially important threat to estuarine organisms and*

ecosystem stability⁷⁻⁹. Hence, the biological consequences of the estuarine plastisphere and ecosystem-level impacts of this floating plastic debris warrant increased attention.

Microbial biomass of the “artificial” plastisphere biofilm is substantial within the global marine environment³, rivaling that of natural biofilms on stone debris and floating wood pieces ($10^8 \sim 10^{11}$ cells g^{-1} wet weight)². We therefore need to explore if the plastisphere mirrors the biogeochemical potential of natural biofilms in estuarine ecosystems or exhibits unique characteristics. In particular, extracellular polymeric substances (EPS) secreted by sessile microorganisms on the plastic surfaces could generate O_2 and nutrient gradients in the plastisphere biofilms^{10, 11}. As a consequence, microbes residing within the self-produced matrix of the plastisphere likely exhibit biogeochemical features different from their planktonic counterparts^{3,10}. Thus, comparison of the characteristics between plastisphere biofilms (sessile mode) and the surrounding seawater (planktonic mode) is vital for the evaluation of biogeochemical fluxes and ecological effects in estuarine ecosystems.” (Lines 47-69).

Methods: *“2.1 Sampling sites and experimental materials. The study areas are located in the estuaries and coasts, including (1) Xiamen (XM), Fujian province (118°11'E, 24°57N); (2) Yantai (YT), Shandong province (121°47E, 37°46N); (3) Nanning (NN), Guangxi province (108°27E, 22°84N). XM and NN have a subtropical climate with 21°C and 22°C mean air temperature and 1000 mm and 1300 mm annual rainfall, respectively. YT has a temperate continental climate with 13°C of the mean air temperature and 524 mm annual rainfall. These estuaries are influenced by anthropogenic activities (such as input of wastewater, nutrients, heavy metals, or other pollutants).*

We sampled the surface seawater of the three sampling sites from June to

August, 2022 and 2023. After collection, the samples were kept in a 4°C ice box and transported back to laboratory. Their water chemical characteristics were measured within 24 hours.

For biofilm type-based materials including plastic and other non-plastic ones (Figure 1b), we used plastic bags (~20 cm×~20 cm polyethylene, purchased from Cleanwrap Co., China), glass balls (3 mm, purchased from Jinggong Co., China), stone debris and wood debris (~1-5 cm collected from environment). These materials were cleaned with 70% ethanol and sterile water prior to experiments. For plastic type-based materials (Figure 1b), we used three types of commercial plastics (polyethylene-PE, polystyrene-PS, and polyvinylchloride-PVC) with densities ranging from 0.88 to 0.97 g cm⁻³. These plastics are used for food bags (PE purchased from Cleanwrap Co., China) or cling films (PS from Chuanguan Co., China; PVC from Jusu Co., China). Prior to in-situ incubation, the plastics were immersed in 70% ethanol solution for 4 h to mitigate the impacts of surface microorganisms and additives in the plastics. The plastic samples (~20 cm×~20 cm) were chosen to be representative of the common types that are present in surface water⁹, rather than attempting to cover all types of commercial plastics.

2.2 Incubation experiments Four principal experiments were conducted in this study to investigate the nitrifying capability of the plastisphere and distinguish keystone nitrifiers, as illustrated in Figure S1.

Experiment 1: in-situ incubations for 28 days. The prepared biofilm type-based materials (plastic, glass, stone, and wood) were each placed into 1 mm net bags, which were then connected by cotton cords²⁴ and placed in the three estuarine seawaters (XM, YT and NN) for 28 days. The plastic type-based materials (PE, PS and PVC) were connected using cotton cords as well, and then submerged at the XM site for 28 days. All the materials floated at 0.1-0.3 m under the water surface. We

also placed 15~50 pieces of materials as back-up samples in nearby regions at 10-m intervals. After 28 days, the samples and the surrounding surface seawater (50 L, 0.3 m below) were harvested, taken to the laboratory and split up into subsamples. A portion of the samples was used for water chemical analysis. Another portion was used for the nitrification assays (Experiment 2 and 3) and rate measurements. The third subsample of the plastic type-based materials was used in the ¹³C-labelled incubation (DNA-SIP assays, Experiment 4). As negligible variation in the nitrification process and nitrifier community in seawater was observed before and after 28 days (Figure S2), we only collected the surface seawater at the end of the incubation.

Experiment 2: biofilm type-based lab-scale nitrification assays for 36 hours. We established five groups in this experiment, including (i) plastic biofilm, (ii) glass biofilm, (iii) stone biofilm, (iv) wood biofilm and (v) surrounding seawater groups. Each group was in quadruplicate (n=4). A total of 3 (three sampling sites)×5 (four types of biofilms and seawater)×4 (quadruplicate samples) = 60 experimental units were obtained in Experiment 2. The harvested materials from Experiment 1 were divided into 10 pieces and transferred into a 120-mL serum bottle. Meanwhile, sterile seawater from the three sites was achieved by filtration with a 0.22- μ m polycarbonate membrane and this water was then used for the biofilm groups to avoid the effects of seawater microorganisms. The bottles were tightly capped after adding 50 mL of sterile seawater and in-situ seawater. Selection of the 50 mL seawater here was due to the minimum water volume for sample submersion. 1 mL of (NH₄)₂SO₄ and 5 mL of O₂ were injected, respectively, reaching final concentrations of 50 μ M NH₄⁺ and 26% O₂ in the bottles. The higher NH₄⁺ level was selected as the initial concentration mainly aiming at comparatively assessing the potential nitrifying capacity of the biofilms and surrounding seawater. These bottles

were then incubated in the dark at 25°C and 120 rpm for 36 h. During the incubation, NH_4^+ , NO_2^- , NO_3^- concentrations were measured with an Ion Chromatograph (Dionex, IC-3000, USA, detection limits: <100 ng/L)²⁵. O_2 and N_2O were determined with a gas chromatography (Agilent 7890A, USA) equipped with TCD and ECD detectors²⁶. Detection limits of O_2 and N_2O are ~3000 ppm and ~320 ppb, respectively. After the incubation, N_2O emission and isotopocules, microbial biomass (16S rRNA gene abundance and cell number) and nitrifier abundances (bacterial and archaeal amoA) were measured. To differentiate the contributions of bacteria and archaea to the nitrification process, we repeated the nitrification assays with adding penicillin. Penicillin can inhibit bacterial growth but not archaea^{27, 28}. Other procedures are as described above.

Experiment 3: plastic type-based lab-scale nitrification assays for 36 hours. Following Experiment 2, we further explored the effects of different plastics on the nitrification process. Two treatments with and without adding 60 μM of the nitrification inhibitor allylthiourea (ATU)²⁹ were established in this experiment. Each treatment included four groups, i.e., the surrounding surface seawater group and the three plastisphere groups (PE, PS and PVC). Each group was in quadruplicate ($n=4$). A total of 1 (one sampling site, XM) \times 2 (presence or absence of ATU) \times 4 (three types of plastisphere and seawater) \times 4 (quadruplicate samples) = 32 experimental units were obtained in Experiment 3. In the plastisphere groups, each type of harvested plastics from Experiment 1 was cut into 10 pieces and transferred into a 120-mL serum bottle. The following procedures were consistent with those in Experiment 2. " (Lines 124-214).

Results: " 3.1 Nitrifying activity in different biofilms and seawater. Four types of plastic and other non-plastic materials, including plastic bags, glass balls, stone, and wood debris, were placed in the three estuaries (XM, YT and NN sites) spanning a

distance of 1870 km for 28 days (Figure 1a and b). Microbial aggregates densely adhered to stone surfaces, more so than to plastics (Figure S5). The biofilms were less abundant on the glass surfaces, and were loose and easily dispersed on the wood surfaces. The stone and wood biofilms harbored more microbial biomass and nitrifiers (AOA and AOB) than the plastisphere and seawater ($P < 0.001$, Figure S6 and S7). Nitrifier biomass in the glass biofilms was the lowest. Except at NN site, AOB abundances ($0.16 \times 10^4 \sim 0.72 \times 10^6$ copies L^{-1}) consistently surpassed those of AOA ($0.54 \times 10^2 \sim 0.21 \times 10^4$ copies L^{-1}) at XM and YT sites, regardless of biofilms and seawater (Figure S7).

We subsequently conducted a 36-h incubation to explore nitrification potential of different biofilms and estuarine seawater. During the incubation, the headspace O_2 concentrations decreased rapidly from 26% to 3% (Figure S8a), but oxic conditions were still maintained throughout the experiment. Among the three sampling sites, variations of NH_4^+ and NO_3^- concentrations in the plastisphere and stone biofilms were more pronounced than in the glass and wood biofilms as well as in the seawater (Figure S9). Notable NO_2^- accumulation was observed in the plastisphere and stone biofilms, except at the YT site. The average rates of ammonia and nitrite oxidation in the plastisphere were $1.79 \sim 3.59$ $fmol\ cell\ h^{-1}$ and $1.39 \sim 3.71$ $fmol\ cell\ h^{-1}$, respectively, significantly higher than in the glass, wood biofilms and the surrounding seawater ($P < 0.01 \sim 0.28$, Figure 1c-e). The rates of glass and wood biofilms were lower than those of the surroundings at all sites ($P = 0.002 \sim 0.072$). More importantly, we found that despite stone biofilms harboring more nitrifying biomass (Figure S6 and S7), their nitrification rates were comparable ($P > 0.05$, Figure 1d-e) or even lower ($P = 0.004 \sim 0.039$, Figure 1c) than those in the plastisphere.

At the end of the incubation, nitrification-derived N_2O emission from the plastisphere was significantly higher than that from stone biofilm at the XM site

($P < 0.001$, Figure 2a), and also higher than emissions from other biofilms and the surrounding seawater at all sites ($P < 0.001$, Figure 2a-c). We further used N_2O isotopocules coupled with the N_2O -SP value to discern the relative contributions of each N_2O emission pathway during nitrification, including NH_2OH oxidation and nitrifier denitrification¹². The values of SP, $^{15}N^\alpha$ and $^{15}N^\beta$ of N_2O are shown in Figure S10. By calculating the N_2O -SP ($\delta^{15}N^\alpha - \delta^{15}N^\beta$), we found that the average values in the biofilms (4.58~9.46‰) were remarkably lower than those in the seawater (19.81~26.68‰, $P < 0.001$, Figure S10), implying distinct N_2O emission dynamics in the biofilms. The primary N_2O emission in the biofilms was from nitrifier denitrification, contributing 53~70% of the total emissions (Figure 2e); however, NH_2OH oxidation dominated in the seawater (62~79%). To elucidate contributions of AOB and AOA to the ammonia oxidation process, we conducted an additional 36 h incubation with and without adding penicillin which inhibits bacteria but not archaea²⁷. With the addition of penicillin (Figure S11), 67~85% of bacteria-mediated NH_4^+ transformation was significantly inhibited, indicating a dominant contribution from AOB. In combination, these findings indicate that the plastisphere biofilms exhibited a greater nitrifying potential compared to other biofilms as well as the surrounding seawater, acting as an overlooked nitrifying niche in estuarine ecosystems.

3.2 Nitrifying activity in different plastisphere and seawater. We further explored the effects of plastisphere types on nitrification potential (Figure S12). Three types of plastic debris (PE, PS and PVC) were incubated in XM estuarine seawater for 28 days, followed by a 36-h incubation with and without adding ATU, a nitrification inhibitor²⁹. Oxidic conditions were still maintained throughout the experiment (Figure S8b). With ATU addition, the transformations of NH_4^+ , NO_2^- and NO_3^- were inhibited (Figure S12b and d), suggesting that microbially mediated nitrification, rather than

biotic assimilation, was responsible for the nitrogen transformations. Consistent with the results from Experiment 2, the average rates of ammonia and nitrite oxidation in the plastsphere were higher than in the surroundings ($P < 0.01$, Figure 1f). No significant difference in nitrification rates was observed between the plastsphere types ($P > 0.05$), except the lower nitrite oxidation rate in the PE plastsphere ($P = 0.041$, Figure 1f).

Nitrification-derived N_2O emission from the plastsphere ($3.5 \sim 4.7 \text{ fmol } N_2O \text{ cell}^{-1}$) after 36 h was 1.6~2.2-fold higher than that from the surrounding seawater ($2.2 \text{ fmol } N_2O \text{ cell}^{-1}$, $P < 0.001$, Figure 2d). Furthermore, the emission via PVC plastsphere was slightly higher than from PE and PS plastsphere ($P = 0.014$ and 0.047 , Figure 2d). Similar with previous results (Figure 2e), nitrifier denitrification dominated the N_2O emission in the plastsphere (67-77%) and NH_2OH oxidation was the main source in the seawater (58-65%, Figure 2f), with AOB-derived NH_2OH oxidation (77%-85%) contributing more than the AOA-derived path (15%-23%). No significant differences in pathway contribution of N_2O emission were found between the three types of plastsphere ($P = 0.23 \sim 0.86$). Overall, our results demonstrate that while the type of plastsphere exerted a minimal influence on the nitrification process, all the plastsphere exhibited nitrifying activities surpassing those of the surrounding seawater. This implies that sessile-mode nitrifiers likely display an enhanced activity compared to their planktonic counterparts. To confirm this, below we will explore biogeochemical mechanisms inherent to the plastsphere and compare them with those in seawater." (Lines 410-486).

Discussion: *"Estuarine ecosystems offer vast surfaces for microbial colonization and growth, fostering dense microbial networks that drive key biochemical processes^{1, 2}. The plastsphere as a footprint of anthropogenic activities represents a new artificial interface³, and its role and significance in biogeochemical cycling are*

*igniting global interest^{24, 58, 59}. Our results show that the estuarine plastisphere exhibited heightened bacterial nitrifying activity relative to the glass and wood biofilms (Figure 1 and 2). Although stone biofilms hosted abundant nitrifying biomass, their nitrifying activities were either comparable or inferior to those in the plastisphere, further underscoring the high nitrification potential of the plastisphere. Notably, we find that the sessile nitrifiers within the plastisphere presented an elevated nitrifying activity compared to the planktonic counterparts in surrounding seawater. The striking niche partitioning among the active core nitrifiers, especially COM nitrifiers, was further observed between the plastisphere and seawater (Figure 3). The plastisphere harbored more biomass of COM nitrifiers, which also had greater expression activities. Through the analysis of high-quality MAGs, we identified a COM nitrifier *Nitrospira_defluvii* (MAG262) in the plastisphere but failed to assemble such nitrifiers in the seawater (Figure 4 and 5). These insights corroborate our hypotheses, highlighting the plastisphere as an underestimated and unique nitrifying niche in estuarine environments." (Lines 599-618).*

3. Data and methodology: The comparison of plastic biofilm is important, but I am disappointed to see that the authors only collected seawater at one time point. For example, it would have been enlightening to collect seawater at the start and end of the experiment. They discuss the importance of microbial succession in their introduction, which can be based on changes in the surrounding seawater microbial community. If they authors have good reason to believe the seawater community did not change significantly over the 28-day experiment, please include that rationale and any citation(s).

Answer:

Thanks. In our previous study (Sci. Total Environ., 2023, 866, 161322), we noted

the minor differences in bacterial and fungal communities in the seawater before and after a short period of *in-situ* incubation. Thus, we expected minimal variations in nitrifier communities as well, and collected the seawater only at the end of the *in-situ* incubation. However, to confirm this, we have now conducted additional assays including nitrification rate, N₂O emission and nitrifier community at the initial and end of the 28-d *in-situ* incubation.

Response to comments Fig 10 (now added in the Supporting information, **Fig. S2**). Nitrification rate, N₂O emission and bacterial *amoA*-based nitrifiers before and after the 28-d *in-situ* incubation. (a) Ammonia oxidation rate and nitrite oxidation rate. (b) N₂O emission after 36-h incubation. (c) Bacterial *amoA*-type nitrifier communities at the genus level.

The results show that nitrification process in the seawater was not strongly impacted during the 28-d *in-situ* incubation. This is likely because compared to the plastisphere, seawater is always in a status of dynamic equilibrium and is frequently replenished by substances and microorganisms from the surroundings. This leads to small variations in the nitrification process and nitrifier community during a short incubation period.

We have added this figure in the Supporting Information and the related explanations in the main text.

"As negligible variation in the nitrification process and nitrifier community in seawater was observed before and after 28 days (Figure S2), we only collected the

surface seawater at the end of the incubation." (Lines 171-174).

4. Are there published papers that include some of the methods described here (e.g., for experiment 2, ¹³C work, etc.)? If so, please add these. Some details of the method – e.g., gas chromatograph settings, limits of detection, etc. – are not included here.

Answer:

Thanks. The references (ref. 24-30) and the details of methods have been added in the revised manuscript.

"..... *Ion Chromatograph (Dionex, IC-3000, USA, detection limits: <100 ng/L)*" (Line 193).

"*Detection limits of gas chromatography for O₂ and N₂O are ~3000 ppm and ~320 ppb, respectively.*" (Line 195).

"*In this study, N₂O and O₂ were analyzed using an Agilent 7890 Gas Chromatograph (GC) equipped with a ⁶³Ni electron capture detector (ECD) and a thermal capture detector (TCD). The sensitivity of the GC is calculated by the coefficient of variation (CV) and is considered to be acceptable when CV <10%. O₂ and N₂O is determined after separation on a Porapak Q column (1.8 m, 80/100 mesh) at 80 °C. N₂O is analyzed with the ECD (350 °C) using argon/methane (90/10, P10) as a carrier gas. O₂ is analyzed with TCD using nitrogen carrier gas. The carrier flow rate was set at ~27 cm³ min⁻¹, which yielded a retention time of ~2.5 min and ~4.6 min for appearance of N₂O and O₂ peaks, respectively.*" (Supporting Information).

5. In the separation of active nitrifiers with ¹³C, there seems to be greater overlap in active vs inactive community in the plastisphere (layers 9, 10; Figure 3a)

than in the surrounding seawater. This could account for differences in the abundances observed. Please see point about clarifying this figure to determine if this is a valid concern.

Answer:

Thanks. As suggested by the reviewer 2, we have now supplemented assays using *amoA* gene as the marker gene instead of 16S rRNA gene to re-analyze the DNA-SIP results (please see the figure below). This is because the *amoA* gene more effectively distinguishes between the ^{13}C - and ^{12}C -labelled DNA in samples. Results show that there is a clear separation for AOB between the heavy DNA (^{13}C -DNA) and the light DNA (^{12}C -DNA) in the plastisphere and seawater. We have now revised our manuscript. Other details please see the Response to Comment 21.

Response to comments Fig 11 (now updated in the main text, **Fig. 3**). DNA-SIP assays. The ^{13}C -DNA (heavy) and the ^{12}C -DNA (light) of active nitrifiers are shown by qPCR of *amoA* across the CsCl buoyant density gradient after the 30-d flush-feeding incubation. The results are normalized using the ratio of *amoA* copy number in each DNA fraction to the maximum *amoA* copy number. Most of the ^{13}C -DNA and the ^{12}C -DNA are accumulated in 9th-10th and 5th-7th fractions, respectively, in both the plastisphere and seawater.

"Quantification of amoA gene abundance as a function of CsCl-DNA buoyant density illustrated the labeling of active nitrifiers (Figure 3a). High peaks of active bacterial nitrifiers in the ^{13}C -DNA were observed in fractions 9-10, while they

remained in fractions 5-7 in the ^{12}C -microcosms in the plastisphere or seawater samples. The shift in CsCl buoyant density from the ^{12}C to the ^{13}C genomes for AOB is $0.016\sim 0.022\text{ g mL}^{-1}$. For active archaeal nitrifiers, the ^{13}C -DNA and ^{12}C -DNA were in fractions 8-9 and 7-8, respectively, likely implying a poor fractionation and less abundance of AOA. " (Lines 491-498).

6. Analytical approach: Generally, very thorough. For statistical analysis, did the authors also confirm homogeneity of variance (e.g., Levene's test)? This is a requirement for ANOVA, and should be confirmed. I suspect it will pass, but if it doesn't pass this test, a non-parametric version of ANOVA should be used. Otherwise, the authors perform a variety of analyses to bolster their conclusions.

Answer:

Yes, the homogeneity of variance test of data regarding 16S rRNA and *amoA* abundances, nitrification rate, N_2O emission and isotopic abundances was conducted.

Response to comments table 1. Homogeneity of variance test of N_2O -SP value.

		Levene Statistic	df 1	df 2	Sig.
SP	Based on Mean	7.841	4	10	0.653
	Based on Median	7.385	4	10	0.949
	Based on Median and with adjusted df	7.385	4	8.15	0.946
	Based on trimmed Mean	7.254	4	10	0.679

Response to comments table 2. Homogeneity of variance test of nitrification rate.

		Levene Statistic	df 1	df 2	Sig.
Rate	Based on Mean	3.868	4	10	0.238

Replies and Explanations to Comments

Based on Median	3.414	4	10	0.351
Based on Median and with adjusted df	3.414	4	6.859	0.354
Based on trimmed Mean	3.652	4	10	0.237

Response to comments table 3. Homogeneity of variance test of N₂O emission.

		Levene Statistic	df 1	df 2	Sig.
Emission	Based on Mean	6.533	4	10	0.457
	Based on Median	6.233	4	10	0.515
	Based on Median and with adjusted df	6.233	4	7.29	0.548
	Based on trimmed Mean	5.646	4	10	0.412

Response to comments table 4. Homogeneity of variance test of 16S rRNA abundance.

		Levene Statistic	df 1	df 2	Sig.
16S	Based on Mean	9.503	4	23	0.712
	Based on Median	8.644	4	23	0.777
	Based on Median and with adjusted df	8.644	4	12.15	0.748
	Based on trimmed Mean	7.585	4	23	0.723

Response to comments table 5. Homogeneity of variance test of *amoA* abundance.

		Levene Statistic	df 1	df 2	Sig.
amoA	Based on Mean	3.735	4	10	0.201
	Based on Median	2.525	4	10	0.237
	Based on Median and with adjusted df	2.525	4	8.877	0.243
	Based on trimmed Mean	3.662	4	10	0.172

We have now clarified in the revised manuscript.

"Prior to one-way ANOVA analysis, we tested for the homogeneity of variances (Levene's test) and the normality of residuals." (Line 404).

Suggested improvements:

7. Title: I think the title is missing some words for clarity. I suggest "Nitrifying niche in the ocean is expanded by the plastisphere". Also, 'nitrifying niche' is a bit confusing. Could 'nitrification' be a more direct wording?

Answer:

Thanks for the suggestion. We have now revised the title.

Nitrifying niche used in the title refers to nitrification process (including rate and dynamics), nitrifier communities and nitrifier metabolisms, which are all involved in the present study. On the other hand, we aim to highlight that the plastisphere can be a "niche" for nitrifier colonization in estuarine ecosystems. Although we think the "nitrification" is a direct wording, it may have a narrow scope and do not cover all of work in this study. After carefully considering, we think "nitrifying niche" could be more appropriate in the title.

"Title: Nitrifying niche in estuaries is expanded by the plastisphere" (Line 1).

8. Abstract: Line 27: Please rephrase to "The oceanic plastisphere...". Can delete 'here' in line 32.

Answer:

Thanks. Corrected.

9. Methods: Please provide further details on the plastics. Where were they sourced from? Did you perform any analysis on additives in these plastics? If not, do you suspect any based on their applications (e.g., where you purchased from or

the product type). Additive content may be a bigger driver of biofilm formation than polymer chemistry itself.

Answer:

Thanks. We have now provided the source information of these plastics.

"These plastics are used for food bags (PE purchased from Cleanwrap Co., China) or cling films (PS from Chuanguan Co., China; PVC from Jusu Co., China)." (Line 148).

In the present study, we did not analyze the additives of these plastics for the following reasons. (1) Prior to the *in-situ* incubation, the plastics were immersed in 70% ethanol solution for 4 h to minimize the impacts of surface microorganisms and additives in the plastics. (2) Based on our results, the differences in nitrification process among the three plastic types were slight (Figures 1 and 2 in the main text). In combination, we think that the effects of plastic additives on nitrifier communities and nitrification process are likely negligible in this study. We have now added some descriptions to avoid confusion.

"Prior to in-situ incubations, the plastics were immersed in 70% ethanol solution for 4 h to mitigate the impacts of surface microorganisms and additives in the plastics." (Lines 150-152).

10. Line 154 – don't spell out milliliter

Answer:

Thanks. Corrected.

"1 mL of (NH₄)₂SO₄..." (Line 186).

11. Are the experiment numbers in figure S2 consistent with the text? I think experiment 2 and 3 are flipped.

Answer:

We have now revised the figure to make the experiment numbers consistent with those in the main text (section Materials and Methods, 2.2).

Response to comments Fig. 12 (now updated in the Supporting Information, **Supplementary Fig. 1**). Experimental workflow of this study. The experiments in this study includes **1** *in-situ* (28 days, Experiment 1), **2** biofilm type-based lab-scale incubations (36 hours, Experiment 2), **3** plastic type-based lab-scale incubations (36 hours, Experiment 3), and **4** ^{13}C -substrate flush feeding incubation of DNA-SIP assays (Experiment 4).

12. Can the reader better define site preference in the text? Functionally, what does this mean? This is difficult to interpret, despite the figure (2b) and method being clear.

Answer:

Sorry for the confusion. The site preference (SP) of N_2O , introduced by two Japanese scientists in 2000 in Nature, has been widely applied in the research associated with N_2O cycling. The SP value is a promising method to distinguish various N_2O production sources in natural ecosystems. This is because N_2O

produced from different processes has distinctive preferential cleavages of the $^{14}/^{15}\text{N}-^{16}\text{O}$ bond in their symmetric intermediates. It thus generates different enrichments of $^{15}\text{N}^\alpha$ ($^{14}\text{N}-^{15}\text{N}-^{16}\text{O}$) or $^{15}\text{N}^\beta$ ($^{15}\text{N}-^{14}\text{N}-^{16}\text{O}$) in N_2O and leads to the distinct SP value.

To avoid confusion, we have now added some descriptions to introduce the function of the N_2O -SP method.

" N_2O produced from the two pathways has a unique preferential cleavage of the $^{14}\text{N}-^{16}\text{O}$ and $^{15}\text{N}-^{16}\text{O}$ in the intermediates, generating different enrichments of $^{15}\text{N}^\alpha$ ($^{14}\text{N}-^{15}\text{N}-^{16}\text{O}$) and $^{15}\text{N}^\beta$ ($^{15}\text{N}-^{14}\text{N}-^{16}\text{O}$) and thus leading to the unique SP- N_2O value^{31, 32}" (Lines 239-242).

13. Figures and captions: I suggest not using W and P for seawater and plastic. This is confusing in the caption, and not necessary for reading the figures. These words are fairly short, so please spell them out.

Answer:

Thanks. Revised.

14. In figure 3a, can you add vertical lines to show collection of layers 9 & 10 (or, transparent shaded boxes over the density of regions 9 and 10)? This would help address potential bias of this approach mentioned previously. Figure 3c is also very blurry; please use high resolution copy for figure upload, if you have not already.

Answer:

Thanks. We have now revised the figure as suggested.

Replies and Explanations to Comments

Response to comments Fig. 13 (now updated in the main text, **Figure 3**). Active nitrifiers in the plastisphere and the surrounding seawater. (a) DNA-SIP (n=3). The ¹³C-DNA (heavy) and the ¹²C-DNA (light) of active nitrifiers are shown by qPCR of *amoA* gene across the CsCl buoyant density gradient after the 30-d flush-feeding incubation (Experiment 4). The results are normalized using the ratio of *amoA* gene copy number in each DNA fraction to the maximum *amoA* gene copy number. (b) Compositions (%) of the active nitrifiers including AOA, AOB, NOB and COMAMMOX, based on the sequencing of metagenomics with ¹³C-DNA (circle) and the metatranscriptomics (star). (c) Phylogenetic analysis of active nitrifiers based on the sequencing of 16S rRNA, *amoA*-AOA, *amoA*-AOB and metagenomics.

15. Figure 5: in the third part (FPKM) I assume each box correlates to the gene to the left, but this isn't explicitly detailed in the caption. Please clarify. Also, is there a test to validate statistical differences between plastisphere and seawater here?

Although in different genus, the genes look very similar overall (including all MAGs) in gene presence/absence and abundance between seawater and plastisphere.

Answer:

Thanks. We have now clarified it in this revised version.

In this figure, we did not conduct statistical tests of the abundances of these nitrifying genes. This is because we aim to depict the different MAGs associated with nitrifiers detected in the plastisphere and seawater, as well as the nitrifying genes identified in these MAGs and their expression levels. Statistical tests of the genes and transcripts associated with microbial metabolisms and nitrification process in these MAGs were conducted and shown in another figure (Supporting Information Fig. S17).

Response to comments Fig. 14 (now updated in the main text, **Figure 5**). Genomic and transcriptional information of the high-quality MAGs associated with nitrification process. The presence or absence of the seven nitrifying genes is shown at both the gene and transcript levels. The transcriptional levels of each nitrifying gene were calculated on the basis of Fragments Per Kilobase of exon model per Million mapped reads (FPKM).

16. Figure 6: it is difficult to distinguish between the different green colors, as

well as the reds/oranges. Please consider different color choices.

Answer:

Thanks. Revised.

Response to comments Fig. 15 (now updated in the main text, Figure 6). Metatranscriptomics showing the differences in metabolisms and cooperations of active nitrifiers in the plastisphere and the surrounding seawater.

17. Reference 2 – first author last name is misspelled.

Answer:

Corrected. We are sorry for the misspelling.

In addition, insights gleaned from another seminal paper by the author, published in Nature Computational Science journal, have also significantly informed our perspective. This paper posits that process-based mass-balance models may serve as a robust platform for the systematic integration of knowledge pertaining to plastic pollution, based on its quantifiable intrinsic properties. Such an approach potentially facilitates the assessment of the plastic pollution footprint and the

qualification of plastisphere surfaces within global estuarine environments. This influential work has been cited in the manuscript.

"6. MacLeod, M.; Arp, H. P. H.; Tekman, M. B.; Jahnke, A., *The global threat from plastic pollution. Science, 373, (6550), 61-65 (2021).*"(Reference 7).

"76. MacLeod, M.; Domercq, P.; Harrison, S.; Praetorius, A., *Computational models to confront the complex pollution footprint of plastic in the environment. Nat. Comput. Sci. 3, (6), 486-494 (2023).*"(Reference 76).

Reviewer #2 Comments:

This manuscript describes research into the colonization, growth, and activity of nitrifiers in the plastisphere – i.e., determining whether the colonization of plastic which is now abundant in the ocean represents a new nitrifying niche. The study involves an impressive amount of work using stable isotopes, metagenomics /transcriptomics, and biogeochemical data to demonstrate (successfully) which populations are likely active in these samples.

18. However, I have reservations about the choice of environment and interpretation of some of the experimental data, and whether the study demonstrates that the plastisphere is really a unique habitat for nitrification, or whether it simply shows that plastic provides a surface for microbial colonization like any other material in the ocean (e.g., glass, metal, rope, etc.).

Answer:

Thanks for these comments. We have conducted additional experiments as part of the revisions, and selected four types of plastic and other non-plastic materials (i.e., plastic bags, glass balls, stone debris, and wood debris) as suggested to reveal if the plastisphere is a "unique" nitrifying niche or just offers a "surface"

like any other materials. These selected materials are very common in estuarine environments. The detailed results and explanations are shown in the Response to Comments 2. Our results indicate that compared to other biofilms and surrounding seawater, the plastisphere exhibited an elevated nitrification potential. Combining with the results of metabolic disparities and niche differentiation of active nitrifying communities between the plastisphere (sessile or attached mode) and seawater (planktonic or free-living mode), our results highlight that the plastisphere likely acts as a unique nitrifying niche in estuarine ecosystems.

19. A central assumption of the manuscript is that the plastisphere is a unique and previously unrecognized habitat for nitrifiers and nitrification in the ocean. For this to be something novel compared to communities and nitrifiers growing on surfaces and biofilms, I think there would have to be a demonstration that there is something unique to plastics which support their growth e.g., comparing growth/activity of nitrifiers on other control surfaces (e.g., glass). There is a lot of research showing that nitrifiers like to grow on surfaces (e.g., wastewater, laboratory reactors of both AOA AOB, etc.) and are hotspots for activity compared to planktonic cells, and I am uncertain that this really demonstrates something surprising.

Answer:

Thanks for this comment. We have conducted additional experiments as mentioned in the Response to Comments 2, and have now clarified the novelty of this study in the revised manuscript. Aspects of novelties include:

(1) Compared to the surrounding seawater and other types of biofilms (glass, stone, and wood biofilms), whether does the plastisphere present a "unique" nitrifying niche in estuarine environment and why?

(2) Do active AOA, AOB, NOB and COM nitrifiers differ between the plastisphere and surrounding seawater, and what are their contributions to estuarine nitrification?

(3) Do the metabolisms of active nitrifiers differ between the plastisphere (sessile mode) and seawater (planktonic mode), and what are the differences?

"The specialized features of the plastisphere lead us to hypothesize that (1) the estuarine plastisphere represents an overlooked and even unique niche of nitrification with higher nitrifying activity than the surrounding seawater and other biofilms, and (2) it harbors distinctive active nitrifiers and metabolic behaviors from the seawater." (Lines 107-111).

20. The title of the paper and discussion would indicate that this study aims to look at the interaction of nitrifiers and plastics in the (open?) ocean. This study appears to have used an estuarine environment with a significant terrestrial influence. For example, the pH of the water is 6.8 (the open ocean is ~8.1 I think) and *Nitrosocosmicus*, *Nitrososphaera* and *Nitrotoga* are not typical marine nitrifiers found or active in the open ocean. This is fine, but I think interpretation to the ocean should be reconsidered.

Answer:

Thanks. We agree with the point and have now revised the manuscript to focus on the estuarine environments.

21. DNA-SIP. The data presented and interpretation is puzzling. For both control water and plastic samples, a small increase in the buoyant of the most abundant fraction/the peak in the ¹³C samples is interpreted as indicating growth of nitrifiers on ¹³C. However, as this is profiling total 16S rRNA genes, and not

nitrifier specific amplicons (e.g., amoA), would you expect to see shifts in the total (not partial) community from autotrophic ^{13}C incorporation alone, particularly when these populations represent less than 0.01% of the total community (based on numbers in Figure S7)? This would require a lot of cross-feeding. Secondly, the shift in buoyant density from a fully ^{12}C to a fully ^{13}C genome is $\sim 0.04 \text{ g ml}^{-1}$ (e.g., see profiles of Jia and Conrad, 2009; Zhang et al., 2012 ISME), and in this data, the shifts are in the order of 0.005 g ml^{-1} , and where the whole community is shifting slightly, and not just a small proportion of the (autotrophic) community growing. I am not sure what could explain these very slight shifts, but I assume it is unlikely that in a total community, all populations have received a minor and equal ^{13}C enrichment. But in summary, interpreting these as indications of activity should be treated with caution. As you are preparing metagenomes and metatranscriptomes from recently colonized material, these will likely represent active communities at some level, but the use of DNA-SIP for targeting these is not convincing.

Answer:

Thanks for this comment, and we agree with the point.

As suggested, we have now re-conducted qPCR assays of the fractionated DNA from ^{12}C - and ^{13}C -microcosms using *amoA* gene as the marker gene instead of 16S rRNA gene. Please see the figure below.

Response to comments Fig 16 (now updated in the main text, **Fig. 3**). DNA-SIP assays. The ^{13}C -DNA (heavy) and the ^{12}C -DNA (light) of active nitrifiers are shown by qPCR of *amoA* across the CsCl buoyant density gradient after the 30-d flush-feeding incubation. The results are normalized using the ratio of *amoA* copy number in each DNA fraction to the maximum *amoA* copy number. Most of the ^{13}C -DNA and the ^{12}C -DNA are accumulated in 9th-10th and 5th-7th fractions, respectively, in both the plastsphere and seawater.

Results show that there is a clear separation in bacterial *amoA* gene between the heavy DNA (^{13}C -DNA) and the light DNA (^{12}C -DNA) in the plastsphere and seawater samples. The shift in CsCl buoyant density from the ^{12}C to the ^{13}C genomes for AOB is 0.016~0.022 g mL⁻¹. By contrast, the poor fractionation of archaeal *amoA* gene likely indicates low abundance of AOA in samples, which is consistent with the results of amplicon and metagenome sequencing.

The manuscript has been revised accordingly.

"Quantification of amoA gene abundance as a function of CsCl-DNA buoyant density illustrated the labeling of active nitrifiers (Figure 3a). High peaks of active bacterial nitrifiers in the ^{13}C -DNA were observed in fractions 9-10, while they remained in fractions 5-7 in the ^{12}C -microcosms in the plastsphere or seawater samples. The shift in CsCl buoyant density from the ^{12}C to the ^{13}C genomes for AOB is 0.016~0.022 g mL⁻¹. For active archaeal nitrifiers, the ^{13}C -DNA and ^{12}C -DNA were in fractions 8-9 and 7-8, respectively, likely implying a poor fractionation and less abundance of AOA. " (Lines 491-498).

22. Are the (high?) ammonium concentrations used in incubation experiments realistic for looking at realistic in situ activities of nitrifiers in the ocean? For the SIP incubations, the samples receive a lot of NH_4^+ every two days, perhaps making it more like a laboratory enrichment than a realistic analogue of the open ocean. !

think you maybe need explain the choice of these concentrations.

Answer:

Thanks. Yes. A slightly higher concentration of initial NH_4^+ (50 μM) than that under *in-situ* estuarine conditions (12~30 μM) was used in the lab-scale incubations. This is because, on the one hand, we aim to explore and compare the potential nitrifying capacity and nitrification rate in different biofilms and the surrounding seawater. On the other hand, under *in-situ* conditions, substantial fluxes of NH_4^+ from surroundings would naturally replenish the NH_4^+ consumed, and thus the NH_4^+ concentration could remain at a similar level. In laboratory microcosms, however, the NH_4^+ consumed by microorganisms cannot be supplied immediately from the surroundings. Thus, using the *in-situ* NH_4^+ levels would result in rapid consumption, hindering explicit exploration of nitrification potential. Thus, we applied a higher NH_4^+ level in the laboratory incubations.

For the SIP incubation experiment, we aim to culture, select, and distinguish the active nitrifiers between the plastsphere (sessile mode) and seawater (planktonic mode). Thus, to maintain a high level of nitrifying activity and to enrich ^{13}C -labelled DNA in nitrifiers, we supplied NH_4^+ and $\text{HCO}_3^-/\text{CO}_2$ every two days.

To avoid confusion, we have now explained the selection of a higher NH_4^+ level used in the revised manuscript.

"The slightly higher NH_4^+ level was selected as the initial concentration mainly aiming at comparatively assessing the potential nitrifying capacity of the biofilms and surrounding seawater." (Lines 188-190).

"This is because we aim to culture and enrich ^{13}C -labelled DNA in nitrifiers, and thus to distinguish active nitrifiers between the plastsphere and seawater." (Line 232).

23. Line 71 – ‘*Nitrospira*-like’. They are not ‘like’, but unambiguously *Nitrospira*.

Answer:

Thanks. Corrected.

"..... *Nitrospira* COM nitrifiers" (Line 79).

24. Line 72 – A few issues with this statement. It reads as if NOB are fixing N₂. Also, this is a 2008 reference which predates, for example, discovery of comammox and likely good predictions of AOA activity. Do you mean that 90% of fixed nitrogen is transformed from nitrite to nitrate by NOB?

Answer:

To avoid confusion, we have now deleted the sentence.

25. Line 78 – I believe this is a purely AOB-centric view of N₂O production. N₂O production in AOA comes from the abiotic reactions of intermediates and generally are not considered to perform nitrifier denitrification. Maybe expand this sentence a little to differentiate the pathways of AOA and AOB (and Comammox?).

Answer:

Thanks. Revised.

"Nitrification can cause the emission of the greenhouse gas N₂O as a byproduct, but exists different mechanisms in AOB and AOA. N₂O emission from AOB strains occurs via NH₂OH oxidation and nitrifier denitrification pathways¹⁵⁻¹⁷, whereas AOA-mediated emission primarily arises from abiotic (hybrid) formation, involving one N atom from NH₂OH and another from NO₂^{-18,19}." (Lines 84-88).

"The contribution of COM nitrifiers to N₂O emission remains contentious²¹, with some suggesting a non-negligible contribution despite lower levels²², while others

arguing that COM nitrifiers do not produce N_2O^{23} . This discrepancy largely hinges on the biomass of COM nitrifiers and environmental factors²¹." (Lines 95-98).

26. Line 132 – ‘To prevent any loss’? I am uncertain what this means. To you mean back-up samples?

Answer:

Yes, it means the back-up samples. Corrected.

"We also placed 15~50 pieces of materials as back-up samples in nearby regions at 10-m intervals." (Line 165).

27. Line 151 – Add ‘filtration’ into this sentence.

Answer:

Thanks. Added.

"Meanwhile, the sterile seawater from the three sites was achieved by filtration with a 0.22- μ m polycarbonate membrane and this water was then used for the biofilm groups to avoid the effects of seawater microorganisms." (Line 182).

28. Line 273 – Do you need to repeat the cycling conditions here? They are also in the cited table.

Answer:

Deleted.

29. Line 279 – Conditions.

Answer:

Corrected.

30. Line 380 – ‘The plastisphere harbored more microbial biomass, especially the nitrifiers, than the seawater ($P < 0.001$, Figure S7), which explains the higher nitrification rates of the plastisphere (Figure 1).’ I am uncertain why this is ‘especially the nitrifiers’ as they seem to constitute a very low proportion of the community i.e., for the four plastics, total 16S rRNA gene abundance and cell numbers were greater than 10^8 , but AOA and AOB amoA gene abundances were around 10^2 and 10^4 respectively. While I appreciate that there are differences between cell vs gene abundances, or copy numbers per genome, nevertheless, this would indicate that they ~0.01 and 0.0001% of the total communities?

Answer:

Sorry for the confusion. We have now deleted the word "especially". As the additional experiments and results have been added during revisions, we have rewritten this part.

"Microbial aggregates densely adhered to stone surfaces, more so than to plastics (Figure S5). The biofilms were less abundant on the glass surfaces, and were loose and easily dispersed on the wood surfaces. The stone and wood biofilms harbored more microbial biomass and nitrifiers (AOA and AOB) than the plastisphere and seawater ($P < 0.001$, Figure S6 and S7). Nitrifier biomass in the glass biofilms was the lowest. Except at NN site, AOB abundances ($0.16 \times 10^4 \sim 0.72 \times 10^6$ copies L^{-1}) consistently surpassed those of AOA ($0.54 \times 10^2 \sim 0.21 \times 10^4$ copies L^{-1}) at XM and YT sites, regardless of biofilms and seawater (Figure S7)." (Lines 413-421).

31. Figure 6 – This figure indicates NOB as a source of N_2O production. Can you confirm there is experimental evidence from cultures that NOB produce N_2O ?

Answer:

Sorry for the error. It is the AOB *Nitrosospira* (MAG397) as a source of N₂O emission via nitrifier denitrification pathway. We have corrected the figure.

Response to comments Fig. 17 (now updated in the main text, **Figure 6**). Schematic representations, at the transcriptional level, of microbial metabolisms of the five nitrifying MAGs and their cooperations in the plastisphere (a) and surrounding seawater (b).

32. Table S2. Add references for primers. Archaea is spelt incorrectly. The Svedberg unit of 16S is with an uppercase S.

Answer:

Thanks. Added and corrected.

Response to comments table 6. (now updated in the Supporting Information, **Table S2**). Primers for targeting microbial communities and nitrifiers in the plastisphere and surrounding seawater.

Replies and Explanations to Comments

Genes	Primers	Primers (5'-3')	PCR conditions
AOA amoA ⁽⁵⁾	archaea- amoA F	STAATGGTCTGGCTTAGACG	5 min at 95°C, 35 cycles consisting of 45 s at 94°C, 1 min at 53°C and 1 min at 72°C.
	archaea- amoA R	GCGGCCATCCATCTGTATGT	
AOB amoA ⁽⁵⁾	amoA -1F	GGGGTTTCTACTGGTGGT	5 min at 95°C, 35 cycles consisting of 30 s at 94°C, 30 s at 55°C and 1 min at 72°C.
	amoA -2R	CCCCTCKGSAAGCCTTCTTC	
NOB Nitrospina 16S (qPCR) ⁽⁶⁾	130F	GGGTGAGTAACACGTGAATAA	94°C for 3min; 35×(94°C for 15s; 57.5°C for 15s; 72°C for 30s; 77°C for 7s)
	282R	TCAGGCCGGCTAAMCA	
NOB Nitrospira 16S (qPCR) ⁽⁷⁾	616F	AGAGTTTGATYMTGGCTC	95°C for 4min; 35 cycles×(94°C for 30s; 56°C for 30s; 72°C for 60s; 72°C for 7min)
	1158R	CCCGTTMTCCTGGGCAGT	
NOB Nitrospira nxB (qPCR) ⁽⁸⁾	169F	TACATGTGGTGAACA	95°C for 5min; 35×(95°C for 40s; 56°C for 40s; 72°C for 90s; 72°C for 10min)
	638R	CGGTTCTGGTCRATCA	
NOB Nitrobacter nxA (qPCR) ⁽⁹⁾	F1370 F1	CAGACCGACGTGTGCGAAAG	3 min at 94°C followed by 35 cycles of 94°C for 30s, annealing at 55°C for 45s and at 72°C for 45s with terminal elongation at 72°C for 5min
	F2843 R2	TCCACAAGGAACGGAAGGTC	
Comammox amoA clade A ⁽¹⁰⁾	comaA-244F	TAYAAVTGGGTSAAAYTA	50 °C for 2 min and 95 °C for 10 min, followed by 40 cycles of 15 s at 95 °C, 30 s at 52 °C, and 40 s at 72 °C
	comaA-659R	ARATCATSGTGCTRTG	
Comammox amoA clade B ⁽¹⁰⁾	comaB-244F	TAYTTCTGGACRTTYTA	
	comaB-659R	ARATCCARACDGTGTG	
16S rRNA ⁽¹¹⁾	515F	GTGCCAGCMGCCGCGG	95°C for 3 min, 30 cycles of 95°C for 30s, 55°C for 30s and 72°C for 45s, 72°C for 10min
	907R	CCGTCAATTCMTTTRAGTTT	
16S rRNA ⁽¹²⁾	Arch344F	ACGGGGYGCAGCAGGCGCGA	95°C for 3 min, 30 cycles of 95°C for 30s, 55°C for 30s and 72°C for 45s, 72°C for 10min
	Arch915R	GTGCTCCCCCGCAATTCCT	

33. Figure S1 – spaces missing between numbers and units.

Answer:

Thanks. Revised accordingly. Please see the Response to Comments Fig. 12.

REVIEWER COMMENTS

Reviewer #2 (Remarks to the Author):

Figure 3c – I apologise I didn't notice this before. There seems to be a large issue with phylogeny which doesn't conform with standard/expected lineages. Firstly, from the legend it is not clear what gene(s) are being used to perform the phylogeny with 16S rRNA, amoA and MAGs all mentioned. But no details of how the tree is constructed. Secondly, the general branching order doesn't make sense – e.g. the Nitrobacter and Nitrospira sequences are split into multiple lineages with some Nitrobacter and Nitrospira grouping together to the exclusion of other Nitrobacter and Nitrospira lineages. I don't understand why all Nitrospira or Nitrobacter sequences don't individually form a monophyletic group. This is exemplified with Nitrospira defluvii forming a monophyletic group with AOA sequences, and with good bootstrap support, to the exclusion of other Nitrospira.

I found the description of reciprocal cross-feeding very definitive when it should be quite speculative. E.g. 'We observed a distinctive cooperation of substrate exchange' when as far as I can tell you did not observe transfer of chemicals but are building models based on genes/transcripts alone. Is exchange of NO between nitrifiers a recognised phenomenon as it is for say urea or nitrite? I couldn't tell from the cited references.

Isotopomer analysis. I would recommend carefully describing what the ranges are for site preference for the different processes, including AOA vs AOB and comammox if available. For example, I think some of the values interpreted as 'nitrifier denitrification' overlap with those observed for AOA (Jung et al. 2014).

General comments

48 – I don't know what this means.

95 – lower levels of what?

103 – Remove '-based'

116 – I'm not really sure what 'keystone' means and seems a bit of an unnecessary buzzword. Removing it won't change the meaning.

116 – potential metabolic differences?

132 - We sampled the surface seawater of the three sampling sites from June to August, 2022 and 2023. In the original MS this was July and August 2021. But Figure 1f is the same data as Figure 1 in the original.

133 – Same months in both years? Could be rephrased.

206 – Why was ATU added? Is this not the same assay as experiment 2?

171 – ‘As negligible variation in the nitrification process and nitrifier community in seawater was observed before and after 28 days (Figure S2), we only collected the surface seawater at the end of the incubation.’ This confuses me – are you not talking about the same samples – i.e. day 28? You observed little variation at day 28 so you only collected certain samples at day 28?

227 – 1 mM

314 – red water?

319 – remove slightly

336 – GenBank

354 – What PCR assays?

360 – ‘The RNA integrity number ranges from 8.0 to 9.1’. What units?

413 – ‘... more so than to plastic.’ This is not quantitative and also cannot be determined from the pictures. I would remove.

454 – ‘In combination, these findings indicate that plastisphere biofilm exhibited a greater nitrifying potential compared to other biofilms as well as the surrounding seawater, acting as an overlooked nitrifying niche in estuarine ecosystems.’ The data suggest to me that for 2 of the three sites that the biofilms have similar N transformation rates compared with stone. I think a more accurate description is required.

494 – I would describe buoyant density variation rather than fraction numbers

497 - “likely implying a poor fractionation and less abundance of AOA”. Why would the fractionation be poor when you got good separation for bacteria. Also, is ‘bacterial nitrifiers’ AOB or comammox? The primers don’t pick up both. Why not to all three groups?

498 – What fractions (with BD values) were specifically sequenced? There seems to be quite a range separating the AOA and bacteria.

500 – No labelling of nxrA or B? Nitrospira or other groups? This contradicts the next paragraph where you describe detection of active Nitrospira and Nitrobacter.

521 - followed by Nitrospira_sp. (6.5% and 0.12%) and uncultured Nitrospira_sp. (3.9% and 0.08%). These names are not informative. Can you use different descriptors?

527 - These are not high but medium quality. Using the Bowers et al.2017 community standard, high quality is >90% and <5% contamination.

673 – ‘We observed a distinctive cooperation of substrate exchange (NO₂- and NO) among the COM’. I am not sure how you observed this. Is this just speculation from the metaG/T data?

686 – I am uncertain whether the Daims et al. models (cited) describe sharing of NO

Reviewer #3 (Remarks to the Author):

I appreciate the opportunity to review this interesting article. I think the authors have properly addressed and resolved all of the questions and critiques raised by the two reviewers.

Reviewer #4 (Remarks to the Author):

The plastisphere, which comprises the microbial community on plastic debris, is a growing emphasis in environmental sciences as the occurrence of plastic debris is present in aquatic and terrestrial systems. Less known is how the microbial communities on the surface of plastics facilitate biogeochemical processes. The manuscript that was submitted and reviewed provides insight into nitrogen cycle processes that occur in the plastisphere. In their revision, they conducted several additional experiments to compare nitrogen cycle processes on plastics to those of other substrates and demonstrate the varying degree to how nitrogen fixation and cycling differs between plastics and other substrates (glass, wood, stone).

The submitted work is a valuable contribution to the field as plastic research is garnering research attention; there has been less of a focus on biogeochemical processes governed by microbial communities. As microbes colonize plastics, their function may differ for various reasons. This submitted paper starts to uncover how processes may vary, and if better understood, we can understand how specific biogeochemical cycling may differ in areas where pollution is more significant.

The authors have done a great job revising the document in response to reviewers. They have taken the extra step to include experiments supporting their findings while addressing the uncertainty described in their initial review. The author's thorough review leaves little flaws to take away from their work and conclusions. The methodology is sound, from their experimental design to their analysis and molecular approaches, while informing their work from previous lab work and expanding to answer relevant questions. This work's findings are important and provide insight for future work in other matrices, like freshwater ecosystems.

Replies and Explanations to Comments

We thank the three reviewers for their constructive comments. We have expanded our data analyses of DNA-SIP as suggested by the reviewer 2. We have now revised the manuscript thoroughly and highlighted the changes in track-change mode. All questions raised by the reviewers have been answered point-by-point below.

Reviewer #2 Comments:

1. Figure 3c – I apologise I didn't notice this before. There seems to be a large issue with phylogeny which doesn't conform with standard/expected lineages. Firstly, from the legend it is not clear what gene(s) are being used to perform the phylogeny with 16S rRNA, *amoA* and MAGs all mentioned. But no details of how the tree is constructed. Secondly, the general branching order doesn't make sense – e.g. the *Nitrobacter* and *Nitrospira* sequences are split into multiple lineages with some *Nitrobacter* and *Nitrospira* grouping together to the exclusion of other *Nitrobacter* and *Nitrospira* lineages. I don't understand why all *Nitrospira* or *Nitrobacter* sequences don't individually form a monophyletic group. This is exemplified with *Nitrospira defluvii* forming a monophyletic group with AOA sequences, and with good bootstrap support, to the exclusion of other *Nitrospira*.

Answer:

We appreciate the thorough review and insightful comments from the reviewer, which greatly improves the manuscript quality.

We are sorry for errors in the phylogenetic tree in Figure 3c. Initially, we attempted to consolidate AOA, AOB, COM and NOB into one tree using 16S rRNA, *amoA*, and MAG sequences, which likely introduced inaccuracies. We have now

rectified this by constructing separate, maximum-likelihood phylogenetic trees for each group based on amplicon sequencing data (please see the figures below). Specifically, AOA, AOB and COM trees were built using sequences from their respective *amoA* gene datasets. For NOB, we designated ASVs from 16S rRNA sequences affiliated with *Nitrospira*, *Nitrobacter* and *Nitrotoga* as candidate NOB ASVs for phylogenetic reconstruction. Additionally, sequences from AOB-*amoA*, COM-*amoA*, or 16S rRNA extracted from the five nitrifying MAGs were integrated into the corresponding AOB, COM, or NOB trees.

Response to comments figure 1. (now updated in **main text, Fig. 3c**). Phylogenetic tree of AOA *amoA* gene. The 9 typical ASVs contributing to 52.9~73.4% of the total AOA sequences were used to construct the tree at 97% similarity. Labels “P” and “W” represent the plastisphere and seawater origins, respectively. For example, the designation of ASV32 (P: 7.6; W:1.4) indicates its proportion of the total AOA *amoA* sequences-7.6% in the plastisphere and 1.4% in seawater. Bootstrap support values exceeding 50% are displayed at branching nodes, based on 1000 replicates.

Response to comments figure 2. (now updated in **main text, Fig. 3c**). Phylogenetic tree of AOB *amoA* gene. The 12 typical ASVs contributing to 58~77.1% of the total AOB sequences were selected based on AOB-associated MAGs (MAG876 and 397) and used to construct the tree at 97% similarity.

Response to comments figure 3. (now updated in **main text, Fig. 3c**). Phylogenetic tree of COM *amoA* gene. The 11 typical ASVs contributing to 73% of the total COM sequences were used to construct the tree at 97% similarity. The COM-*amoA* gene was not successfully amplified in the seawater samples.

Response to comments figure 4. (now updated in **main text, Fig. 3c**). Phylogenetic tree of NOB 16S rRNA gene. The 7 typical ASVs contributing to 2.8~2.9% of the total 16S rRNA sequences were selected based on COM/NOB-associated MAGs (MAG262, 715 and 1473) and used to construct the tree at 97% similarity.

We have now corrected the figure 3 and revised the relevant text.

Replies and Explanations to Comments

Response to comments figure 5. (now updated in **main text**, **Fig. 3**). Active nitrifiers in the plastisphere and the surrounding seawater. **(a)** DNA-SIP (n=3). The ¹³C-DNA (heavy) and the ¹²C-DNA (light) of AOA, AOB and COM bacteria are shown by qPCR of *amoA* across the CsCl buoyant density gradient after the 30-d flush-feeding incubation (Experiment 4). The results are normalized using the ratio of *amoA* copy number in each DNA fraction to the maximum *amoA* copy number. Most of the ¹³C-DNA and the ¹²C-DNA are accumulated in buoyant density 1.699-1.702 and 1.680-1.686 g mL⁻¹, respectively, in both the plastisphere and seawater. **(b)** Compositions (%) of the active nitrifiers including AOA, AOB, NOB and COM bacteria, based on the sequencing of metagenomics with ¹³C-DNA (circle) and the metatranscriptomics (star). The species of the major genera, such as *Nitrosomonas*, *Nitrospira*, *Nitrobacter*, and *Nitrotoga* are shown in Fig. S11. **(c)** Phylogenetic analysis of active nitrifiers based on the sequencing of AOA-*amoA*, AOB-*amoA*, COM-*amoA* and NOB-16S rRNA (maximum-likelihood method) at 97% similarity. The 9, 12, 11 and 7 typical ASVs (highlighted in blue) contributing to 52.9~73.4%, 58~77.1%, 73% and 2.8~2.9% of the total AOA, AOB, COM and 16S rRNA sequences, respectively, were selected to construct the trees. For MAGs, AOB-*amoA*, COM-*amoA* or 16S rRNA sequences were extracted and then integrated into the trees. Colors denote sequence origin: black for NCBI database, blue for amplicon data, and red for MAGs linked to nitrifiers. Labels "P" and "W" represent the plastisphere and seawater origins, respectively. For example, the designation of ASV32 (P: 7.6; W:1.4) indicates its proportion of the total sequences-7.6% in the plastisphere and 1.4% in seawater. Bootstrap support values exceeding 50% are noted at branching nodes, based on 1000 replicates. Information of all MAGs is detailed in Fig. 4.

2. I found the description of reciprocal cross-feeding very definitive when it should be quite speculative. E.g. 'We observed a distinctive cooperation of substrate exchange' when as far as I can tell you did not observe transfer of chemicals but are building models based on genes/transcripts alone. Is exchange of NO between nitrifiers a recognised phenomenon as it is for say urea or nitrite? I couldn't tell from the cited references.

Answer:

Thanks for the good comment.

In this study, the postulated "cooperation of substrate exchange" among nitrifiers was based on MAGs-centric metatranscriptomics without a direct quantification of substrate (NO_2^- and/or NO) transfers.

In fact, there is a significant lack of evidence directly supporting NO exchange among nitrifiers as pinpointed by the reviewer. Upon reflection, we agree with the reviewer's points, and thus have now deleted the sentences regarding the exchange of NO among nitrifiers. We have also refined the assertions about this "reciprocal feeding" cooperation.

"Utilizing MAGs-centric metatranscriptomics, we observed a possible cooperation of substrate (NO_2^-) exchange among the COM nitrifier Nitrospira_defluvii (MAG262), AOB Nitrosospira (MAG397) and NOB Nitrobacter (MAG715) in the plastisphere (Figure 6a). Elevated expressions of nirK (NO_2^- reductase) and norBD (NO reductase) genes in Nitrobacter and Nitrosospira, respectively, indicate their capabilities of NO_2^- and NO reduction. Although Nitrospira_defluvii possessed high expression levels of amoAC and nirK genes, the absence of norBD expressions suggests that this COM nitrifier cannot produce N_2O via nitrifier denitrification but could provide NO and NO_2^- . To date, biogeochemical

evidence of NO exchange among nitrifiers remains scant; however, a form of “reciprocal feeding” has been observed, whereby certain AOB, NOB, and COM nitrifiers trade NO₂⁻ to counteract substrate deficiency^{15, 69}. Thus, we hypothesized that these Nitrospira-like COM nitrifiers may provide NO₂⁻ for neighboring AOB (Nitrosospira) to produce N₂O, and in return, can receive NO₂⁻ from neighboring NOB (Nitrobacter) for NO production. The released NO may reshape the cell membrane of nitrifiers to generate symbiont-like aggregates^{15, 73}, which could further enhance collaborative N₂O production in the plastisphere. Such interspecific cooperation among the plastisphere nitrifiers, inferred metatranscriptomically, occurs outside cells to ensure substrate sharing²⁵, likely thereby increasing N₂O emission in the plastisphere. Confirmation through NanoSIMS and isotopic tracing of substrate transfer (NO₂⁻, NO etc.) among nitrifiers is encouraged for future research.” (Lines 394-415).

3. Isotopomer analysis. I would recommend carefully describing what the ranges are for site preference for the different processes, including AOA vs AOB and comammox if available. For example, I think some of the values interpreted as ‘nitrifier denitrification’ overlap with those observed for AOA (Jung et al. 2014).

Answer:

Thanks. We have carefully read the recommended paper (Jung et al. ISME J., 2014), and found that the N₂O-SP levels of AOA were 13~30‰ (please see the table below), which do not overlap with the N₂O-SP for nitrifier denitrification process (SP: -13.6~+1.9‰, mean: -5.9‰, Table S1, SI) used in our study.

Response to comments table 1. "Table 2" of the recommended paper (Jung et al. ISME J., 2014).

Replies and Explanations to Comments

Table 2 Isotopic characteristics of N₂O produced during ammonia oxidation by soil AOA strains

Strain	$\delta^{18}O\text{-N}_2O$ (‰)	$\delta^{15}N^{bulk}\text{-N}_2O$ (‰)	$\delta^{15}N^e$ (‰)	$\delta^{15}N^p$ (‰)	SP (‰)
AOA					
MY1	30.91 (2.06)	-13.53 (2.12)	-1.55 (3.63)	-25.42 (1.68)	23.87 (3.75)
MY2	29.26 (2.47)	-16.96 (1.81)	-6.04 (3.44)	-27.80 (1.33)	21.76 (3.75)
MY3	33.56 (3.27)	-16.49 (2.18)	-3.89 (4.15)	-28.98 (1.40)	25.09 (4.39)
JG1	21.59 (0.34)	-15.32 (0.16)	-5.48 (0.28)	-25.11 (0.03)	19.62 (0.24)
AR	36.11 (0.67)	-12.91 (1.50)	2.07 (1.33)	-27.77 (1.70)	29.83 (0.67)
CS ^a	22.44 (0.72)	-35.54 (0.89)	-28.97 (1.48)	-42.07 (0.32)	13.10 (1.20)
AOB					
N. europaea ATCC 19718	26.58 (0.18)	-19.88 (0.39)	-5.37 (0.58)	-34.31 (0.50)	28.94 (1.01)
Addition of 2 mM nitrite					
MY1	22.22 (3.23)	-17.67 (0.79)	-11.48 (3.10)	-23.86 (2.29)	12.38 (5.22)
N. europaea ATCC 19718	21.85 (0.49)	-27.39 (0.39)	-19.57 (1.37)	-35.14 (0.58)	15.57 (1.95)

It is generally accepted that the ranges of N₂O-SP values for nitrification process (commonly NH₂OH oxidation) are 32.0-38.7‰ (mean value 35.0‰); the ranges for nitrifier denitrification are from -13.6 to +1.9‰ (mean value -5.9‰)¹⁻⁴. We accordingly applied these mean values in our study.

As the reviewer pointed out despite existing several studies on N₂O-SP values of some pure-culture AOA and AOB strains, their large overlap complicates N₂O isotopomer-based source tracking. In our study, we applied a respiration inhibition method by adding penicillin, an inhibitor on bacteria but not on archaea, to identify contributions of AOB and AOA. In addition, due to the paucity of isotopic data on comammox-derived N₂O, its contribution was not assessed.

We have now revised the related parts to avoid confusion.

"..... SP_A is the $SP\text{-N}_2O$ value of NH_2OH oxidation (32.0~38.7‰, average value: 35‰)^{33, 34}, and SP_N is the $SP\text{-N}_2O$ value of nitrifier denitrification (-13.6~1.9‰, average value: -5.9‰)³³" (Line 578).

"Here we omitted N₂O contributions from AOA, AOB and COM due to overlapping $SP\text{-N}_2O$ values between AOA and AOB, and scant isotopic data on COM-derived N₂O. Nevertheless, we applied a respiration inhibitor to discern the AOA and AOB contributions" (Lines 581-583).

General comments:

4. Line 48 – I don't know what this means.

Answer:

Sorry for the confusion. We have now clarified this.

"*The microorganisms found in biofilms encompass diverse microbial kingdoms (i.e., bacteria, fungi, archaea, protists, viruses), contributing markedly to global marine biogeochemical fluxes*" (Lines 46-49).

5. Line 95 – lower levels of what?

Answer:

Clarified.

".....*despite lower levels of N₂O*," (Line 95).

6. Line 103 – Remove '-based'.

Answer:

Deleted.

7. Line 116 – I'm not really sure what 'keystone' means and seems a bit of an unnecessary buzzword. Removing it won't change the meaning. Also, the potential metabolic differences?

Answer:

Thanks. Revised.

".....*we identified active ~~keystone~~ nitrifier communities. Finally, we revealed the potential metabolic differences of*" (Line 116).

8. Line 132 - We sampled the surface seawater of the three sampling sites from June to August, 2022 and 2023. In the original MS this was July and August 2021.

But Figure 1f is the same data as Figure 1 in the original.

Answer:

Sorry for the typo. It should be the 2022 rather than the 2021 in the original submission.

All the experiments regarding plastic types (PE, PS and PVC) were conducted from June to August in 2022 during the original submission. The Figure 1f as mentioned was performed at this time. During the first round of revision, we supplemented a series of experiments regarding biofilm types (plastic, stone, wood and glass) as suggested by the two reviewers at the same months in 2023. Thus, we mentioned "We sampled from June to August, 2022 and 2023" in the manuscript. In the Figure 1, the data of 1a, 1b, 1c and 1d were obtained during the first round of revision (2023), and 1f was obtained during the original submission (2022).

9. Line 133 – Same months in both years? Could be rephrased.

Answer:

Thanks. Revised

".....from June to August of 2022 and 2023." (Line 440).

10. Line 206 – Why was ATU added? Is this not the same assay as experiment 2?

Answer:

Thanks. Yes, this assay (Experiment 3) mirrored the conditions of the Experiment 2. We added ATU, a nitrification inhibitor, in this assay aiming to verify that the decrease of NH_4^+ was mainly due to nitrification rather than biotic assimilation. As we measured nitrification rate via ^{15}N tracing in this study, we have

now omitted ATU-related sections, including "Methods", "Results" and "Supporting Information" sections.

Response to comments figure 6. (now updated in the Supporting Information, **Supplementary Fig. 9**). Changes of NH_4^+ , NO_2^- (a) and NO_3^- (b) concentrations during the 36-h incubation (Experiment 3) in the three types of plastisphere and surrounding seawater.

11. Line 171 – ‘As negligible variation in the nitrification process and nitrifier community in seawater was observed before and after 28 days (Figure S2), we only collected the surface seawater at the end of the incubation.’ This confuses me – are you not talking about the same samples – i.e. day 28? You observed little variation at day 28 so you only collected certain samples at day 28?

Answer:

Sorry. Yes, it is the same sample that was only collected at the day 28 due to the observed little variations of nitrification activity and nitrifier populations between the initial (day 1) and end (day 28) of the incubation (Experiment 1).

In our original submission, we mentioned "we collected the seawater at the end of the *in-situ* incubation". The reviewer 1 commented that "why did authors only collected seawater at one time point? If they authors have good reason to believe the seawater community did not change significantly over the 28-day experiment, please include that rationale and any citation(s)". During the first round of revision, therefore, we conducted additional assays including nitrification rate, N₂O emission and nitrifier community at the initial and end of the 28-d *in-situ* incubation to reply the comment. Meanwhile, we supplemented the sentence "As negligible variation in the nitrification process and nitrifier community in seawater was observed before and after 28 days (Figure S2), we only collected the surface seawater at the end of the incubation" in the manuscript.

To avoid confusion, we have now revised the sentence and moved it to sample collection part.

"It should be noted that given the negligible changes in both nitrification activity and nitrifier community in seawater between the initial and end of the in-situ incubation (Figure S2), surface seawater was only sampled at the day 28." (Lines 475-478).

12. Line 227 – 1 mM

Answer:

Corrected.

"After adding 1 mL 1 mM " (Line 533).

13. Line 314 – red water?

Answer:

Yes. As the mixture of DNA, GB buffer and CsCl solution is colorless and transparent, we used the sterile water with adding several drops of sterile red ink to better distinguish the sample and the water during DNA fractionation. Please see the figure below. To avoid confusion, we have deleted it in the revised manuscript.

Response to comments figure 7. Operations of DNA-SIP assays.

"The gradient mixture was fractionated ~~with red water~~ using an automatic-sampler " (Line 627).

14. Line 319 – remove slightly

Answer:

Deleted.

15. Line 336 – GenBank

Answer:

Corrected.

"..... compared with GenBank and UNITE databases." (Line 649).

16. Line 354 – What PCR assays?

Answer:

The PCR assays were conducted to confirm successful amplification of the concentrated DNA and verify the fragment length. We have clarified it now.

"..... subjected to PCR assays to verify the fragment length." (Line 667).

17. Line 360 – ‘The RNA integrity number ranges from 8.0 to 9.1’. What units?

Answer:

Thanks. The RNA integrity number (RIN) is a quality control test that measures the level of degradation of RNA in the extract on a value scale from 1 to 10 with 10 being the least degraded. This RIN number is obtained directly from Agilent 2100 Bioanalyzer, and thus it does not have a specific unit. The RIN numbers of RNA samples were 8.0~9.1 in this study, indicating a less degradation of extracted RNA.

18. Line 413 – ‘... more so than to plastic.’ This is not quantitative and also cannot be determined from the pictures. I would remove.

Answer:

Deleted.

19. Line 454 – ‘In combination, these findings indicate that plastisphere biofilm exhibited a greater nitrifying potential compared to other biofilms as well as the surrounding seawater, acting as an overlooked nitrifying niche in estuarine ecosystems.’ The data suggest to me that for 2 of the three sites that the biofilms have similar N transformation rates compared with stone. I think a more accurate

description is required.

Answer:

Thanks. Corrected accordingly.

"In combination, these findings indicate that plastisphere exhibited a greater nitrifying potential compared to wood and glass biofilms as well as the surrounding seawater, acting as an overlooked nitrifying niche in estuarine ecosystems. Stone biofilms harbored more nitrifier biomass yet comparable activity to the plastisphere at most sites, further highlighting the distinct niche of this artificial interface." (Lines 168-173).

20. Line 494 – I would describe buoyant density variation rather than fraction numbers.

Answer:

Corrected.

"..... the ¹³C-DNA were observed in buoyant density 1.699-1.702 g mL⁻¹, while they remained 1.680-1.686 g mL⁻¹ in the ¹²C-microcosms in the plastisphere or seawater samples." (Line 207).

"..... the buoyant densities of ¹³C-DNA and ¹²C-DNA were in 1.695-1.699 and 1.687-1.695 g mL⁻¹, respectively, " (Line 210).

"..... are accumulated in buoyant density 1.699-1.702 and 1.680-1.686 g mL⁻¹, respectively" (Figure 3 Caption)

21. Line 497 - "likely implying a poor fractionation and less abundance of AOA". Why would the fractionation be poor when you got good separation for bacteria. Also, is 'bacterial nitrifiers' AOB or Comammox? The primers don't pick up both. Why not to all three groups?

Answer:

Sorry for the confusion.

The statement "..... likely implying a poor fractionation and less abundance of AOA." we mentioned denotes the AOA populations, not AOB. The shift in CsCl buoyant density from the ^{12}C to the ^{13}C genomes for AOA is $<0.012 \text{ g mL}^{-1}$. Compared to AOB ($0.016\sim 0.022 \text{ g mL}^{-1}$), this small change of buoyant density suggests a poor fractionation of AOA, regardless of the plastisphere and seawater.

The "bacterial nitrifiers" refer to AOB, not including comammox bacteria. Owing to the less abundances of comammox populations compared to AOA and AOB in the plastisphere, and undetected in seawater samples, the quantification of COM-*amoA* abundances along with buoyant density was not presented in the previous version.

As the reviewer suggests, we have now corrected "bacterial nitrifiers" to "AOB" and provided the variations of COM-*amoA* along with buoyant densities in the revised manuscript.

Response to comments figure 8. DNA-SIP (now updated in the **main text**, **Fig. 3**). The ^{13}C -DNA (heavy) and the ^{12}C -DNA (light) of AOA, AOB and COM are shown by qPCR of *amoA* across the CsCl buoyant density gradient after the 30-d flush-feeding incubation (Experiment 4).

*"Quantification of AOA-, AOB- and COM-*amoA* gene abundances as a function of CsCl-DNA buoyant density illustrated the labelling of active nitrifiers (Figure 3a)."*

(Line 205).

"For active AOA, the peak buoyant densities of ^{13}C -DNA and ^{12}C -DNA were in 1.695-1.699 and 1.687-1.695 g mL^{-1} , respectively. This minimal change (0.008-0.012 g mL^{-1}) of buoyant density suggested a poor fractionation of AOA, in comparison to AOB, across both plastisphere and seawater (Figure 3a). For COM nitrifiers, they presented markedly lower abundances than AOA and AOB, with minimal fractionation in the plastisphere evidenced by a buoyant density shift from ^{12}C to ^{13}C genomes (0.002~0.009 g mL^{-1}), and were undetected in seawater fractions." (Lines 210-216).

22. Line 498 – What fractions (with BD values) were specifically sequenced?

There seems to be quite a range separating the AOA and bacteria.

Answer:

Thanks. The ^{13}C -labelled AOA, AOB and COM bacteria exhibited similar peak buoyant densities within fractions 9 to 10, ranging from 1.695 to 1.702 g mL^{-1} (as illustrated in above figure). Sequencing was thus conducted on these two fractions.

We have now clarified in the revised manuscript.

"Sequencing of metagenomes, 16S rRNA, AOA-amoA, AOB-amoA, and COM-amoA was subsequently performed on fractions 9-10 of the ^{13}C -DNA samples, characterized by a buoyant density of 1.695~1.702 g mL^{-1} , to explore the composition of the active nitrifiers." (Lines 217-219).

23. Line 500 – No labelling of nxrA or B? Nitrospira or other groups? This contradicts the next paragraph where you describe detection of active Nitrospira and Nitrobacter.

Answer:

Thanks. Yes, the NOB-*nxrA/B* genes were not successfully amplified in all

fractions of ^{12/13}C-DNA samples though we used different primers in this study. Here, the mentioned active NOBs such as *Nitrospira* and *Nitrobacter* were indicated via ¹³C-DNA-based metagenomics and RNA-based metatranscriptomics.

To avoid confusion, we have now revised the description to clarify this.

"*Compositions and abundances of active nitrifiers revealed by ¹³C-DNA-based metagenomics and metatranscriptomics showed that*" (Line 222).

24. Line 521 - followed by *Nitrospira*_sp. (6.5% and 0.12%) and uncultured *Nitrospira*_sp. (3.9% and 0.08%). These names are not informative. Can you use different descriptors?

Answer:

Corrected.

"..... followed by some uncharacterized *Nitrospira* members (3.9-6.5% and 0.08-0.12%)" (Line 241).

25. Line 527 - These are not high but medium quality. Using the Bowers et al. 2017 community standard, high quality is >90% and <5% contamination.

Answer:

Thanks. We have now corrected it throughout the manuscript.

"..... were recovered as medium- (completeness:>75%, contamination:<10%) and high-quality MAGs (completeness:>90%, contamination:<5%)" (Line 247).

"Five medium- and high-quality MAGs" (Line 251).

"Through the analysis of medium- and high-quality MAGs," (Line 337).

"A total of five medium- and high-quality MAGs....." (Figure 4 Caption).

26. Line 673 – ‘We observed a distinctive cooperation of substrate exchange

(NO₂⁻ and NO) among the COM?. I am not sure how you observed this. Is this just speculation from the metaG/T data?

Answer:

Yes, the postulated "cooperation of substrate exchange" among nitrifiers was based on MAGs-centric metatranscriptomics in our study. We have now recognized this concern, and moderated the statements accordingly. The detailed responses are provided in the Response to Comment 2.

27. Line 686 – I am uncertain whether the Daims et al. models (cited) describe sharing of NO?

Answer:

Thanks. We have now revised the assertions regarding the substrate cooperation among nitrifiers. Please see the Response to Comment 2 for details.

Reviewer #3 Comments:

28. I appreciate the opportunity to review this interesting article. I think the authors have properly addressed and resolved all of the questions and critiques raised by the two reviewers.

Answer:

We sincerely thank the reviewer for examining our manuscript and providing positive feedback.

Reviewer #4 Comments:

29. The plastisphere, which comprises the microbial community on plastic debris, is a growing emphasis in environmental sciences as the occurrence of plastic

debris is present in aquatic and terrestrial systems. Less known is how the microbial communities on the surface of plastics facilitate biogeochemical processes. The manuscript that was submitted and reviewed provides insight into nitrogen cycle processes that occur in the plastisphere. In their revision, they conducted several additional experiments to compare nitrogen cycle processes on plastics to those of other substrates and demonstrate the varying degree to how nitrogen fixation and cycling differs between plastics and other substrates (glass, wood, stone).

The submitted work is a valuable contribution to the field as plastic research is garnering research attention; there has been less of a focus on biogeochemical processes governed by microbial communities. As microbes colonize plastics, their function may differ for various reasons. This submitted paper starts to uncover how processes may vary, and if better understood, we can understand how specific biogeochemical cycling may differ in areas where pollution is more significant.

The authors have done a great job revising the document in response to reviewers. They have taken the extra step to include experiments supporting their findings while addressing the uncertainty described in their initial review. The author's thorough review leaves little flaws to take away from their work and conclusions. The methodology is sound, from their experimental design to their analysis and molecular approaches, while informing their work from previous lab work and expanding to answer relevant questions. This work's findings are important and provide insight for future work in other matrices, like freshwater ecosystems.

Answer:

We greatly appreciate the reviewer for the positive assessment on our manuscript.

References

1. Yu, L.; Harris, E.; Lewicka-Szczebak, D.; Barthel, M.; Blomberg, M. R. A.; Harris, S. J.; Johnson, M. S.; Lehmann, M. F.; Liisberg, J.; Müller, C.; Ostrom, N. E.; Six, J.; Toyoda, S.; Yoshida, N.; Mohn, J., What can we learn from N₂O isotope data? – Analytics, processes and modelling. *Rap. Commun. Mass Spectrom.* **34**, (20), e8858 (2020)
2. Toyoda, S.; Yoshida, N.; Koba, K., Isotopocule analysis of biologically produced nitrous oxide in various environments. *Mass Spectrom. Rev.* **36**, (2), 135-160 (2017)
3. Sutka, R. L.; Ostrom, N. E.; Ostrom, P. H.; Breznak, J. A.; Gandhi, H.; Pitt, A. J.; Li, F., Distinguishing nitrous oxide production from nitrification and denitrification on the basis of isotopomer abundances. *Appl. Environ. Microb.* **72**, (1), 638-644 (2006)
4. Frame, C. H.; Casciotti, K. L., Biogeochemical controls and isotopic signatures of nitrous oxide production by a marine ammonia-oxidizing bacterium. *Biogeosciences* **7**, (9), 2695-2709 (2010)

Reviewers' comments:

Reviewer #5 (Remarks to the Author):

I was initially delighted to read this manuscript but have been strongly disappointed. The topic is novel and interesting, and the introduction reasonably sets the context (even if more niche specialisation of nitrifiers in estuarine environments would have been appreciated). Then, I started to read the method sections (which I rarely do, but there were 4 experiments mentioned, so I wanted to understand the design), but this section did not provide enough details or showed some flaws in the design (e.g. SIP interpretation, amoA phylogeny, MAG annotation, RNA replication). When reading the results, I discovered some issues which are, in my opinion, unacceptable to read in a review for Nat Com. One of my main concerns is the differing results obtained in the SIP experiment and the in-situ environment experiments. The SIP had supplementation of ammonia, which led to the unique growth of AOB (with absence of AOA and COM growth in the SIP experiment (see below)) but AOA are present in situ (see Fig 2f and Fig S4). In addition, I have a series of concerns listed below (chronologically and not by order of importance):

Methods:

- L491: Unclear which biofilm group
- L496: "The higher NH₄⁺ level was selected as the initial concentration mainly aiming at comparatively assessing the potential nitrifying capacity of the biofilms and surrounding seawater". However, the study uses the SIP to understand the active nitrifiers in the plastisphere (see L. 538 "This is because we aim to culture and enrich ¹³C-labelled DNA in nitrifiers, and thus to distinguish active nitrifiers between the plastisphere and seawater."). However, the addition of NH₄ would favor AOB versus AOA, not representing natural estuarine conditions and providing different conditions than in situ .
- L506: no DNA extraction protocol is described but plastic, stone, glass and water would be very different!
- L508: no concentration given for penicillin used
- L637: bacterial or AOB?
- L664: "the pooled ¹³C-DNA" – what was pooled? It seems that it was likely not the fractions but the samples, removing all replication for both DNA and RNA approaches. This is a serious issue.
- L688: states MAGs with completeness ≤75% but later in results 90% is mentioned
- L702: more details of markers and models selected for FastTree. Same for the single-gene trees presented

Results:

- Fig 1: I am confused by the nitrification rate being the sum of ammonia and nitrite oxidation rates. They should be plotted separately rather than on a single bar
- Fig S3: I could not find any details about the method of DNA extraction used. How much material? Which lysis approach? Which kit used? Also, the unit of 16s rRNA gene abundance is unusual (copies . L-1), especially for solid material (usually per g-1 or ml-1 for liquid). How can we be sure that similar amounts of material were compared?
- Fig S6: I am confused here by these concentrations. It says in the legend "in the biofilms and surrounding seawater" but how could it be really measured in the biofilm itself compared to the seawater.... Terminology is really frustrating. There is also absence of stoichiometry in some

treatments (e.g. wood) between NH₄ decrease and NO₃ increase, but this was not explained. The figures for NH₄ and NO₂ are too crowded to see anything.

- Fig S9: Again, absence of stoichiometry in all treatments for the plastisphere between NH₄ decrease and NO₃ increase. Is some adsorption happening here?

- L193: this was not the case in situ (fig 1f) suggesting that in situ and in vitro are difficult to compare.

- SIP experiment and Fig S10: I have some concerns about this experiment. The x-axis in a SIP graph usually represents the percentage of DNA in each fraction compared to the total amount of DNA (sum of all the fractions). May be here the “the maximum amoA quantities” represent the total amount? If so, it would make sense for AOB-amoA, where 80% of the community have shifted following labelling incorporation. However, the graphs have some issues for AOA and COM as the ratio are very low (10E-08 for AOA, corresponding to noise level as less than 0.00001%). Same is true for COM with either 10E-14 or 10E-014, which is unclear. In any case, for both AOA and COM where is the DNA? This is very well explained in the seawater, where a total absence of peak shows that there are no COM DNA in my opinion. The same is true for COM-plastisphere and a similar story for AOA. In all those labelled and unlabelled treatments, there are no significant peaks, so no DNA. The authors did not interpret the results like this and mentioned that activity existed when there was the shift of BD (see L. 212 and 215). However, the extent of the shift is in fact not dependent on the percentage of incorporation but on the “weight” of the DNA following the labelling, hence related to the GC content of the DNA. In my opinion, these results demonstrate that AOB were growing but not COM or AOA, which are both nearly absent. The authors did not interpret these results like that (see L237: “Notably, the plastisphere surprisingly harbored more abundant COM nitrifiers than the seawater (Fig. 3b), suggesting the plastisphere likely being a hotspot for COM process.” Finally, I may have missed it, but why NOB were not included in the SIP analysis? This represents a major flaws as the rest of the manuscript aims to understand those active nitrifiers in the plastisphere.

- Fig 3b: The methanosarcina clade is not known to have an amoA gene, so I would need more evidence to believe this result. Such incorporation falsifies the percentage quite massively.

- Fig 3b, no details are given on the bioinformatics of the sequencing, while a HiSeq 250bp can not recover the length of the genes (600bp being too long to recover the full gene and QIIME assembly likely requires some overlap of the reads?). The authors only state QIIME pipeline, but this is not explanatory enough.

- Fig 3b: The SIP showed growth of AOB. The amplicon sequencing showed growth of Nitrosomonas but no growth of Nitrosopira as not a single amplicon was affiliated to Nitrosopira. However, the MAG reconstructed a Nitrosopira MAG. This is not coherent.

- L227: A proper statistical comparison is required - looking at the proportion of some sequences on a tree is not meaningful. A community representation on a PCoA or similar plot followed by PERMANOVA or similar would be better. Or Fig S12 could be brought in.

- MAG analysis: L276: These module numbers are the first thing I checked for the genome analysis and M00529 is denitrification not nitrite oxidation. I lost all confidence in this manuscript and stopped my review there, I am sorry.

Replies and Explanations to Comments

We thank the reviewer for the comments. We have now revised the manuscript thoroughly and highlighted the changes in track-change mode. All questions raised by the reviewer have been answered point-by-point below.

Reviewer #5 Comments:

I was initially delighted to read this manuscript but have been strongly disappointed. The topic is novel and interesting, and the introduction reasonably sets the context (even if more niche specialization of nitrifiers in estuarine environments would have been appreciated). Then, I started to read the method sections (which I rarely do, but there were 4 experiments mentioned, so I wanted to understand the design), but this section did not provide enough details or showed some flaws in the design (e.g. SIP interpretation, amoA phylogeny, MAG annotation, RNA replication). When reading the results, I discovered some issues which are, in my opinion, unacceptable to read in a review for Nat Com.

1. One of my main concerns is the differing results obtained in the SIP experiment and the in-situ environment experiments. The SIP had supplementation of NH_4^+ , which led to the unique growth of AOB (with absence of AOA and COM growth in the SIP experiment (see below)) but AOA are present in situ (see Fig 2f and Fig S4). In addition, I have a series of concerns listed below (chronologically and not by order of importance).

Answer:

Thanks for the comments. We have noted the reviewer's concern regarding the high NH_4^+ levels in our SIP assays, favoring AOB over AOA and COM, which potentially results in lower AOA abundances and absence of COM nitrifiers.

However, our study did detect AOA abundances under *in-situ* conditions. This was viewed as inconsistent with the SIP results by the reviewer. In fact, this seeming discrepancy reflects a progressive research approach rather than inconsistency, as explained below.

(1) ***In-situ* environments (AOB dominance)**. Following *in-situ* or lab incubations, our measurements of AOA, AOB and COM abundances revealed AOB dominance across all samples and sites, except for the wood biofilms at the sampling site 3 (Nanning). AOB abundances generally surpassed AOA by 1-2 orders of magnitude, with COM undetected (please see Supplementary Fig. S4). The penicillin inhibition experiments further confirmed the dominant role of AOB in nitrification and N₂O emission, despite some contributions from AOA.

(2) **SIP assays (focusing on AOB)**. Building upon the above findings, we further concentrated our efforts primarily on AOB, investigating differences in the active communities and metabolic mechanisms between sessile (plastisphere) and free-living (bulk seawater) lifestyles. Thus, the SIP assays utilized NH₄⁺ supplementation, which likely favored AOB growth over AOA and COM. However, this does not preclude the growth of AOA and COM under these conditions, albeit with less competitive advantages compared to AOB, as evidenced by the detected abundances of AOA and COM after the SIP incubations. Moreover, NH₄⁺ supplementation in SIP incubations to enrich ¹³C-DNA has also been employed in many previous studies¹⁻⁴.

Overall, the SIP assays in this study represent a further exploration of the metabolic mechanisms of active nitrifiers, primarily focusing on nitrifying bacteria, following *in-situ* or lab incubations. Therefore, rather than being inconsistent, it reflects a progressive investigation of nitrification mechanisms. Nonetheless, we

have included statements to address the limitations.

"..... our SIP assays utilized supplementary high levels of NH₄⁺ to enrich ¹³C-labelled DNA in nitrifiers, potentially biasing towards AOB and overlooking AOA. While our mechanistic study reveals AOB metabolic differences between sessile (plastisphere) and free-living (seawater) environments, future research should explore AOA dynamics and their metabolic adaptations for a comprehensive nitrifying blueprint in the plastisphere." (Line 425).

2. - L491: Unclear which biofilm group.

Answer:

The term "biofilm group" here refers to all biofilm types in this study, including those on plastic, stone, wood, and glass surfaces. We have now clarified this.

"..... was then used for all biofilm groups" (Line 498).

3. - L496: "The higher NH₄⁺ level was selected as the initial concentration mainly aiming at comparatively assessing the potential nitrifying capacity of the biofilms and surrounding seawater". However, the study uses the SIP to understand the active nitrifiers in the plastisphere (see L. 538 "This is because we aim to culture and enrich ¹³C-labelled DNA in nitrifiers, and thus to distinguish active nitrifiers between the plastisphere and seawater."). However, the addition of NH₄ would favor AOB versus AOA, not representing natural estuarine conditions and providing different conditions than in situ.

Answer:

Thanks. Please see the Responses to Comments 1 for detailed responses.

4. - L506: no DNA extraction protocol is described but plastic, stone, glass, and

water would be very different!

- Fig S3: I could not find any details about the method of DNA extraction used. How much material? Which lysis approach? Which kit used? Also, the unit of 16s rRNA gene abundance is unusual (copies L⁻¹), especially for solid material (usually per g⁻¹ or ml⁻¹ for liquid). How can we be sure that similar amounts of material were compared?

Answer:

Thanks for the comments and added accordingly.

"To compare the biomass and microbial communities across different biofilms, we standardized the sample preparation by processing 3 g of each biofilm material, including plastic debris, wood debris, stone debris, and glass balls, after incubations. The DNA was extracted using a FastDNA™ Kit (MP, Biomedicals, USA) according to the manufacturer instructions. The detailed procedures are as follows:

(a) We prepared 3 g of each material known to form biofilms.

(b) These materials were transferred into 150-mL sterile serum bottles filled with 30 mL of sterile water.

(c) The bottles were shaken at 15°C and 200 rpm for 6 hours.

(d) Biofilms were detached from the materials using a knife as completely as possible, resulting in a 30-mL biofilm suspension.

(e) From this suspension, 1.2 mL was taken and added to the sample tubes of the MP kit, and DNA extraction proceeded according to the kit guidelines." (in SI)

Consequently, the unit of gene abundances in Fig. S3 and S4 was expressed as copies per liter. For the seawater group, water was filtered through 0.22-µm filters, and these filters were then processed for DNA extraction with the same kit, as described in the *DNA fractionation* section of the main text. We have now added these procedural details to the Supplementary Information and revised the main

text.

5. - L508: no concentration given for penicillin used.

Answer:

Sorry. It is the 100 μM of penicillin.

"..... *with and without adding 100 μM of penicillin*" (Line 165).

"..... *with adding 100 μM of penicillin*" (Line 516).

6. - L637: bacterial or AOB?

Answer:

Yes, "bacterial-*amoA*" means AOB. We have now clarified this.

"..... *bacterial-amoA (AOB), archaeal-amoA (AOA) and Comammox-amoA genes (COM)*" (Line 647).

7. - L664: “the pooled ^{13}C -DNA” – what was pooled? It seems that it was likely not the fractions but the samples, removing all replication for both DNA and RNA approaches. This is a serious issue.

Answer:

Sorry for the confusion. The term "pooled ^{13}C -DNA" refers specifically to the combination of fractions 9 and 10 within each individual sample after DNA fractionation, not a mixture of ^{13}C -DNA from the three replicates. Each treatment, including both plastisphere and seawater, was conducted in triplicates. DNA from these individual replicates underwent separate quality control and sequencing.

To avoid confusion, we have now clarified this in the revised manuscript.

" *The pooled ^{13}C -DNA (consisting of fractions 9 and 10)*" (Line 685).

8. - L688: states MAGs with completeness $\leq 75\%$ but later in results 90% is mentioned.

Answer:

Yes, we specified that only MAGs with completeness $\geq 75\%$ and contamination $\leq 10\%$ were used for analyses in this study. Among these, five MAGs related to nitrification were the focus of this study, and their completeness all exceeded 90%. Thus, we mentioned the "90%" in Results section.

9. - L702: more details of markers and models selected for FastTree. Same for the single-gene trees presented.

Answer:

Thanks. Added.

"Phylogenetic analysis of all medium- and high-quality MAGs was conducted with FastTree (version 2.1.10) based on 120 bacterial and 122 archaeal marker genes to evaluate the phylogenetic placement and relative evolutionary divergence (RED) of genomes within the GTDB reference tree. Phylogenomic trees were inferred with WAG and GAMMA models and 1000 bootstraps, based on alignments of these marker genes, and visualized using iTOL(v4)." (Line 724).

"Phylogenetic analysis of nitrifiers was conducted using AOA-amoA, AOB-amoA, COM-amoA, and NOB-16S rRNA sequencing data. The 5, 14, 11 and 7 typical ASVs contributing to 30.4~42.9%, 60.3~77.3%, 73% and 3.0~3.8% of the total AOA, AOB, COM and 16S rRNA sequences, respectively, were selected to construct the trees. Specially for NOB, we designated ASVs from 16S rRNA sequences affiliated with Nitrospira, Nitrobacter and Nitrotoga as candidate NOB ASVs for phylogenetic reconstruction. Homologous sequences from NCBI were used for constructing maximum likelihood phylogenetic trees with the Kimura 2-parameter model and

1000 bootstraps in MEGA7." (Line 662).

10. - Fig 1: I am confused by the nitrification rate being the sum of ammonia and nitrite oxidation rates. They should be plotted separately rather than on a single bar.

Answer:

Revised.

Response to comments figure 1. (now updated in **main text, Fig. 1**). Sampling and incubation sites, biofilm type- and plastic type-based materials, and nitrification rates. (a) Sampling and *in-situ* incubation sites, including site 1: Xiamen (XM) in Fujian Province (118°11'E, 24°57'N), site 2: Yantai (YT) in Shandong province (121°47'E, 37°46'N), and site 3: Nanning (NN) in Guangxi Province (108°27'E, 22°48'N). (b) Biofilm type materials include plastic bags, glass balls, stone debris, and wood debris; plastic type materials include PE, PS, and PVC. (c)-(f) Ammonia and nitrite oxidation rates in different biofilms, plastic types and surrounding seawater across the three sampling sites.

11. - Fig S6: I am confused here by these concentrations. It says in the legend “in the biofilms and surrounding seawater” but how could it be really measured in the biofilm itself compared to the seawater.... Terminology is really frustrating. There is also absence of stoichiometry in some treatments (e.g. wood) between NH_4 decrease and NO_3 increase, but this was not explained. The figures for NH_4 and NO_2 are too crowded to see anything.

Answer:

Thanks. (1) Concentrations of N speciation (NH_4^+ , NO_2^- , and NO_3^-) in the biofilm groups were measured in water phase. Please see the schematic diagram below. The sterile seawater was achieved by filtration with a 0.22- μm polycarbonate membrane. We have mentioned “the sterile seawater from the three sites was achieved by filtration with a 0.22- μm polycarbonate membrane and this water was then used for all biofilm groups to” in the previous version of the manuscript.

Response to comments figure 2. Schematic diagram illustrating the use of sterile seawater across all biofilm groups and *in-situ* seawater in the Seawater group for measuring concentrations of nitrogen species.

(2) Regarding the nitrification stoichiometry, the observed imbalance between the decrease in NH_4^+ and increase in NO_3^- is likely attributed to microbial assimilation and biofilm adsorption of NH_4^+ . In a preliminary experiment, we supplemented the system with the nitrification inhibitor allylthiourea (ATU, 60 μM), which specifically targets ammonia monooxygenase. This addition aimed to elucidate that the variations in NH_4^+ , NO_2^- and NO_3^- concentrations mainly

stemmed from the nitrification process rather than from assimilation or adsorption (please see the figure below). Despite the inhibition on nitrification by ATU, we still observed a reduction in NH_4^+ , particularly in biofilm groups, indicating likely microbial assimilation and biofilm adsorption of NH_4^+ . As the previous reviewers stated "as the authors measured nitrification rate using ^{15}N tracing, why did they still include ATU assays?", we omitted this part in the last revised manuscript.

We have clarified this in the revised manuscript.

"From the perspective of nitrification stoichiometry, the decrease in NH_4^+ and increase in NO_3^- appeared to be unbalanced. This is likely due to microbial assimilation and biofilm adsorption of NH_4^+ ." (in SI).

Response to comments figure 3. Changes of NH_4^+ and NO_3^- concentrations during the 36-h incubation in the plastisphere and surrounding seawater with adding ATU.

(3) We have now revised the figure to make clearer.

Response to comments figure 4. (now updated in the **Supplementary Information, Fig. S6**) Changes of NH_4^+ , NO_2^- and NO_3^- concentrations during the 36-h incubation (Experiment 2) in the biofilm groups and the surrounding seawater group. (a), (c) and (e) are the variations of NH_4^+ and NO_2^- concentrations at XM, YT and NN sites, respectively. (b), (d) and (f) are the variations of NO_3^- at XM, YT and NN sites, respectively.

12. - Fig S9: Again, absence of stoichiometry in all treatments for the plastsphere between NH_4 decrease and NO_3 increase. Is some adsorption happening here?

Answer:

Thanks. The disparity between the decrease in NH_4^+ and the increase in NO_3^- may be attributed to microbial assimilation and biofilm adsorption of NH_4^+ . Please see the above responses and explanations for details.

13. - L193: this was not the case in situ (fig 1f) suggesting that in situ and in vitro are difficult to compare.

Answer:

Indeed, in this study, the Experiment 1 (*in-situ* incubations) was conducted under field conditions, while the Experiments 2, 3, and 4 were carried out in the lab, as outlined in the previous version of the manuscript. This differentiation in experimental settings is necessary because accurately measuring nitrification rates and nitrogen transformations is challenging under *in-situ* conditions. We agree that it is difficult to compare nitrification performance between *in-situ* and in vitro conditions. To address this, we have now discussed the limitations in the revised manuscript.

"Secondly, our exploration of nitrifying potential and metabolic mechanisms within the plastisphere predominantly conducted under laboratory conditions. While insightful, expanding these findings to estuarine ecosystems requires large-scale, in-situ monitoring. Such endeavors are essential for comprehensively grasping N cycling nuances, offering a holistic understanding of nitrification's ecological significance in estuarine plastisphere." (Line 431).

14. - SIP experiment and Fig S10: I have some concerns about this experiment. The x-axis in a SIP graph usually represents the percentage of DNA in each fraction compared to the total amount of DNA (sum of all the fractions). May be here the “the maximum amoA quantities” represent the total amount? If so, it would make

sense for AOB-*amoA*, where 80% of the community have shifted following labelling incorporation. However, the graphs have some issues for AOA and COM as the ratios are very low ($10E-08$ for AOA, corresponding to noise level as less than 0.00001%). Same is true for COM with either $10E-14$ or $10E-014$, which is unclear. In any case, for both AOA and COM where is the DNA? This is very well explained in the seawater, where a total absence of peak shows that there is no COM DNA in my opinion. The same is true for COM-plastisphere and a similar story for AOA. In all those labelled and unlabeled treatments, there are no significant peaks, so no DNA. The authors did not interpret the results like this and mentioned that activity existed when there was the shift of BD (see L. 212 and 215). However, the extent of the shift is in fact not dependent on the percentage of incorporation but on the “weight” of the DNA following the labelling, hence related to the GC content of the DNA. In my opinion, these results demonstrate that AOB were growing but not COM or AOA, which are both nearly absent. The authors did not interpret these results like that (see L237: “Notably, the plastisphere surprisingly harbored more abundant COM nitrifiers than the seawater (Fig. 3b), suggesting the plastisphere likely being a hotspot for COM process.”

Answer:

Thanks for the comments.

(1) Yes, "the maximum *amoA* quantities" represents the total *amoA* amounts.

(2) As noted by the reviewer, AOA and COM displayed lower abundances compared to the AOB, with COM not amplified successfully in the Seawater group. Despite the reduced presence of AOA and COM, AOA was detectable in both the plastisphere and seawater groups via qPCR assays, and COM nitrifiers were identified in the plastisphere relative to negative controls. More importantly, ^{13}C -DNA was primarily enriched in fractions 9 and 10, regardless of AOA, AOB, and COM

communities, which were then pooled for sequencing, ensuring that the DNA analyzed retained information on AOA and COM. In addition, as previously addressed in Responses to Comments 1, our SIP assays primarily focused on AOB, exploring differences in the active community and metabolic mechanisms between sessile (plastisphere) and free-living (bulk seawater) lifestyles. Thus, the design and results of SIP assays are feasible. We have now revised this part to make it clearer.

"To identify the active nitrifiers, mainly nitrifying bacteria, in the plastisphere and surrounding seawater" (Line 201).

(3) While we were unable to target COM nitrifiers in the seawater using qPCR, amplicon sequencing or ¹³C-DNA based metagenomics, but we did detect them in the plastisphere. Consequently, we mentioned "... suggesting the plastisphere likely being a hotspot for COM process" in the main text. To avoid confusion, we have now deleted this statement.

15. - Finally, I may have missed it, but why NOB were not included in the SIP analysis? This represents a major flaw as the rest of the manuscript aims to understand those active nitrifiers in the plastisphere.

Answer:

Thanks. Although we employed various primers reported in previous studies, abundance of the NOB-*nxrA/B* gene was not detected in this study, likely due to the high specificity of these primers. We have acknowledged this point in the previous version of the manuscript and the Response letter.

While the inability to include NOB in SIP analysis is regrettable, it should not be seen as a flaw but rather as a specific challenge related to the methodologies available. To mitigate this and still explore active NOB community, we focused on metagenomics and metatranscriptomics, specifically analyzing sequencing data

from fractions 9 and 10, which allowed us to investigate the presence and activity of NOB at the genetic and transcriptomic levels. These efforts were aimed at understanding the potential roles and contributions of NOB within the plastisphere.

16. - Fig 3b: (1) The methanosarcina clade is not known to have an *amoA* gene, so I would need more evidence to believe this result. (2) No details are given on the bioinformatics of the sequencing, while a HiSeq 250bp can not recover the length of the genes (600bp being too long to recover the full gene and QIIME assembly likely requires some overlap of the reads?). The authors only state QIIME pipeline, but this is not explanatory enough. (3) The SIP showed growth of AOB. The amplicon sequencing showed growth of Nitrosomonas but no growth of Nitrosopira as not a single amplicon was affiliated to Nitrosopira. However, the MAG reconstructed a Nitrosopira MAG. This is not coherent.

Answer:

Thanks. (1) Corrected. Please see the figure below.

Response to comments figure 5. (now updated in **main text, Fig. 3c**). Phylogenetic tree of AOA *amoA* gene (maximum-likelihood-based). The 5 typical ASVs contributing to 30.4~42.9% of the total AOA sequences were used to construct the tree at 97% similarity. Labels "P" and "W" represent the plastisphere and seawater origins, respectively. For example, the designation of ASV32 (P: 7.6; W:1.4) indicates its proportion of the total AOA *amoA* sequences-7.6% in the plastisphere and 1.4% in seawater. Bootstrap support values exceeding 50% are displayed at branching nodes, based on 1000 replicates.

(2) Added.

"..... Purified libraries containing 16S rRNA, AOB-amoA, and COM-amoA genes were sequenced on the Illumina MiSeq PE300 platform (Illumina, San Diego, CA), while the AOA-amoA gene was sequenced on the PacBio Sequel Ite System (Pacific Biosciences, CA, USA)." (Line 657)

(3) We did detect the genus *Nitrosospira* using bacterial *amoA* amplicon sequencing, but its relative abundance was markedly low, constituting less than 2% of the total AOB sequences identified. Due to its minimal representation, initial presentations excluded these sequences for brevity.

To prevent any misunderstanding, we have now included sequences of two ASVs affiliated with *Nitrosospira* in the AOB phylogenetic tree. Please see the figure below.

Response to comments figure 5. (now updated in **main text, Fig. 3c**). Phylogenetic tree of AOB *amoA* gene (maximum-likelihood-based). The 14 typical ASVs contributing to 60.3~77.3% of the total AOB sequences were selected based on AOB-associated MAGs (MAG876 and 397) and used to construct the tree at 97% similarity.

17. - L227: A proper statistical comparison is required-looking at the proportion of some sequences on a tree is not meaningful. A community representation on a PCoA or similar plot followed by PERMANOVA or similar would be better. Or Fig S12 could be brought in.

Answer:

Thanks, and added accordingly.

Response to comments figure 7. (now added in **Supplementary Information, Fig. S11**). β -diversity of bacterial community (16S rRNA), AOA community (archaeal-*amoA*), and AOB community (bacterial-*amoA*). Principal coordinate analysis (PCoA) along with permutational multivariate analysis of variance (PERMANOVA) based on Bray-Curtis distances were conducted. As COM community was only detected in the plastisphere, we did not present the data in this figure.

18. - MAG analysis: L276: These module numbers are the first thing I checked for the genome analysis and M00529 is denitrification not nitrite oxidation.

Answer:

Thanks. In analysis of metagenomics and metatranscriptomics, we identified that one of the annotated KEGG modules for some putative NOB, such as *Nitrospira* and *Nitrobacter*, was M00529 (nitrate reduction, the reverse step of nitrite

oxidation). Thus, we used M00529 as representative of nitrite oxidation in this study. To avoid any misunderstanding, we have now removed the M00529 from the revised manuscript.

References

1. Gulay, A.; Fowler, S. J.; Tatari, K.; Thamdrup, B.; Albrechtsen, H.-J.; Abu Al-Soud, W.; Sorensen, S. J.; Smets, B. F., DNA- and RNA-SIP Reveal *Nitrospira* spp. as Key Drivers of Nitrification in Groundwater-Fed Biofilters. *Mbio* **10**, (6), (2019)
2. Sun, D.; Tang, X.; Li, J.; Liu, M.; Hou, L.; Yin, G.; Chen, C.; Zhao, Q.; Klumper, U.; Han, P., Chlorate as a comammox *Nitrospira* specific inhibitor reveals nitrification and N₂O production activity in coastal wetland. *Soil Biology & Biochemistry* **173**, (2022)
3. Sun, X.; Zhao, J.; Zhou, X.; Bei, Q.; Xia, W.; Zhao, B.; Zhang, J.; Jia, Z., Salt tolerance-based niche differentiation of soil ammonia oxidizers. *Isme Journal* **16**, (2), 412-422 (2022)
4. Zhang, Q.; Li, Y.; He, Y.; Liu, H.; Dumont, M. G.; Brookes, P. C.; Xu, J., *Nitrosospira* cluster 3-like bacterial ammonia oxidizers and *Nitrospira*-like nitrite oxidizers dominate nitrification activity in acidic terrace paddy soils. *Soil Biology & Biochemistry* **131**, 229-237 (2019)

REVIEWER COMMENTS

Reviewer #5 (Remarks to the Author):

I would like to thank the authors for clarifying some of my concerns.

I still have some questions regarding two points:

Point 14: The authors clarified the x-axis of the Fig3a as being “total amoA amounts” and it currently is written as “Ratio of maximum amoA quantities”. The legend indicates that “The results are normalized using the ratio of amoA copy number in each DNA fraction to the maximum amoA copy number.” This is confusing and it would be easier to represent the abundance in each fraction as the proportion of the sum of abundance in all the fractions. Using such representation would have the x-axis between 0 and 100%, and one could compare easily between the 12C or 13C treatments, even for AOA and COM. In fact, I still do not understand the current plotting and it still seems that AOA and COM are not growing (with the little detected peaks representing more “artefacts” than real incorporation).

The authors explained in their rebuttal that SIP mainly focused at APOB, so why presenting AOA and COM results in the main figure – it confuses the readers.

Point 16: The authors named the sequencing approach but did they include all the bioinformatics pipelines related to the cleaning and assembly of the reads? Also, how good is PacBio compared the MiSeq for amoA sequencing (several papers have assembled amoA using MiSeq but I am not aware of any PacBio sequencing yet). All the scripts should also be made available with a GitHub link or similar. This is also true for the metagenomics and metatranscriptomics pipelines used.

Replies and Explanations to Comments

We thank the reviewer for further valuable comments. We have now clarified and revised the manuscript thoroughly and highlighted the changes in track-change mode. Two questions raised are explicitly answered point-by-point below.

Reviewer #5 Comments:

I would like to thank the authors for clarifying some of my concerns. I still have some questions regarding two points:

1. Point 14: The authors clarified the x-axis of the Fig3a as being “total amoA amounts” and it currently is written as “Ratio of maximum amoA quantities”. The legend indicates that “The results are normalized using the ratio of amoA copy number in each DNA fraction to the maximum amoA copy number.” (1) This is confusing and it would be easier to represent the abundance in each fraction as the proportion of the sum of abundance in all the fractions. Using such representation would have the x-axis between 0 and 100%, and one could compare easily between the 12C or 13C treatments, even for AOA and COM. (2) In fact, I still do not understand the current plotting and it still seems that AOA and COM are not growing (with the little detected peaks representing more “artefacts” than real incorporation). The authors explained in their rebuttal that SIP mainly focused at AOB, so (3) why presenting AOA and COM results in the main figure – it confuses the readers.

Answer:

We greatly thank the reviewer for the positive feedback and valuable suggestions. We have now addressed the reviewer’s concerns regarding AOA and COM abundances in each fraction and their representation on the x-axis of Figure

3a, as detailed below.

(1) The term "maximum" in the text represents the meaning "total", the sum of AOA-*amoA*, AOB-*amoA*, and COM-*amoA* amounts in the plastisphere or seawater samples. We have now corrected the word in the revised manuscript accordingly.

We have also incorporated the reviewer's suggestion to use the ratios of AOA-, AOB-, or COM-*amoA* quantity in each fraction relative to the total corresponding *amoA* quantities across all fractions in each sample, rather than the sum of AOA-*amoA*, AOB-*amoA*, and COM-*amoA* amounts. The detailed calculation involves determining the proportion of each fraction as follows: (AOA/AOB/COM-*amoA* quantity in each fraction) / (total AOA/AOB/COM-*amoA* quantities of all fractions in each sample). Accordingly, we have re-analyzed and re-plotted Figure 3a (please see the revised figures below).

Response to comment figure 1 (now updated in the main text, **figure 3a** and **SI**). The abundances of the total AOA-, AOB-, and COM-*amoA* genes after the ¹²C/¹³C-labelled incubations are shown. The ¹³C-DNA and the ¹²C-DNA of AOA and AOB are shown by qPCR of *amoA* across the CsCl buoyant density gradient in the DNA-SIP assays. The results are normalized using the ratio of AOA- or AOB-*amoA* copy number in each DNA fraction to the

total AOA- or AOB-*amoA* copy numbers of all fractions in each sample.

(2) Based on the revised figure, we observed that despite the lower abundances of AOA-*amoA* in the plastisphere and seawater incubations compared to AOB, AOA-*amoA* levels after the SIP incubations were an order of magnitude higher than its abundance under *in-situ* conditions, suggesting a slight AOA growth. In addition, the fractionation of AOA-*amoA* between ¹²C-DNA and ¹³C-DNA remained poor, consistent with previous results. For COM-*amoA*, however, the abundances were very low. After careful consideration, we agree with the reviewer's point that COM nitrifiers may not grow in SIP incubations. Thus, we have now deleted the figure showing COM-*amoA* abundances across the CsCl buoyant density gradient and revised the manuscript accordingly.

"AOB abundances were the highest, followed by AOA (Fig. 3a). COM nitrifiers presented markedly lower abundances than both AOA and AOB." (Line 206).

*"The ¹³C-DNA (heavy) and the ¹²C-DNA (light) of AOB are shown by qPCR of *amoA* across the CsCl buoyant density gradient after the 30-d flush-feeding incubation (Experiment 4). The results are normalized using the ratio of AOB-*amoA* copy number in each DNA fraction to the total AOB-*amoA* copy numbers of all fractions in each sample." (Line 1042).*

(3) To avoid any confusion, we have now retained AOB-*amoA* results in the main text (Figure 3a), moved AOA-*amoA* results into SI (Figure S6), and deleted COM-*amoA* results.

2. Point 16: The authors named the sequencing approach but (1) did they include all the bioinformatics pipelines related to the cleaning and assembly of the reads? Also, (2) how good is PAcBIO compared the MiSeq for *amoA* sequencing (several papers have assembled *amoA* using MiSeq but I am not aware of any

PacBio sequencing yet). (3) All the scripts should be made available with a GitHub link or similar. This is also true for the metagenomics and metatranscriptomics pipelines used.

Answer:

Thanks. (1) We have now added the information of bioinformatics pipelines, including quality control, sequence de-noising, and read assembly, in the revised manuscript.

"Data from the MiSeq PE300 system were processed by demultiplexing and quality filtering the obtained sequences using Fastp (version 0.20.0)⁶², followed by merging with FLASH (version 1.2.11)⁶³. The high-quality sequences were then denoised using the DADA2 pipeline⁶⁴ in QIIME 2 (version 2020.2)⁶⁵ with default parameters. Data from the PacBio system were processed by first obtaining high-fidelity reads from raw sub-reads generated via circular consensus sequencing (CCS) by Single Molecule Real-Time (SMRT, version 11.0)⁶⁶. These high-fidelity reads were then length-filtered and denoised as described above." (Line 660).

"The products were purified, amplified, and sequenced with the NextSeq550 platform (Illumina, USA), finally generating 2×150bp paired-end reads, which were then processed using Fastp (version 0.20.0) to eliminate low-quality sequences and reads containing ambiguous N bases." (Line 696).

"The raw reads were processed with Fastp to trim bases with a quality score <30 and to remove sequences containing adapters and contaminants. The quality-controlled reads were then co-assembled using Megahit (version 1.2.9) with iterative k-mer sizes of 31, 41, 51, 61, 71, 81, and 91." (Line 706).

"The clipped reads were assembled using Megahit (version 1.2.9) to obtain clean contigs, with k-mer sizes ranging from 47 to 97 in steps of 10." (Line 715).

(2) The PacBio system offers significant advantages for sequencing the AOA-*amoA* gene, as it can generate longer read lengths that cover the entire gene with no need to assembly, thereby avoiding potential assembly errors. Several studies have employed the PacBio to analyze AOA-*amoA* communities (Reference 1). This capability is particularly beneficial for exploring the diversity and structural variation of the AOA-*amoA* gene (635 bp) in complex microbial communities in the environment. Additionally, the high-fidelity reads produced by the PacBio system provide high accuracy, significantly reducing sequencing errors (References 2-3). Consequently, we anticipate that this sequencing system will see widespread use in the future.

References:

1. Kou Y, Li C, Tu B, Li J, Li X. 2023. The Responses of Ammonia-Oxidizing Microorganisms to Different Environmental Factors Determine Their Elevational Distribution and Assembly Patterns. *Microbial Ecology* 86:485-496.
2. Wenger AM, Peluso P, Hunkapiller MW. et al., 2019. Accurate circular consensus long-read sequencing improves variant detection and assembly of a human genome. *Nature Biotechnology* 37:1155.
3. Mahmoud M, Huang Y, Sedlazeck FJ. et al., 2024. Utility of long-read sequencing for All of Us. *Nature Communications* 15.

(3) As suggested, we have now uploaded our scripts, sequences, and data used for figure plotting to GitHub. The link (<https://github.com/xuangood/estuarine-plastisphere>) is provided in the revised manuscript.

"Custom scripts, sequences, and data for the figures in this study can be searched on the GitHub (<https://github.com/xuangood/estuarine-plastisphere>). Figures were created by Origin 9.0 and Adobe Illustrator CS6." (Code availability)